# IMPLICIT BIAS OF PER-SAMPLE ADAM ON SEPARABLE DATA: DEPARTURE FROM THE FULL-BATCH REGIME

**Beomhan Baek**[*][†]
Seoul National University
bhbaek2001@snu.ac.kr

**Minhak Song,**[*] **Chulhee Yun**
KAIST
{minhaksong,chulhee.yun}@kaist.ac.kr

## ABSTRACT

Adam (Kingma & Ba, 2015) is the de facto optimizer in deep learning, yet its theoretical understanding remains limited. Prior analyses show that Adam favors solutions aligned with $\ell_\infty$-geometry, but these results are restricted to the full-batch regime. In this work, we study the implicit bias of incremental Adam (using one sample per step) for logistic regression on linearly separable data, and show that its bias can deviate from the full-batch behavior. As an extreme example, we construct datasets on which incremental Adam provably converges to the $\ell_2$-max-margin classifier, in contrast to the $\ell_\infty$-max-margin bias of full-batch Adam. For general datasets, we characterize its bias using a proxy algorithm for the $\beta_2 \to 1$ limit. This proxy maximizes a *data-adaptive* Mahalanobis-norm margin, whose associated covariance matrix is determined by a *data-dependent* dual fixed-point formulation. We further present concrete datasets where this bias reduces to the standard $\ell_2$- and $\ell_\infty$-max-margin classifiers. As a counterpoint, we prove that Signum (Bernstein et al., 2018) converges to the $\ell_\infty$-max-margin classifier for any batch size. Overall, our results highlight that the implicit bias of Adam crucially depends on both the batching scheme and the dataset, while Signum remains invariant.

## 1 INTRODUCTION

The *implicit bias* of optimization algorithms plays a crucial role in training deep neural networks (Vardi, 2023). Even without explicit regularization, these algorithms steer learning toward solutions with specific structural properties. In over-parameterized models, where the training data can be perfectly classified and many global minima exist, the implicit bias dictates which solutions are selected. Understanding this phenomenon has become central to explaining why over-parameterized models often generalize well despite their ability to fit arbitrary labels (Zhang et al., 2017).

A canonical setting for studying implicit bias is linear classification on separable data with logistic loss. In this setup, achieving zero training loss requires the model's weights to diverge to infinity, making the *directional convergence*—which defines the decision boundary—the key object of study. Seminal work by Soudry et al. (2018) establishes that gradient descent (GD) directionally converges to the $\ell_2$-max-margin solution. This foundational result has inspired extensive research extending the analysis to neural networks, alternative optimizers, and other loss functions (Gunasekar et al., 2018b; Ji & Telgarsky, 2019; 2020; Lyu & Li, 2020; Chizat & Bach, 2020; Yun et al., 2021). In this work, we revisit the simplest setting—linear classification on separable data—to examine how the choice of optimizer shapes implicit bias.

Among modern optimization algorithms, Adam (Kingma & Ba, 2015) is one of the most widely used, making its implicit bias particularly important to understand. Zhang et al. (2024a) show that, unlike GD, full-batch Adam converges in direction to the $\ell_\infty$-max-margin solution. This behavior is closely related to sign gradient descent (SignGD), which can be interpreted as normalized steepest descent in the $\ell_\infty$-norm and is also known to converge to the $\ell_\infty$-max-margin direction (Gunasekar et al., 2018a; Fan et al., 2025). Xie et al. (2025) further attribute Adam's empirical success in language model training to its ability to exploit the favorable $\ell_\infty$-geometry of the loss landscape.

---

[*]Authors contributed equally to this paper.
[†]Work done as an undergraduate intern at KAIST.

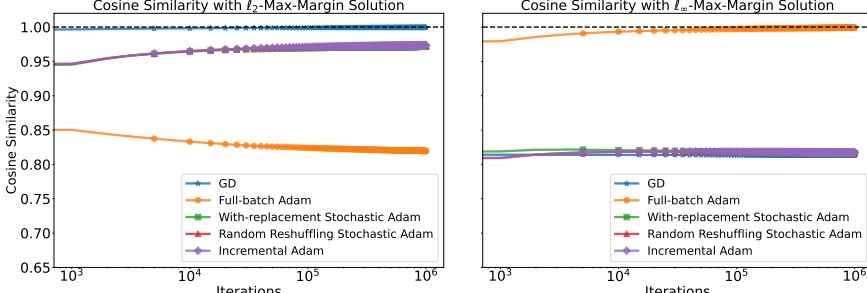

Figure 1: **Mini-batch Adam loses the $\ell_\infty$-max-margin bias of full-batch Adam.** Cosine similarity between the weight vector and the $\ell_2$-max-margin (left) and $\ell_\infty$-max-margin (right) solutions in a linear classification task on 10 data points drawn from the 50-dimensional standard Gaussian. Full-batch Adam with $(\beta_1, \beta_2) = (0.9, 0.95)$ converges to the $\ell_\infty$-max-margin solution, whereas mini-batch variants with a batch size of 1 converge to a different direction (see Section 4 for the detailed characterization). See Appendix C for experimental details.

Yet, prior work on implicit bias in linear classification has almost exclusively focused on the full-batch setting. In contrast, modern training relies on stochastic mini-batches, a regime where theoretical understanding remains limited. Notably, Nacson et al. (2019) show that SGD with an arbitrary batch size preserves the same $\ell_2$-max-margin bias as GD, suggesting that mini-batching may not alter an optimizer's implicit bias. But does this extend to adaptive methods such as Adam?

*Does Adam's characteristic $\ell_\infty$-bias persist under the mini-batch setting?*

Perhaps surprisingly, we find that the answer is *no*. Our experiments (Figure 1) illustrate that when trained on Gaussian data, full-batch Adam converges to the $\ell_\infty$-max-margin direction, whereas Adam variants with a batch size of 1 converge to a different direction, which is even closer to the $\ell_2$-max-margin solution. To explain this phenomenon, we develop a theoretical framework for analyzing the implicit bias of mini-batch Adam and focus on the batch size 1 case as a representative contrast to the full-batch regime. To the best of our knowledge, this work provides the first theoretical evidence describing the *data-dependent* characterization of the implicit bias of Adam with a batch size of 1.

Our contributions are summarized as follows:

- We analyze *incremental Adam*, which processes one sample per step in a cyclic order. Despite its momentum-based updates, we show that its epoch-wise dynamics can be approximated by a recurrence depending only on the current iterate, which becomes a key tool in our analysis (see Section 2).

- We demonstrate a sharp contrast between full-batch and mini-batch Adam using a family of structured datasets, *Scaled Rademacher (SR) data*. On SR data, we prove that incremental Adam converges to the $\ell_2$-max-margin solution, while full-batch Adam converges to the $\ell_\infty$-max-margin solution (see Section 3).

- For general dataset, we introduce a *uniform-averaging proxy* that characterizes the limiting behavior of incremental Adam as $\beta_2 \to 1$. We identify its convergence direction as the solution of a *data-adaptive* margin-maximization problem, induced by a Mahalanobis norm whose covariance matrix is determined by a *data-dependent* dual fixed-point equation. We further present concrete datasets where this bias reduces to the standard $\ell_2$- and $\ell_\infty$-max-margin classifiers (see Section 4).

- Finally, we prove that Signum (SignSGD with momentum; Bernstein et al. (2018)), unlike Adam, maintains its bias toward the $\ell_\infty$-max-margin solution for *any* batch size when the momentum parameter is sufficiently close to 1 (see Section 5).

## 2 HOW CAN WE APPROXIMATE WITHOUT-REPLACEMENT ADAM?

**Notation.** For a vector $\mathbf{v}$, let $\mathbf{v}[k]$ denote its $k$-th entry. For a matrix $\mathbf{M}$, let $\mathbf{M}[i,j]$ denote its $(i,j)$-th entry. We use $\Delta^{N-1}$ to denote the probability simplex in $\mathbb{R}^N$. Let $[N] = \{0, 1, \cdots, N-1\}$ denote the set of the first $N$ non-negative integers. For a PSD matrix $\mathbf{M}$, define the Mahalanobis norm

| **Algorithm 1** `Det-Adam` | **Algorithm 2** `Inc-Adam` |
|---|---|
| **Hyperparams:** Learning rate schedule $\{\eta_t\}_{t=0}^{T-1}$, momentum parameters $\beta_1, \beta_2 \in [0, 1)$ | **Hyperparams:** Learning rate schedule $\{\eta_t\}_{t=0}^{T-1}$, momentum parameters $\beta_1, \beta_2 \in [0, 1)$ |
| **Input:** Initial weight $\mathbf{w}_0$, dataset $\{\mathbf{x}_i\}_{i \in [N]}$ | **Input:** Initial weight $\mathbf{w}_0$, dataset $\{\mathbf{x}_i\}_{i \in [N]}$ |
| 1: Initialize momentum $\mathbf{m}_{-1} = \mathbf{v}_{-1} = \mathbf{0}$ | 1: Initialize momentum $\mathbf{m}_{-1} = \mathbf{v}_{-1} = \mathbf{0}$ |
| 2: **for** $t = 0, 1, 2, \ldots, T-1$ **do** | 2: **for** $t = 0, 1, 2, \ldots, T-1$ **do** |
| 3: $\quad \mathbf{g}_t \leftarrow \nabla \mathcal{L}(\mathbf{w}_t)$ | 3: $\quad \mathbf{g}_t \leftarrow \nabla \mathcal{L}_{i_t}(\mathbf{w}_t), \quad i_t = t \bmod N$ |
| 4: $\quad \mathbf{m}_t \leftarrow \beta_1 \mathbf{m}_{t-1} + (1 - \beta_1)\mathbf{g}_t$ | 4: $\quad \mathbf{m}_t \leftarrow \beta_1 \mathbf{m}_{t-1} + (1 - \beta_1)\mathbf{g}_t$ |
| 5: $\quad \mathbf{v}_t \leftarrow \beta_2 \mathbf{v}_{t-1} + (1 - \beta_2)\mathbf{g}_t^2$ | 5: $\quad \mathbf{v}_t \leftarrow \beta_2 \mathbf{v}_{t-1} + (1 - \beta_2)\mathbf{g}_t^2$ |
| 6: $\quad \mathbf{w}_{t+1} \leftarrow \mathbf{w}_t - \eta_t \frac{\mathbf{m}_t}{\sqrt{\mathbf{v}_t}}$ | 6: $\quad \mathbf{w}_{t+1} \leftarrow \mathbf{w}_t - \eta_t \frac{\mathbf{m}_t}{\sqrt{\mathbf{v}_t}}$ |
| 7: **end for** | 7: **end for** |
| 8: **return** $\mathbf{w}_T$ | 8: **return** $\mathbf{w}_T$ |

as $\|\mathbf{x}\|_{\mathbf{M}} \triangleq \sqrt{\mathbf{x}^\top \mathbf{M} \mathbf{x}}$. For vectors, $\sqrt{\cdot}$, $(\cdot)^2$, and $\div$ operations are applied entry-wise unless stated otherwise. Given two functions $f(t), g(t)$, we denote $f(t) = \mathcal{O}(g(t))$ if there exist $C, T > 0$ such that $t \geq T$ implies $|f(t)| \leq C|g(t)|$. For two vectors $\mathbf{v}$ and $\mathbf{w}$, we denote $\mathbf{v} \propto \mathbf{w}$ if $\mathbf{v} = c \cdot \mathbf{w}$ for a *positive* scalar $c > 0$. Let $r = a \bmod b$ denote the remainder when dividing $a$ by $b$, i.e., $0 \leq r < b$.

**Algorithms.** We focus on incremental Adam (`Inc-Adam`), which processes mini-batch gradients sequentially from indices $0$ to $N-1$ in each epoch. Studying `Inc-Adam` provides a tractable way to understand the implicit bias of mini-batch Adam under various sampling schemes: our experiments show that `Inc-Adam`'s directional convergence closely aligns with that of Adam with a batch size of $1$ under both sampling with replacement and random-reshuffling. Sharing the same mini-batch accumulation mechanism, `Inc-Adam` serves as a faithful surrogate for theoretical analysis. Pseudocodes for `Inc-Adam` and full-batch deterministic Adam (`Det-Adam`) are given in Algorithms 1 and 2.

**Stability Constant $\epsilon$.** In practice, we often consider an additional $\epsilon$ term for numerical stability and update with $\mathbf{w}_{t+1} = \mathbf{w}_t - \eta_t \frac{\mathbf{m}_t}{\sqrt{\mathbf{v}_t} + \epsilon}$. In fact, when investigating the asymptotic behavior of Adam, the stability constant significantly affects the converging direction, since $\mathbf{v}_t \to 0$ as $t \to \infty$ and $\epsilon$ dominates $\mathbf{v}_t$. Wang et al. (2021) investigate RMSprop and Adam with the stability constant, yielding their directional convergence to $\ell_2$-max-margin solution. More recent approaches, however, point out that analyzing Adam without the stability constant is more suitable for describing its intrinsic behavior (Xie & Li, 2024; Zhang et al., 2024a; Fan et al., 2025). We adopt this view and consider the version of Adam without $\epsilon$.

**Problem Settings.** We primarily focus on binary linear classification tasks. To be specific, training data are given by $\{(\mathbf{x}_i, y_i)\}_{i \in [N]}$, where $\mathbf{x}_i \in \mathbb{R}^d$, $y_i \in \{-1, +1\}$, and $N$ is the number of data points. We aim to find a linear classifier $\mathbf{w}$ which minimizes the loss

$$\mathcal{L}(\mathbf{w}) = \frac{1}{N} \sum_{i \in [N]} \ell(y_i \langle \mathbf{w}, \mathbf{x}_i \rangle) = \frac{1}{N} \sum_{i \in [N]} \mathcal{L}_i(\mathbf{w}),$$

where $\ell : \mathbb{R} \to \mathbb{R}$ is a surrogate loss for classification accuracy and $\mathcal{L}_i(\mathbf{w}) = \ell(y_i \langle \mathbf{w}, \mathbf{x}_i \rangle)$ denotes the loss value on the $i$-th data point. Without loss of generality, we assume $y_i = +1$, since we can newly define $\tilde{\mathbf{x}}_i = y_i \mathbf{x}_i$. In this paper, we consider two loss functions $\ell \in \{\ell_{\exp}, \ell_{\log}\}$, where $\ell_{\exp}(z) = \exp(-z)$ denotes the exponential loss and $\ell_{\log}(z) = \log(1 + e^{-z})$ denotes the logistic loss. Given a sequence of vectors $\{\mathbf{v}_t\}_{t=0}^{\infty}$, for notational simplicity, let $\mathbf{v}_r^s \triangleq \mathbf{v}_{rN+s}$ denote the $s$-th element in the $r$-th group of $N$ consecutive terms.

To investigate the implicit bias of Adam variants, we make the following assumptions.

**Assumption 2.1** (Separable data). There exists $\mathbf{w} \in \mathbb{R}^d$ such that $\mathbf{w}^\top \mathbf{x}_i > 0$, $\forall i \in [N]$.

**Assumption 2.2** (Nonzero coordinates). For all $i \in [N]$ and $k \in [d]$, $\mathbf{x}_i[k] \neq 0$.

**Assumption 2.3** (Learning rate schedule). The sequence of learning rates, $\{\eta_t\}_{t=1}^{\infty}$, satisfies

(a) $\{\eta_t\}_{t=1}^{\infty}$ is decreasing in $t$, $\sum_{t=1}^{\infty} \eta_t = \infty$, and $\lim_{t \to \infty} \eta_t = 0$.

(b) For all $\beta \in (0, 1), c_1 > 0$, there exist $t_1 \in \mathbb{N}_+, c_2 > 0$ such that $\sum_{\tau=0}^{t} \beta^\tau (e^{c_1 \sum_{\tau'=1}^{\tau} \eta_{t-\tau'}} - 1) \leq c_2 \eta_t$ for all $t \geq t_1$.

Assumption 2.1 guarantees linear separability of the data. Assumption 2.2 holds with probability 1 if the data is sampled from a continuous distribution. Assumption 2.3 originates from Zhang et al. (2024a) and it takes a crucial role to bound the error from the movement of weights. We note that a polynomial decaying learning rate schedule $\eta_t = (t + 2)^{-a}, a \in (0, 1]$ satisfies Assumption 2.3, which is proved by Lemma C.1 in Zhang et al. (2024a).

The dependence of the Adam update on the full gradient history makes its asymptotic analysis largely intractable. We address this challenge with the following propositions, which show that the *epoch-wise* updates of Inc-Adam and the updates of Det-Adam can be approximated by a function that depends only on the current iterate. Detailed proofs are deferred to Appendix D.

**Proposition 2.4.** *Let $\{\mathbf{w}_t\}_{t=0}^{\infty}$ be the iterates of* Det-Adam *with $\beta_1 \leq \beta_2$. Then, under Assumptions 2.2 and 2.3, if $\lim_{t \to \infty} \frac{\eta_t^{1/2} \mathcal{L}(\mathbf{w}_t)}{|\nabla \mathcal{L}(\mathbf{w}_t)[k]|} = 0$, then the update of k-th coordinate $\mathbf{w}_{t+1}[k] - \mathbf{w}_t[k]$ can be represented by*

$$\mathbf{w}_{t+1}[k] - \mathbf{w}_t[k] = -\eta_t \left( \mathrm{sign}(\nabla \mathcal{L}(\mathbf{w}_t)[k]) + \epsilon_t \right), \tag{1}$$

*for some $\lim_{t \to \infty} \epsilon_t = 0$.*

**Proposition 2.5.** *Let $\{\mathbf{w}_t\}_{t=0}^{\infty}$ be the iterates of* Inc-Adam *with $\beta_1 \leq \beta_2$. Then, under Assumptions 2.2 and 2.3, the epoch-wise update $\mathbf{w}_{r+1}^0 - \mathbf{w}_r^0$ can be represented by*

$$\mathbf{w}_{r+1}^0 - \mathbf{w}_r^0 = -\eta_{rN} \left( C_{inc}(\beta_1, \beta_2) \sum_{i \in [N]} \frac{\sum_{j \in [N]} \beta_1^{(i,j)} \nabla \mathcal{L}_j(\mathbf{w}_r^0)}{\sqrt{\sum_{j \in [N]} \beta_2^{(i,j)} \nabla \mathcal{L}_j(\mathbf{w}_r^0)^2}} + \boldsymbol{\epsilon}_r \right), \tag{2}$$

*where $\beta_1^{(i,j)} = \beta_1^{(i-j) \bmod N}, \beta_2^{(i,j)} = \beta_2^{(i-j) \bmod N}, C_{inc}(\beta_1, \beta_2) = \frac{1-\beta_1}{1-\beta_1^N} \sqrt{\frac{1-\beta_2^N}{1-\beta_2}}$ is a function of $\beta_1, \beta_2$, and $\lim_{r \to \infty} \boldsymbol{\epsilon}_r = \mathbf{0}$. If $\eta_t = (t+2)^{-a}$ for some $a \in (0, 1]$, then $\|\boldsymbol{\epsilon}_r\|_{\infty} = \mathcal{O}(r^{-a/2})$.*

**Discrepancy between Det-Adam and Inc-Adam.** Propositions 2.4 and 2.5 reveal a fundamental discrepancy between the behavior of Det-Adam and one of Inc-Adam. Proposition 2.4 demonstrates that Det-Adam can be approximated by SignGD, which has been reported by previous works (Balles & Hennig, 2018; Zou et al., 2023). Note that the condition is not satisfied when $\nabla \mathcal{L}(\mathbf{w}_t)[k]$ decays at a rate on the order of $\eta_t^{1/2} \mathcal{L}(\mathbf{w}_t)$, which often calls for a more detailed analysis (see Zhang et al. (2024a, Lemma 6.2)). Such an analysis establishes that Det-Adam asymptotically finds an $\ell_{\infty}$-max-margin solution, a property that holds regardless of the choice of momentum hyperparameters satisfying $\beta_1 \leq \beta_2$ (Zhang et al., 2024a).

In stark contrast, our epoch-wise analysis illustrates that Inc-Adam's updates more closely follow a weighted, preconditioned GD. This makes its behavior highly dependent on both the momentum parameters and the current iterate. The discrepancy originates from the use of mini-batch gradients; the preconditioner tracks the sum of squared mini-batch gradients, which diverges from the squared full-batch gradient. This discrepancy results in the highly complex dynamics of Inc-Adam, which are investigated in subsequent sections.

## 3  WARMUP: STRUCTURED DATA

**Eliminating Coordinate-Adaptivity.** To highlight the fundamental discrepancy between Det-Adam and Inc-Adam, we construct a scenario that completely nullifies the coordinate-wise adaptivity of Inc-Adam's preconditioner by introducing the following family of structured datasets.

**Definition 3.1.** *We define Scaled Rademacher (SR) data as a set of vectors $\{\mathbf{x}_i\}_{i \in [N]}$ which satisfy $|\mathbf{x}_i[k]| = |\mathbf{x}_i[l]|, \forall k, l \in [d]$, for each $i \in [N]$. We also assume that SR data satisfy Assumptions 2.1 and 2.2, unless otherwise specified.*

Applying Proposition 2.5 to the SR dataset, we obtain the following corollary.

**Corollary 3.2.** *Consider* Inc-Adam *iterates $\{\mathbf{w}_t\}_{t=0}^{\infty}$ on SR data. Then, under Assumptions 2.2 and 2.3, the epoch-wise update $\mathbf{w}_{r+1}^0 - \mathbf{w}_r^0$ can be approximated by weighted normalized GD, i.e.,*

$$\mathbf{w}_{r+1}^0 - \mathbf{w}_r^0 = -\eta_{rN} \left( \sum_{i \in [N]} \frac{a_i(r)}{\|\nabla \mathcal{L}(\mathbf{w}_r^0)\|_2} \nabla \mathcal{L}_i(\mathbf{w}_r^0) + \boldsymbol{\epsilon}_r \right), \tag{3}$$

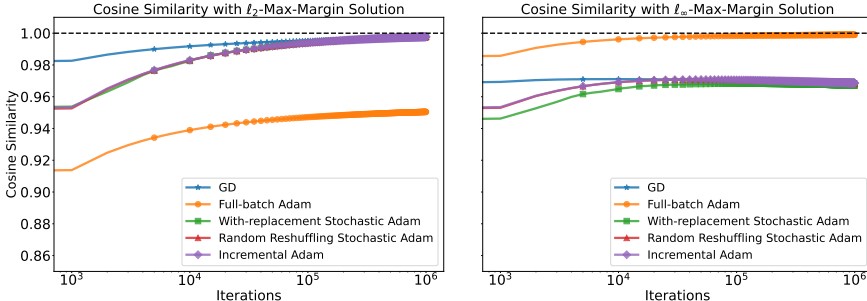

Figure 2: **Mini-batch Adam converges to the $\ell_2$-max-margin solution on the SR dataset.** We train on the dataset $\mathbf{x}_0 = (1, 1, 1, 1)$, $\mathbf{x}_1 = (2, 2, 2, -2)$, $\mathbf{x}_2 = (3, 3, -3, -3)$, and $\mathbf{x}_3 = (4, -4, 4, -4)$. Variants of mini-batch Adam with a batch size of 1 consistently converge to the $\ell_2$-max-margin direction, while full-batch Adam converges to the $\ell_\infty$-max-margin direction.

*where $\lim_{r \to \infty} \boldsymbol{\epsilon}_r = \mathbf{0}$ and $c_1 \leq a_i(r) \leq c_2$ for some positive constants $c_1, c_2$ only depending on $\beta_1, \beta_2, \{\mathbf{x}_i\}_{i \in [N]}$. If $\eta_t = (t+2)^{-a}$ for some $a \in (0, 1]$, then $\|\boldsymbol{\epsilon}_r\|_\infty = \mathcal{O}(r^{-a/2})$.*

Although using a structured dataset simplifies the denominator in Equation (2), the dynamics are still governed by weighted GD, which requires a careful analysis. Prior work studies the implicit bias of weighted GD, particularly in the context of importance weighting (Xu et al., 2021; Zhai et al., 2023), but these analysis typically assume that the weights are constant or convergent. In our setting, the weight $a_i(r)$ varies with the epoch count $r$. We address this challenge and characterize the implicit bias of `Inc-Adam` on the SR data as follows.

**Theorem 3.3.** *Consider `Inc-Adam` iterates $\{\mathbf{w}_t\}_{t=0}^\infty$ with $\beta_1 \leq \beta_2$ on SR data under Assumptions 2.1 to 2.3. If (a) $\mathcal{L}(\mathbf{w}_t) \to 0$ as $t \to \infty$ and (b) $\eta_t = (t+2)^{-a}$ for $a \in (2/3, 1]$, then it satisfies*

$$\lim_{t \to \infty} \frac{\mathbf{w}_t}{\|\mathbf{w}_t\|_2} = \hat{\mathbf{w}}_{\ell_2},$$

*where $\hat{\mathbf{w}}_{\ell_2}$ denotes the (unique) $\ell_2$-max-margin solution of SR data $\{\mathbf{x}_i\}_{i \in [N]}$.*

The analysis in Theorem 3.3 relies on Corollary 3.2, which ensures that the weights $a_i(r)$ are bounded by two positive constants, $c_1$ and $c_2$. This condition is crucial to prevent any individual data from having a vanishing contribution, which could cause the `Inc-Adam` iterates to deviate from the $\ell_2$-max-margin direction. Furthermore, the controlled learning rate schedule is key to bounding the $\epsilon_r$ term in our analysis. The proof and further discussion are deferred to Appendix E. As shown in Figure 2, our experiments on SR data confirm that mini-batch Adam with a batch size of 1 converges in direction to the $\ell_2$-max-margin classifier, in contrast to the $\ell_\infty$-bias of full-batch Adam.

Notably, Theorem 3.3 holds for any choice of momentum hyperparameters satisfying $\beta_1 \leq \beta_2$; see Figure 9 in Appendix B for empirical evidence. This invariance of the bias arises from the structure of SR data, which removes the coordinate adaptivity that momentum hyperparameters would normally affect. For general datasets, the invariance no longer holds; the adaptivity persists and varies with the choice of momentum hyperparameters, as discussed in Appendix A. In the next section, we introduce a proxy algorithm to study the regime where $\beta_2$ is close to 1 and characterize its implicit bias.

## 4 GENERALIZATION: ADAMPROXY

**Uniform-Averaging Proxy.** A key challenge in characterizing the limiting predictor of `Inc-Adam` for a general datasets is that its approximated update (Proposition 2.5) is difficult to analyze directly. To address this, we study a simpler *uniform-averaging* proxy, derived in Proposition 4.1 under the limit $\beta_2 \to 1$. This approximation is well-motivated, as $\beta_2$ is typically chosen close to 1 in practice.

**Proposition 4.1.** *Let $\{\mathbf{w}_t\}_{t=0}^{\infty}$ be the iterates of* Inc-Adam *with $\beta_1 \leq \beta_2$. Then, under Assumptions 2.2 and 2.3, the epoch-wise update $\mathbf{w}_{r+1}^0 - \mathbf{w}_r^0$ can be expressed as*

$$\mathbf{w}_{r+1}^0 - \mathbf{w}_r^0 = -\eta_{rN}\left(\sqrt{\frac{1-\beta_2^N}{1-\beta_2}}\frac{\nabla\mathcal{L}(\mathbf{w}_r^0)}{\sqrt{\sum_{i=1}^{N}\nabla\mathcal{L}_i(\mathbf{w}_r^0)^2}} + \boldsymbol{\epsilon}_{\beta_2}(r)\right),$$

*where $\limsup_{r\to\infty}\|\boldsymbol{\epsilon}_{\beta_2}(r)\|_{\infty} \leq \epsilon(\beta_2)$ and $\lim_{\beta_2\to 1}\epsilon(\beta_2) = 0$.*

**Definition 4.2.** We define an update of AdamProxy as

$$\boldsymbol{\delta}_t = \mathrm{Prx}(\mathbf{w}_t) \triangleq \frac{\nabla\mathcal{L}(\mathbf{w}_t)}{\sqrt{\sum_{i=1}^{N}\nabla\mathcal{L}_i(\mathbf{w}_t)^2}}, \tag{4}$$

$$\mathbf{w}_{t+1} = \mathbf{w}_t - \eta_t\boldsymbol{\delta}_t.$$

**Proposition 4.3** (Loss convergence). *Under Assumptions 2.1 and 2.2, there exists a positive constant $\eta > 0$ depending only on the dataset $\{\mathbf{x}_i\}_{i\in[N]}$, such that if the learning rate schedule satisfies $\eta_t \leq \eta$ and $\sum_{t=0}^{\infty}\eta_t = \infty$, then* AdamProxy *iterates minimize the loss, i.e., $\lim_{t\to\infty}\mathcal{L}(\mathbf{w}_t) = 0$.*

To characterize the convergence direction of AdamProxy, we further assume that the weights $\{\mathbf{w}_t\}_{t=0}^{\infty}$ and the updates $\{\boldsymbol{\delta}_t\}_{t=0}^{\infty}$ converge in direction.

**Assumption 4.4.** We assume that: (a) learning rates $\{\eta_t\}_{t=0}^{\infty}$ satisfy the conditions in Proposition 4.3, (b) $\exists\lim_{t\to\infty}\frac{\mathbf{w}_t}{\|\mathbf{w}_t\|_2} \triangleq \hat{\mathbf{w}}$, and (c) $\exists\lim_{t\to\infty}\frac{\boldsymbol{\delta}_t}{\|\boldsymbol{\delta}_t\|_2} \triangleq \hat{\boldsymbol{\delta}}$.

**Lemma 4.5.** *Under Assumptions 2.1, 2.2 and 4.4, there exists $\mathbf{c} = (c_0, \cdots, c_{N-1}) \in \Delta^{N-1}$ such that the limit direction $\hat{\mathbf{w}}$ of* AdamProxy *satisfies*

$$\hat{\mathbf{w}} \propto \frac{\sum_{i\in[N]}c_i\mathbf{x}_i}{\sqrt{\sum_{i\in[N]}c_i^2\mathbf{x}_i^2}}, \tag{5}$$

*and $c_i = 0$ for $i \notin S$, where $S = \arg\min_{i\in[N]}\hat{\mathbf{w}}^{\top}\mathbf{x}_i$ is the index set of support vectors of $\hat{\mathbf{w}}$.*

Prior research on the implicit bias of optimizers has predominantly focused on characterizing the convergence direction through the formulation of a corresponding optimization problem. For example, the solution to the $\ell_p$-max-margin problem,

$$\max_{\mathbf{w}\in\mathbb{R}^d}\frac{1}{2}\|\mathbf{w}\|_p^2 \quad \text{subject to} \quad \mathbf{w}^{\top}\mathbf{x}_i - 1 \geq 0,\ \forall i \in [N],$$

describes the implicit bias of the steepest descent algorithm with respect to the $\ell_p$-norm in linear classification tasks (Gunasekar et al., 2018a). However, Equation (5) does not correspond to the KKT conditions of a conventional optimization problem. To address this, we introduce a novel framework to describe the convergence direction, based on a *parametric* optimization problem combined with *fixed-point analysis* between dual variables.

**Definition 4.6.** Given $\mathbf{c} \in \Delta^{N-1}$, we define a parametric optimization problem $P_{\mathrm{Adam}}(\mathbf{c})$ as

$$P_{\mathrm{Adam}}(\mathbf{c}):\ \min_{\mathbf{w}\in\mathbb{R}^d}\frac{1}{2}\|\mathbf{w}\|_{\mathbf{M}(\mathbf{c})}^2 \quad \text{subject to} \quad \mathbf{w}^{\top}\mathbf{x}_i - 1 \geq 0,\ \forall i \in [N], \tag{6}$$

where $\mathbf{M}(\mathbf{c}) = \mathrm{diag}(\sqrt{\sum_{j\in[N]}c_j^2\mathbf{x}_j^2}) \in \mathbb{R}^{d\times d}$. We define $\mathbf{p}(\mathbf{c})$ as the set of global optimizers of $P_{\mathrm{Adam}}(\mathbf{c})$ and $\mathbf{d}(\mathbf{c})$ as the set of corresponding dual solutions. Let $S(\mathbf{w}) = \{i \in [N] \mid \mathbf{w}^{\top}\mathbf{x}_i = 1\}$ denote the index set for the support vectors for any $\mathbf{w} \in \mathbf{p}(\mathbf{c})$.

**Assumption 4.7** (Linear Independence Constraint Qualification). For any $\mathbf{c} \in \Delta^{N-1}$ and $\mathbf{w} \in \mathbf{p}(\mathbf{c})$, the set of support vectors $\{\mathbf{x}_i\}_{i\in S(\mathbf{w})}$ is linearly independent.

Assumption 4.7 ensures the uniqueness of the dual solution for $P_{\mathrm{Adam}}(\mathbf{c})$, which is essential for our framework. This assumption naturally holds in the overparameterized regime where the dataset $\{\mathbf{x}_i\}_{i\in[N]}$ consists of linearly independent vectors.

---

**Algorithm 3** Fixed-Point Iteration

---

**Input:** Dataset $\{\mathbf{x}_i\}_{i \in [N]}$, initialization $\mathbf{c}_0 \in \Delta^{N-1}$, threshold $\epsilon_{\text{thr}} > 0$
1: **repeat**
2:    Solve $P_{\text{Adam}}(\mathbf{c}_0) : \min \frac{1}{2}\|\mathbf{w}\|_{\mathbf{M}(\mathbf{c}_0)}$ subject to $\mathbf{w}^\top \mathbf{x}_i - 1 \geq 0, \forall i \in [N]$
3:    $\mathbf{w} \leftarrow \text{Primal}(P_{\text{Adam}})$
4:    $\mathbf{c}_1 \leftarrow \text{Dual}(P_{\text{Adam}})$
5:    $\delta \leftarrow \|\mathbf{c}_1 - \mathbf{c}_0\|_2$
6:    $\mathbf{c}_0 \leftarrow \mathbf{c}_1$
7: **until** $\delta \leq \epsilon_{\text{thr}}$
8: **return** $\mathbf{w}$

---

**Theorem 4.8.** *Under Assumptions 2.1 and 4.7, $P_{Adam}(\mathbf{c})$ admits unique primal and dual solutions, so that $\mathbf{p}(\mathbf{c})$ and $\mathbf{d}(\mathbf{c})$ can be regarded as vector-valued functions. Moreover, under Assumptions 2.1, 2.2, 4.4 and 4.7, the following hold:*

*(a) $\mathbf{p} : \Delta^{N-1} \to \mathbb{R}^d$ is continuous.*

*(b) $\mathbf{d} : \Delta^{N-1} \to \mathbb{R}^N_{\geq 0}\backslash\{\mathbf{0}\}$ is continuous. Consequently, the map $T(\mathbf{c}) \triangleq \frac{\mathbf{d}(\mathbf{c})}{\|\mathbf{d}(\mathbf{c})\|_1}$ is continuous.*

*(c) The map $T : \Delta^{N-1} \to \Delta^{N-1}$ admits at least one fixed point.*

*(d) There exists $\mathbf{c}^* \in \{\mathbf{c} \in \Delta^{N-1} : T(\mathbf{c}) = \mathbf{c}\}$ such that the convergence direction $\hat{\mathbf{w}}$ of* `AdamProxy` *is proportional to $\mathbf{p}(\mathbf{c}^*)$.*

Theorem 4.8 shows how the parametric optimization problem $P_{\text{Adam}}(\mathbf{c})$ captures the characterization from Lemma 4.5. The central idea is to treat the vector $\mathbf{c}$ from Equation (5) in a dual role: as both the parameter of $P_{\text{Adam}}(\mathbf{c})$ and as its corresponding dual variable. The convergence direction is then identified at the point where these two roles coincide, leading naturally to the fixed-point formulation. Detailed proofs are deferred to Appendix F.

To computationally identify the convergence direction of `AdamProxy` based on Theorem 4.8, we introduce the fixed-point iteration described in Algorithm 3. Numerical experiments confirm that the resulting solution accurately predicts the limiting directions of both `AdamProxy` and `Inc-Adam` (see Example 4.10). However, the complexity of the mapping $T$ makes it challenging to establish a formal convergence guarantee for Algorithm 3. A rigorous analysis is left for future work.

**Data-dependent Limit Directions.** We illustrate how structural properties of the data shape the limit direction of `AdamProxy` through three case studies. These examples demonstrate that both `AdamProxy` and `Inc-Adam` converge to directions that are intrinsically data-dependent.

**Example 4.9** (Revisiting SR data)**.** For SR data $\{\mathbf{x}_i\}_{i \in [N]}$, the matrix $\mathbf{M}(\mathbf{c})$ reduces to a scaled identity for every $\mathbf{c} \in \Delta^{N-1}$. Hence, the parametric optimization problem $P_{\text{Adam}}(\mathbf{c})$ narrows down to the standard SVM formulation

$$\min \frac{1}{2}\|\mathbf{w}\|_2^2 \quad \text{subject to} \quad \mathbf{w}^\top \mathbf{x}_i - 1 \geq 0, \ \forall i \in [N].$$

Therefore, Theorem 4.8 implies that `AdamProxy` converges to the $\ell_2$-max-margin solution. This finding is consistent with Theorem 3.3, which establishes the directional convergence of `Inc-Adam` on SR data. Together, these results indicate that the structural property of SR data that eliminates coordinate adaptivity persists in the limit $\beta_2 \to 1$.

**Example 4.10** (Revisiting Gaussian data)**.** We next validate the fixed-point characterization in Theorem 4.8 using the Gaussian dataset from Figure 1. The theoretical limit direction is given by the fixed point of $T$ defined in Theorem 4.8, which we compute via the iteration in Algorithm 3. As shown in Figure 3, both `AdamProxy` and mini-batch Adam variants with a batch size of 1 converge to the predicted solution, confirming the fixed-point formulation and the effectiveness of Algorithm 3. Furthermore, this demonstrates that, depending on the dataset, the limit direction of mini-batch Adam may differ from both the conventional $\ell_2$- and $\ell_\infty$-max-margin solutions.

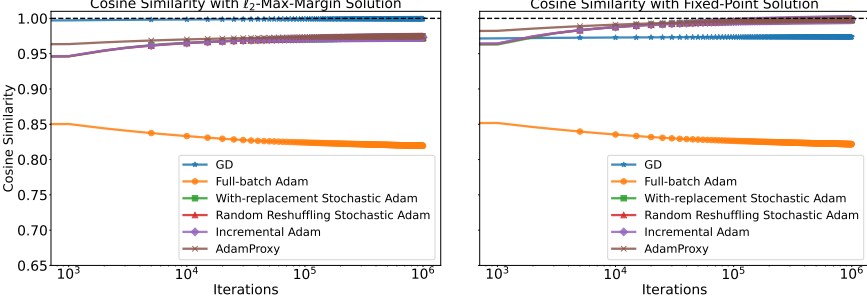

Figure 3: **Mini-batch Adam converges to the fixed-point solution on Gaussian data.** We train on the same Gaussian data as in Figure 1 and plot the cosine similarity of the weight vector with the $\ell_2$-max-margin solution (left) and the fixed-point solution (right). The results show that variants of mini-batch Adam with a batch size of 1 converge to the fixed-point solution obtained by Algorithm 3, consistent with our theoretical prediction (Theorem 4.8).

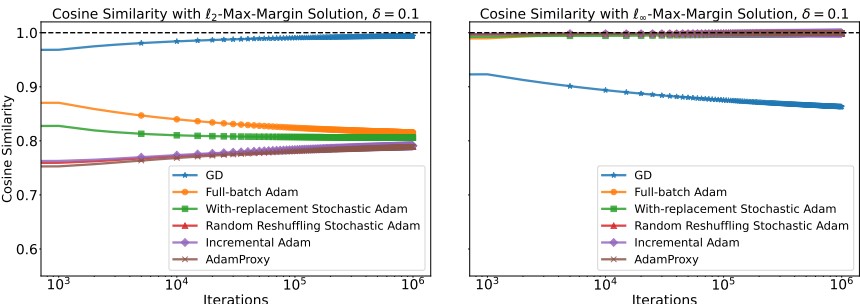

Figure 4: **Mini-batch Adam converges to the $\ell_\infty$-max-margin solution on a shifted-diagonal dataset.** We train on the dataset $\mathbf{x}_0 = (1, \delta, \delta, \delta)$, $\mathbf{x}_1 = (\delta, 2, \delta, \delta)$, $\mathbf{x}_2 = (\delta, \delta, 4, \delta)$, and $\mathbf{x}_3 = (\delta, \delta, \delta, 8)$ with $\delta = 0.1$. Variants of mini-batch Adam with a batch size of 1 converge to the $\ell_\infty$-max-margin direction.

**Example 4.11** (Shifted-diagonal data). Consider $N = d$ and $\{\mathbf{x}_i\}_{i \in [d]} \subseteq \mathbb{R}^d$ with $\mathbf{x}_i = x_i \mathbf{e}_i + \delta \sum_{j \neq i} \mathbf{e}_j$ for some $\delta > 0$ and $0 < x_0 < \cdots < x_{d-1}$. Then, the $\ell_\infty$-max-margin problem

$$\min \frac{1}{2} \|\mathbf{w}\|_\infty^2 \quad \text{subject to} \quad \mathbf{w}^\top \mathbf{x}_i \geq 1, \ \forall i \in [N]$$

has the solution $\hat{\mathbf{w}}_\infty = \left( \frac{1}{x_0 + (d-1)\delta}, \cdots, \frac{1}{x_0 + (d-1)\delta} \right) \in \mathbb{R}^d$. Notice that $\mathbf{c}^* = (1, 0, \cdots, 0) \in \Delta^{d-1}$ is a fixed point of $T$ in Theorem 4.8, and $\hat{\mathbf{w}}_\infty = \mathbf{p}(\mathbf{c}^*)$; detailed calculations are deferred to Appendix F. Consequently, the $\ell_\infty$-max-margin solution serves a candidate for the convergence direction of `AdamProxy` as predicted by Theorem 4.8. To verify this, we run `AdamProxy` and mini-batch Adam variants with a batch size of 1 on shifted-diagonal data given by $\mathbf{x}_0 = (1, \delta, \delta, \delta)$, $\mathbf{x}_1 = (\delta, 2, \delta, \delta)$, $\mathbf{x}_2 = (\delta, \delta, 4, \delta)$, and $\mathbf{x}_3 = (\delta, \delta, \delta, 8)$ with $\delta = 0.1$. As shown in Figure 4, all mini-batch Adam variants converge to the $\ell_\infty$-max-margin solution, consistent with the theoretical prediction.

A key limitation of our analysis is that it assumes $\beta_2 \to 1$ and a batch size of 1. In Appendix A, we provide a preliminary analysis of how batch size and momentum hyperparameters affect the implicit bias of mini-batch Adam. In particular, Appendix A.2 explains why our fixed-point framework does not directly extend to general $\beta_2 < 1$.

## 5 SIGNUM CAN RETAIN $\ell_\infty$-BIAS UNDER MINI-BATCH REGIME

In the previous section, we showed that Adam loses its $\ell_\infty$-max-margin bias under mini-batch updates, drifting toward data-dependent solutions. This motivates the search for a *SignGD-type* algorithm that preserves $\ell_\infty$-geometry even in the mini-batch regime. We prove that Signum (Bernstein et al.,

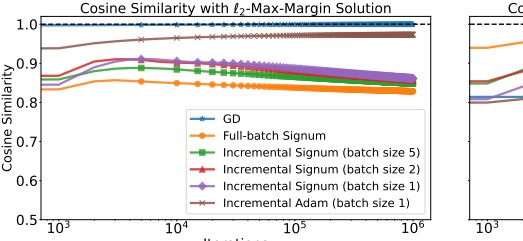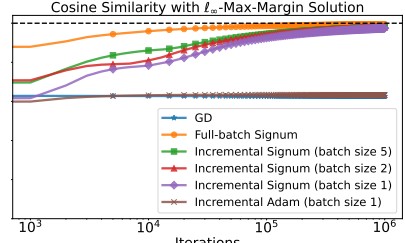

Figure 5: **Mini-batch Signum converges to the $\ell_\infty$-max-margin solution.** We train on the same Gaussian data ($N = 10$, $d = 50$) as in Figure 1, using full-batch Signum and incremental Signum with $\beta = 0.99$, for batch sizes $b \in \{5, 2, 1\}$. Across all batch sizes, incremental Signum consistently converges to the $\ell_\infty$-max-margin solution, in sharp contrast to incremental Adam.

2018) satisfies this property: with momentum close to 1, its iterates converge to the $\ell_\infty$-max-margin direction for *arbitrary* mini-batch sizes.

**Theorem 5.1.** *Let $\delta > 0$. Then there exists $\epsilon > 0$ such that the iterates $\{\mathbf{w}_t\}_{t=0}^{\infty}$ of* Inc-Signum *(Algorithm 4) with batch size $b$ and momentum $\beta \in (1 - \epsilon, 1)$, under Assumptions 2.1 and 2.3, satisfy*

$$\liminf_{t \to \infty} \frac{\min_{i \in [N]} \mathbf{x}_i^\top \mathbf{w}_t}{\|\mathbf{w}_t\|_\infty} \geq \gamma_\infty - \delta, \tag{7}$$

*where*

$$\gamma_\infty \triangleq \max_{\|\mathbf{w}\|_\infty \leq 1} \min_{i \in [N]} \mathbf{w}^\top \mathbf{x}_i, \quad D \triangleq \max_{i \in [N]} \|\mathbf{x}_i\|_1,$$

*and*

$$\epsilon = \frac{1}{2D \cdot \frac{N}{b}(\frac{N}{b} - 1)} \min\{\delta, \tfrac{\gamma_\infty}{2}\} \quad \text{if } b < N, \qquad \epsilon = 1 \quad \text{if } b = N.$$

Theorem 5.1 demonstrates that, unlike Adam, Signum preserves $\ell_\infty$-max-margin bias for any batch size, provided momentum is sufficiently close to 1. This generalizes the full-batch result of Fan et al. (2025). Moreover, the requirement $\beta \approx 1$ is not merely technical but *necessary* in the mini-batch setting to ensure convergence to the $\ell_\infty$-max-margin solution; see Figure 10 in Appendix B for empirical evidence. As shown in Figure 5, our experiments on the Gaussian dataset from Figure 1 show that Inc-Signum ($\beta = 0.99$) maintains $\ell_\infty$-bias, regardless of the choice of batch size. Proofs and further discussion are deferred to Appendix G.

## 6 RELATED WORK

**Understanding Adam.** Adam (Kingma & Ba, 2015) and its variant AdamW (Loshchilov & Hutter, 2019) are standard optimizers for large-scale models, particularly in domains like language modeling where SGD often falls short. A significant body of research seeks to explain this empirical success. One line focuses on convergence guarantees. The influential work of Reddi et al. (2018) demonstrates Adam's failure to converge on certain convex problems, which motivates numerous studies establishing its convergence under various practical conditions (Défossez et al., 2022; Zhang et al., 2022; Li et al., 2023; Hong & Lin, 2024; Ahn & Cutkosky, 2024; Jin et al., 2025). Another line investigates why Adam outperforms SGD, attributing its success to robustness against heavy-tailed gradient noise (Zhang et al., 2020), better adaptation to ill-conditioned landscapes (Jiang et al., 2023; Pan & Li, 2023), and effectiveness in contexts of heavy-tailed class imbalance or gradient/Hessian heterogeneity (Kunstner et al., 2024; Zhang et al., 2024b; Tomihari & Sato, 2025). Ahn et al. (2024) further observe that this performance gap arises even in shallow linear Transformers. Recent works investigate how the choice of momentum hyperparameters (Orvieto & Gower, 2025) and the rotation operation (Zhang et al., 2025) affect the performance of Adam.

**Implicit Bias and Connection to $\ell_\infty$-Geometry.** A growing body of work examines Adam's implicit bias and its connection to $\ell_\infty$-geometry. This link is motivated by Adam's similarity to SignGD (Balles & Hennig, 2018; Bernstein et al., 2018), which performs normalized steepest descent under the $\ell_\infty$-norm. Kunstner et al. (2023) show that the performance gap between Adam and SGD increases with batch size, while SignGD achieves performance similar to Adam in the full-batch

regime, supporting this connection. Zhang et al. (2024a) prove that Adam without a stability constant converges to the $\ell_\infty$-max-margin solution in separable linear classification, later extended to multi-class classification by Fan et al. (2025). Tsilivis et al. (2025) investigate implicit bias of steepest descent in homogeneous neural networks, supporting that SignGD describes a typical dynamics of Adam. Complementing these results, Xie & Li (2024) show that AdamW implicitly solves an $\ell_\infty$-norm-constrained optimization problem, connecting its dynamics to the Frank-Wolfe algorithm. Exploiting this $\ell_\infty$-geometry is argued to be a key factor in Adam's advantage over SGD, particularly for language model training (Xie et al., 2025). Vasudeva et al. (2025) examine how Adam and GD show different implicit biases when training two-layer ReLU networks, describing Adam's richer and more diverse decision boundary.

## 7  DISCUSSION AND FUTURE WORK

We studied the convergence directions of Adam and Signum for logistic regression on linearly separable data in the mini-batch regime. Unlike full-batch Adam, which always converges to the $\ell_\infty$-max-margin solution, mini-batch Adam exhibits data-dependent behavior, revealing a richer implicit bias, while Signum consistently preserves the $\ell_\infty$-max-margin bias across all batch sizes.

**Toward understanding the Adam–SGD gap.** Empirical evidence shows that Adam's advantage over SGD is most pronounced in large-batch training, while the gap diminishes with smaller batches (Kunstner et al., 2023; Srećković et al., 2025; Marek et al., 2025). Our results suggest a possible explanation: the $\ell_\infty$-adaptivity of Adam, proposed as the source of its advantage (Xie et al., 2025), may vanish in the mini-batch regime. An important direction for future work is to investigate whether this loss of $\ell_\infty$-adaptivity extends beyond linear models and how it interacts with practical large-scale training.

**Limitations.** Our analysis for general dataset relies on the asymptotic regime $\beta_2 \to 1$ and on incremental Adam as a tractable surrogate. Extending the framework to incorporate general $\beta_2 < 1$, larger batch sizes, and common sampling schemes (e.g., random reshuffling) would make the theory more complete. See Appendix A for further discussion. Developing additional theoretical tools that can be applied under weaker assumptions also remains an important direction.

## ACKNOWLEDGMENTS

This work was supported by the National Research Foundation of Korea (NRF) grants funded by the Korean government (MSIT) (No. RS-2023-00211352; No. RS-2024-00421203) and the InnoCORE program of the Ministry of Science and ICT (No. N10250156).

## REPRODUCIBILITY STATEMENT

All assumptions and theorems for our theoretical results are stated in the main paper, with their complete proofs deferred to Appendices D to G. The primary experimental setups are described in the main paper and Appendix C, while details for supplementary experiments are provided in Appendices B and C.

## DECLARATION OF LLM USAGE

The authors utilized LLMs to improve the grammar and readability of this manuscript. The core conceptualization, analysis, and writing of the content were performed exclusively by the authors.

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

# Appendix

## A   FURTHER DISCUSSION

### A.1   EFFECT OF HYPERPARAMETERS ON MINI-BATCH ADAM

The scope of our analysis does not fully encompass the effects of batch sizes and momentum hyperparameters on the limit direction of mini-batch Adam. To motivate further investigation, this section presents preliminary empirical evidence that shows the sensitivity of the limit direction to these choices.

**Effect of Batch Size.**   To investigate the effect of batch size on the limiting behavior of mini-batch Adam, we run incremental Adam on the Gaussian data with $N = 10, d = 50$, varying batch sizes among 1, 2, 5, and 10. Figure 6 shows that as the batch size increases, the cosine similarity between the iterate and $\ell_\infty$-max-margin solution increases. This result suggests that the choice of batch size does affect the limiting behavior of mini-batch Adam, wherein larger batch sizes yield dynamics that converge towards those of the full-batch regime. A formal characterization of this dependency presents a compelling direction for future research.

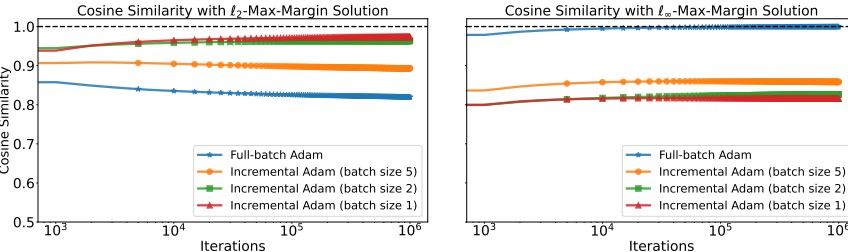

Figure 6: **The choice of batch size influences the limit direction of mini-batch Adam.**   We train on the same Gaussian data ($N = 10, d = 50$) as in Figure 1 and plot the cosine similarity of the weight vector with the $\ell_2$-max-margin solution (left) and the $\ell_\infty$-max-margin solution (right), varying batch sizes in $\{1, 2, 5, 10\}$. As the choice of batch size becomes closer to 10 (full-batch), the limit direction aligns closer to $\ell_\infty$-max-margin solution.

**Effect of Momentum Hyperparameters.**   Theorem 4.8 characterizes the limit direction of `AdamProxy`, which approximates mini-batch Adam with a batch size of one in the high-$\beta_2$ regime. We investigate how this approximation fails in the different choice of momentum hyperparameters. Revisiting the Gaussian data with $N = 10, d = 50$, we run mini-batch Adam with a batch size of 1 (including `Inc-Adam`) using LR schedule $\eta_t = \mathcal{O}(t^{-0.8})$, varying the momentum hyperparameters $(\beta_1, \beta_2) \in \{(0.1, 0.95), (0.5, 0.95), (0.9, 0.95), (0.1, 0.1), (0.1, 0.5), (0.1, 0.9)\}$.

The first experiment investigates the influence of $\beta_1$ by varying $\beta_1 \in \{0.1, 0.5, 0.9\}$ while maintaining a high choice of $\beta_2 = 0.95$. The results, presented in Figure 7, demonstrate that $\beta_1$ does not affect the convergence direction. This finding validates Proposition 4.1, which posits that our `AdamProxy` framework accurately models the high-$\beta_2$ regime, regardless of the choice of $\beta_1$.

Conversely, the choice of $\beta_2$ shows to be critical. We sweep $\beta_2 \in \{0.1, 0.5, 0.9\}$ while maintaining $\beta_1 = 0.1$ and plot the cosine similarities in Figure 8. The results illustrate that for choices of $\beta_2 \in \{0.1, 0.5\}$, the trajectory of mini-batch Adam deviates from the fixed-point solution of Theorem 4.8. It indicates that the high-$\beta_2$ condition is crucial for the approximation via `AdamProxy` and characterizing the limit direction of mini-batch Adam in the low-$\beta_2$ regime remains an important future direction.

### A.2   CAN WE DIRECTLY ANALYZE `INC-ADAM` FOR GENERAL $\beta_2$?

As empirically demonstrated in Appendix A.1, the selection of $\beta_2$ alters the limiting behavior of `Inc-Adam`. This observation motivates an inquiry into whether our fixed-point formulation can be directly generalized to accommodate general choices of $\beta_2$, based on a more *general* proxy algorithm. We proceed by outlining the technical challenges that prevent such a direct application of our framework, even under a stronger assumption on $\beta_1$ and the behavior of $\mathbf{w}_r$.

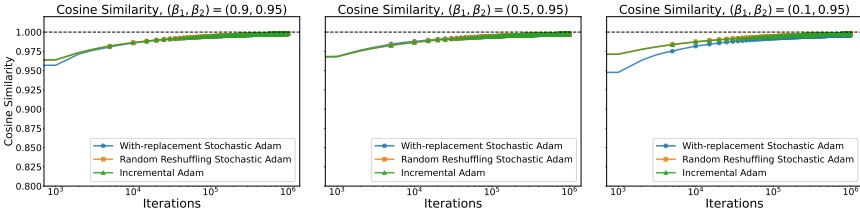

Figure 7: $\beta_1$ **does not affect the convergence direction of mini-batch Adam for large** $\beta_2$**.** We train on the same Gaussian data as in Figure 1, varying $\beta_1 \in \{0.9, 0.5, 0.1\}$ with fixed $\beta_2 = 0.95$, and plot the cosine similarity between the weight vector and the fixed-point solution (Algorithm 3). All mini-batch Adam variants with a batch size of 1 consistently converge to the fixed-point solution.

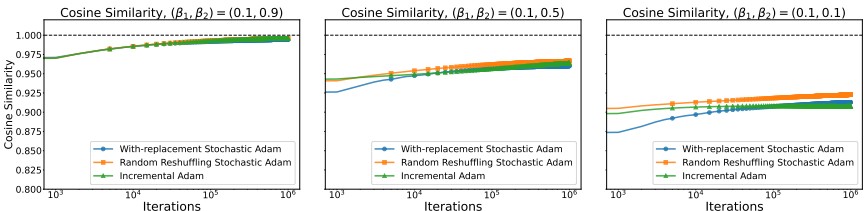

Figure 8: $\beta_2$ **affects the convergence direction of mini-batch Adam.** We train on the same Gaussian data as in Figure 1, varying $\beta_2 \in \{0.9, 0.5, 0.1\}$ with fixed $\beta_1 = 0.1$, and plot the cosine similarity between the weight vector and the fixed-point solution (Algorithm 3). Mini-batch Adam variants with a batch size of 1 deviate increasingly from the fixed-point solution as $\beta_2$ decreases.

Let $\{\mathbf{w}_t\}$ be the `Inc-Adam` iterates with $\beta_1 = 0$. For simplicity, we only consider the epoch-wise update and denote $\mathbf{w}_r = \mathbf{w}_r^0, \eta_r = C_{\text{inc}}(0, \beta_2)\eta_{rN}$ as an abuse of notation. By Proposition 2.5, $\mathbf{w}_r$ can be written by

$$\boldsymbol{\delta}_r \triangleq \underbrace{\sum_{i \in [N]} \frac{\nabla \mathcal{L}_i(\mathbf{w}_r)}{\sqrt{\sum_{j \in [N]} \beta_2^{(i,j)} \nabla \mathcal{L}_j(\mathbf{w}_r)^2}}}_{(\spadesuit)} + \boldsymbol{\epsilon}_r$$

$$\mathbf{w}_{r+1} - \mathbf{w}_r = -\eta_r \boldsymbol{\delta}_r$$

for some $\boldsymbol{\epsilon}_r \to \mathbf{0}$. Note that ($\spadesuit$) replaces `AdamProxy` in Section 4, incorporating the rich behavior induced by a general $\beta_2$. Then, we provide a preliminary characterization of the limit direction of `Inc-Adam` as follows.

**Lemma A.1.** *Suppose that (a) $\mathcal{L}(\mathbf{w}_r) \to 0$ and (b) $\mathbf{w}_r = \|\mathbf{w}_r\|_2 \hat{\mathbf{w}} + \boldsymbol{\rho}(r)$ for some $\hat{\mathbf{w}}$ with $\exists \lim_{r \to} \boldsymbol{\rho}(r)$. Then, under Assumptions 2.1 and 2.2, there exists $\mathbf{c} = (c_0, \cdots, c_{N-1}) \in \Delta^{N-1}$ such that the limit direction $\hat{\mathbf{w}}$ of* `Inc-Adam` *with $\beta_1 = 0$ satisfies*

$$\hat{\mathbf{w}} \propto \sum_{i \in [N]} \frac{c_i \mathbf{x}_i}{\sqrt{\sum_{j \in [N]} \beta_2^{(i,j)} c_j^2 \mathbf{x}_j^2}}, \tag{8}$$

*and $c_i = 0$ for $i \notin S$, where $S = \arg\min_{i \in [N]} \hat{\mathbf{w}}^\top \mathbf{x}_i$ is the index set of support vectors of $\hat{\mathbf{w}}$.*

We recall that the fixed-point formulation in Theorem 4.8 arises from constructing an optimization problem whose KKT conditions are given by Equation (5) fixing the $c_i$'s in the denominator; the convergence direction is then characterized when the dual solutions of the KKT conditions coincide with the $c_i$'s in the denominator. Therefore, to establish an analogous fixed-point type characterization, we should construct an optimization problem whose solution is given by $\mathbf{w}^* = \sum_{i \in [N]} \frac{d_i \mathbf{x}_i}{\sqrt{\sum_{j \in [N]} \beta_2^{(i,j)} c_j^2 \mathbf{x}_j^2}}$ with dual variables $d_i \geq 0$ satisfying that $d_j = 0$ for $j \in S = \arg\min_{i \in [N]} \mathbf{w}^{*\top} \mathbf{x}_i$.

However, this cannot be formulated via KKT conditions of an optimization problem. The index set $S$ indicates support vectors with respect to $\mathbf{x}_i$, while our dual variables are multiplied to

$$\frac{\mathbf{x}_i}{\sqrt{\sum_{j \in [N]} \beta_2^{(i,j)} c_j^2 \mathbf{x}_j^2}} = \tilde{\mathbf{x}}_i(\mathbf{c}).$$ A notable direction for future work is to generalize the proposed methodology for arbitrary values of $\beta_2$.

## B ADDITIONAL EXPERIMENTS

**Supplementary Experiments in Section 3.** To investigate the universality of Theorem 3.3 with respect to the choice of the momentum hyperparameters, we run mini-batch Adam (with batch size 1) on SR dataset $\mathbf{x}_0 = (1, 1, 1, 1)$, $\mathbf{x}_1 = (2, 2, 2, -2)$, $\mathbf{x}_2 = (3, 3, -3, -3)$, and $\mathbf{x}_3 = (4, -4, 4, -4)$, varying the momentum hyperparameters $(\beta_1, \beta_2) \in \{(0.1, 0.1), (0.5, 0.5), (0.9, 0.95)\}$. Figure 9 demonstrates that its limiting behavior toward $\ell_2$-max-margin solution consistently holds on the broad choices of $(\beta_1, \beta_2)$.

**Supplementary Experiments in Section 5.** Theorem 5.1 demonstrates that `Inc-Signum` maintains its bias to $\ell_\infty$-max-margin solution, while the momentum hyperparameter $\beta$ should be close enough to 1 depending on the choice of batch size; the gap between $\beta$ and 1 should decrease as batch size $b$ decreases. To investigate this dependency, we run `Inc-Signum` on the same Gaussian data as in Figure 1, varying batch size $b \in \{1, 2, 5, 10\}$ and the momentum hyperparameter $\beta \in \{0.5, 0.9, 0.95, 0.99\}$. Figure 10 shows that to maintain the $\ell_\infty$-bias, the choice of $\beta$ should be closer to 1 as the batch size decreases.

## C EXPERIMENTAL DETAILS

This section provides details for the experiments presented in the main text and appendix.

We generate synthetic separable data as follows:

- **Gaussian data (Figures 1, 3, 5, 6 to 8 and 10):** Samples are drawn from the standard Gaussian distribution $\mathcal{N}(0, I)$. We set the dimension $d = 50$ and sample $N = 10$ points, ensuring a positive margin so that the data is linearly separable.
- **Scaled Rademacher (SR) data (Figures 2 and 9):** We use $\mathbf{x}_0 = (1, 1, 1, 1)$, $\mathbf{x}_1 = (2, 2, 2, -2)$, $\mathbf{x}_2 = (3, 3, -3, -3)$, and $\mathbf{x}_3 = (4, -4, 4, -4)$.
- **Shifted-diagonal data (Figure 4):** We use $\mathbf{x}_0 = (1, \delta, \delta, \delta)$, $\mathbf{x}_1 = (\delta, 2, \delta, \delta)$, $\mathbf{x}_2 = (\delta, \delta, 4, \delta)$, and $\mathbf{x}_3 = (\delta, \delta, \delta, 8)$ with $\delta = 0.1$.

We minimize the exponential loss using various algorithms. Momentum hyperparameters are $(\beta_1, \beta_2) = (0.9, 0.95)$ for Adam and $\beta = 0.99$ for Signum unless specified otherwise. For Adam and Signum variants, we use a learning rate schedule $\eta_t = \eta_0(t + 2)^{-a}$ with $\eta_0 = 0.1$ and $a = 0.8$, following our theoretical analysis. Gradient descent uses a fixed learning rate $\eta_t = \eta_0 = 0.1$. Margins with respect to different norms are computed using CVXPY (Diamond & Boyd, 2016).

The fixed-point solution (Theorem 4.8) is obtained via fixed-point iteration (Algorithm 3) for Figures 3, 7 and 8. We initialize $\mathbf{c}_0 = (1/N, \dots, 1/N) \in \Delta^{N-1}$, set the threshold $\epsilon_{\text{thr}} = 10^{-8}$, and converge to the fixed-point solution within 20 iterations in all settings.

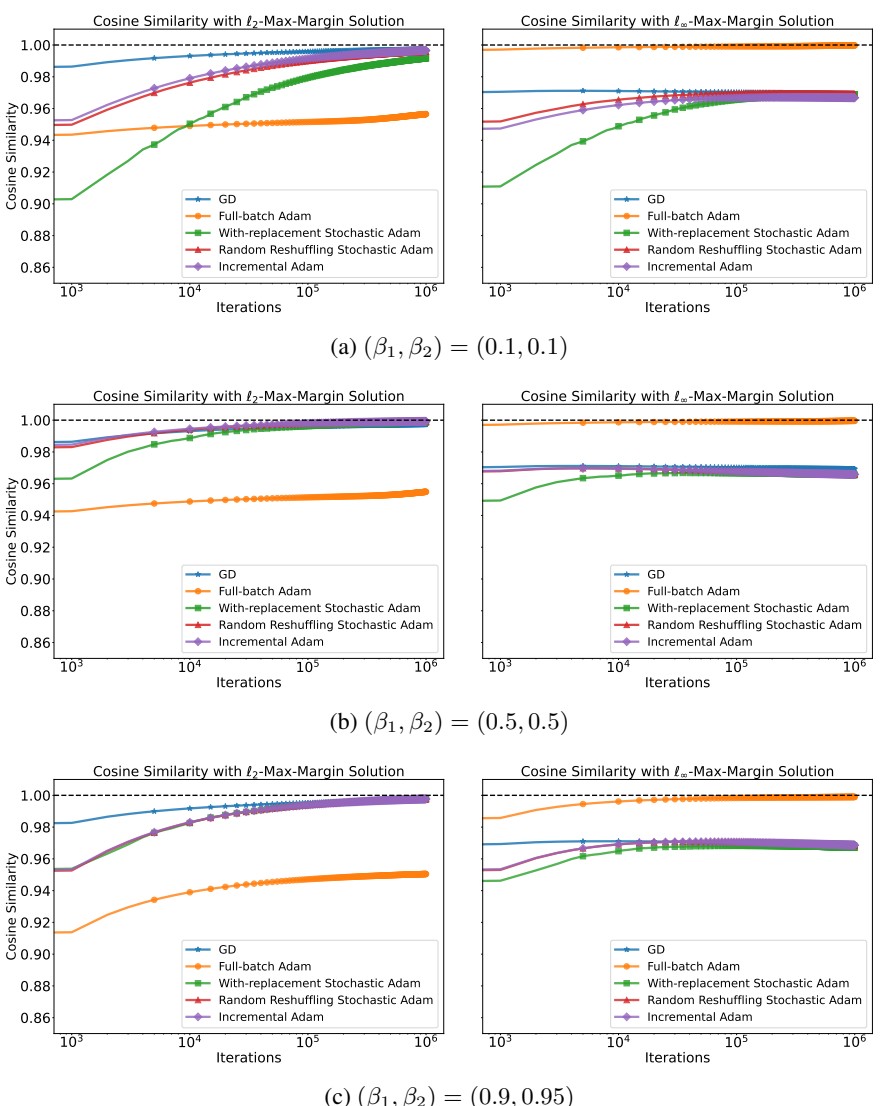

(a) $(\beta_1, \beta_2) = (0.1, 0.1)$

(b) $(\beta_1, \beta_2) = (0.5, 0.5)$

(c) $(\beta_1, \beta_2) = (0.9, 0.95)$

Figure 9: **Mini-batch Adam converges to the max $\ell_2$-margin solution for SR data.** We train on SR dataset $\mathbf{x}_0 = (1, 1, 1, 1)$, $\mathbf{x}_1 = (2, 2, 2, -2)$, $\mathbf{x}_2 = (3, 3, -3, -3)$, and $\mathbf{x}_3 = (4, -4, 4, -4)$, varying the momentum hyperparameters. In all tested configurations, the family of mini-batch Adam algorithms with a batch size of $1$ converge to the $\ell_2$ max-margin solution, which deviate significantly from the $\ell_\infty$ bias of full-batch Adam.

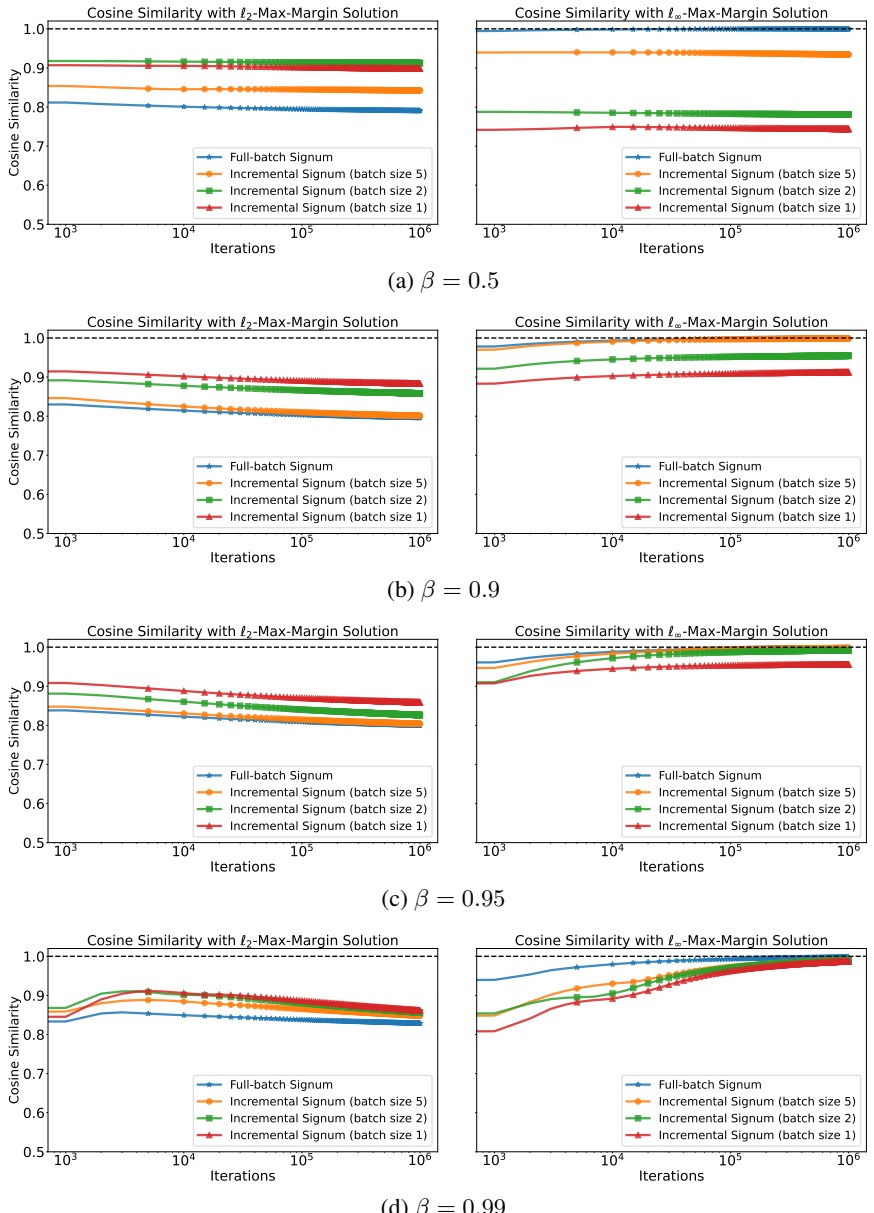

Figure 10: **Effect of Batch Size on `Inc-Signum`.** We run `Inc-Signum` on the same Gaussian data ($N = 10, d = 50$) as in Figure 1 and plot the cosine similarity of the weight vector with the $\ell_2$-max-margin solution (left) and the $\ell_\infty$-max-margin solution (right), varying batch size $b \in \{1, 2, 5, 10\}$ and the momentum hyperparameter $\beta \in \{0.5, 0.9, 0.95, 0.99\}$. As the batch size decreases, we should choose $\beta$ closer to 1 to maintain the limit direction toward $\ell_\infty$-max-margin solution.

# D MISSING PROOFS IN SECTION 2

In this section, we provide the omitted proofs in Section 2, which describes asymptotic behaviors of `Det-Adam` and `Inc-Adam`. We first introduce Lemma D.1 originated from Zou et al. (2023, Lemma A.2), which gives a coordinate-wise upper bound of updates of both `Det-Adam` and `Inc-Adam`. Then, we prove Propositions 2.4 and 2.5 by approximating two momentum terms.

**Notation.** In this section, we introduce the proxy function $\mathcal{G} : \mathbb{R}^d \to \mathbb{R}$ defined as

$$\mathcal{G}(\mathbf{w}) := -\frac{1}{N} \sum_{i \in [N]} \ell'(\mathbf{w}^\top \mathbf{x}_i).$$

**Lemma D.1** (Lemma A.2 in Zou et al. (2023)). *Assume $\beta_1^2 \leq \beta_2$ and let $\alpha = \sqrt{\frac{\beta_2(1-\beta_1)^2}{(1-\beta_2)(\beta_2-\beta_1^2)}}$. Then, for both `Det-Adam` and `Inc-Adam` iterates, $\mathbf{m}_t[k] \leq \alpha\sqrt{\mathbf{v}_t[k]}$ for all $k \in [d]$.*

*Proof.* Following the proof of Zou et al. (2023, Lemma A.2), we can easily show that the given upper bound holds for both `Det-Adam` and `Inc-Adam`. We prove the case of `Inc-Adam`, while it naturally extends to `Det-Adam`. By Cauchy-Schwartz inequality, we get

$$|\mathbf{m}_t[k]| = |\sum_{\tau=0}^{t} \beta_1^\tau(1-\beta_1)\nabla\mathcal{L}_{i_{t-\tau}}(\mathbf{w}_{t-\tau})[k]|$$

$$\leq \sum_{\tau=0}^{t} \beta_1^\tau(1-\beta_1)|\nabla\mathcal{L}_{i_{t-\tau}}(\mathbf{w}_{t-\tau})[k]|$$

$$\leq \left(\sum_{\tau=0}^{t} \beta_2^\tau(1-\beta_2)|\nabla\mathcal{L}_{i_{t-\tau}}(\mathbf{w}_{t-\tau})[k]|^2\right)^{1/2} \left(\sum_{\tau=0}^{t} \frac{\beta_1^{2\tau}(1-\beta_1)^2}{\beta_2^\tau(1-\beta_2)}\right)^{1/2} \quad \text{(CS inequality)}$$

$$\leq \alpha\sqrt{\mathbf{v}_t[k]}.$$

The last inequality is from

$$\sum_{\tau=0}^{t} \frac{\beta_1^{2\tau}(1-\beta_1)^2}{\beta_2^\tau(1-\beta_2)} \leq \frac{(1-\beta_1)^2}{1-\beta_2} \sum_{\tau=0}^{\infty} \left(\frac{\beta_1^2}{\beta_2}\right)^\tau = \frac{\beta_2(1-\beta_1)^2}{(1-\beta_2)(\beta_2-\beta_1^2)} = \alpha^2,$$

where the infinite sum is bounded from $\beta_1^2 \leq \beta_2$. $\square$

## D.1 PROOF OF PROPOSITION 2.4

**Proposition 2.4.** *Let $\{\mathbf{w}_t\}_{t=0}^{\infty}$ be the iterates of `Det-Adam` with $\beta_1 \leq \beta_2$. Then, under Assumptions 2.2 and 2.3, if $\lim_{t\to\infty} \frac{\eta_t^{1/2}\mathcal{L}(\mathbf{w}_t)}{|\nabla\mathcal{L}(\mathbf{w}_t)[k]|} = 0$, then the update of $k$-th coordinate $\mathbf{w}_{t+1}[k] - \mathbf{w}_t[k]$ can be represented by*

$$\mathbf{w}_{t+1}[k] - \mathbf{w}_t[k] = -\eta_t\left(\text{sign}(\nabla\mathcal{L}(\mathbf{w}_t)[k]) + \epsilon_t\right), \tag{1}$$

*for some $\lim_{t\to\infty} \epsilon_t = 0$.*

*Proof.* We recall Lemma 6.1 in Zhang et al. (2024a), stating that

$$\left|\mathbf{m}_t[k] - (1-\beta_1^{t+1})\nabla\mathcal{L}(\mathbf{w}_t)[k]\right| \leq c_m\eta_t\mathcal{G}(\mathbf{w}_t),$$

$$\left|\sqrt{\mathbf{v}_t[k]} - \sqrt{1-\beta_2^{t+1}}|\nabla\mathcal{L}(\mathbf{w}_t)[k]|\right| \leq c_v\sqrt{\eta_t}\mathcal{G}(\mathbf{w}_t)$$

for all $t > t_1$ and $k \in [d]$. Based on these results, we can rewrite $\mathbf{m}_r^s[k]$ and $\sqrt{\mathbf{v}_r^s[k]}$ as

$$\mathbf{m}_t[k] = (1-\beta_1^{t+1})\nabla\mathcal{L}(\mathbf{w}_t)[k] + \epsilon_{\mathbf{m}}(t)\mathcal{G}(\mathbf{w}_t),$$

$$\sqrt{\mathbf{v}_t[k]} = \sqrt{1-\beta_2^{t+1}}|\nabla\mathcal{L}(\mathbf{w}_t)[k]| + \epsilon_{\mathbf{v}}(t)\mathcal{G}(\mathbf{w}_t),$$

where $\epsilon_{\mathbf{m}}(t) = \mathcal{O}(\eta_t), \epsilon_{\mathbf{v}}(t) = \mathcal{O}(\sqrt{\eta_t})$. Note that $\frac{\mathcal{G}(\mathbf{w}_t)}{\mathcal{L}(\mathbf{w}_t)} \leq 1$ from Lemma I.1 and $\left|\frac{a+\epsilon_1}{b+\epsilon_2} - \frac{a}{b}\right| \leq \left|\frac{\epsilon_1}{b+\epsilon_2}\right| + \left|\frac{a}{b} \cdot \frac{\epsilon_2}{b+\epsilon_2}\right| \leq \left|\frac{\epsilon_1}{b}\right| + \left|\frac{a}{b} \cdot \frac{\epsilon_2}{b}\right|$ for positive numbers $\epsilon_1, \epsilon_2, b$. Therefore, if $\lim_{t\to\infty} \frac{\eta_t^{1/2}\mathcal{L}(\mathbf{w}_t)}{|\nabla\mathcal{L}(\mathbf{w}_t)[k]|} = 0$, then we get

$$\left| \frac{\mathbf{m}_t[k]}{\sqrt{\mathbf{v}_t[k]}} - \frac{1-\beta_1^{t+1}}{\sqrt{1-\beta_2^{t+1}}} \operatorname{sign}\left(\nabla\mathcal{L}(\mathbf{w}_t)[k]\right) \right|$$

$$\leq \underbrace{\left| \frac{\epsilon_{\mathbf{m}}(t)\mathcal{G}(\mathbf{w}_t)}{\sqrt{1-\beta_2^{t+1}}\, |\nabla\mathcal{L}(\mathbf{w}_t)[k]|} \right|}_{\to 0} + \left| \underbrace{\frac{1-\beta_1^{t+1}}{\sqrt{1-\beta_2^{t+1}}} \operatorname{sign}\left(\nabla\mathcal{L}(\mathbf{w}_t)[k]\right)}_{\text{bounded}} \cdot \underbrace{\frac{\epsilon_{\mathbf{v}}(t)\mathcal{G}(\mathbf{w}_t)}{\sqrt{1-\beta_2^{t+1}}\, |\nabla\mathcal{L}(\mathbf{w}_t)[k]|}}_{\to 0} \right|$$

$\to 0$.

From $\beta_1^t, \beta_2^t \to 0$, we get $\mathbf{w}_{t+1}[k] - \mathbf{w}_t[k] = -\eta_t \frac{\mathbf{m}_t[k]}{\sqrt{\mathbf{v}_t[k]}} = \eta_t \left(\operatorname{sign}\left(\nabla\mathcal{L}(\mathbf{w}_t)[k]\right) + \epsilon_t\right)$ for some $\lim_{t\to\infty} \epsilon_t = 0$. $\qquad\square$

## D.2 Proof of Proposition 2.5

To prove Proposition 2.5, we start by characterizing the first and second momentum terms $\mathbf{m}_t, \mathbf{v}_t$ in `Inc-Adam`, which track the exponential moving averages of the historical mini-batch gradients and square gradients. As mentioned before, a key technical challenge of analyzing Adam is its dependency in the full gradient history. The following lemma approximates momentum terms with respect to a function of the *first* iterate in each epoch $\mathbf{w}_r^0$, which is crucial for our *epoch-wise* analysis.

**Lemma D.2.** *Under Assumptions 2.2 and 2.3, there exists $t_1$ only depending on $\beta_1, \beta_2$ and the dataset, such that*

$$\left| \mathbf{m}_r^s[k] - \frac{1-\beta_1}{1-\beta_1^N} \sum_{j\in[N]} \beta_1^{(s,j)} \nabla\mathcal{L}_j(\mathbf{w}_r^0)[k] \right| \leq \epsilon_{\mathbf{m}}(t) \max_{j\in[N]} \left| \nabla\mathcal{L}_j(\mathbf{w}_r^0)[k] \right|,$$

$$\left| \mathbf{v}_r^s[k] - \frac{1-\beta_2}{1-\beta_2^N} \sum_{j\in[N]} \beta_2^{(s,j)} \nabla\mathcal{L}_j(\mathbf{w}_r^0)[k]^2 \right| \leq \epsilon_{\mathbf{v}}(t) \max_{j\in[N]} \left| \nabla\mathcal{L}_j(\mathbf{w}_r^0)[k] \right|^2,$$

*for all $r, s$ satisfying $rN + s > t_1$ and $k \in [d]$, where*

$$\epsilon_{\mathbf{m}}(t) \triangleq (1-\beta_1)e^{\alpha ND\eta_{rN}}c_2\eta_t + (e^{\alpha ND\eta_{rN}} - 1) + \beta_1^{t+1},$$
$$\epsilon_{\mathbf{v}}(t) \triangleq 3(1-\beta_2)e^{2\alpha ND\eta_{rN}}c_2'\eta_t + 3(e^{2\alpha ND\eta_{rN}} - 1) + \beta_2^{t+1},$$

$D = \max_{j\in[N]} \|\mathbf{x}_j\|_1$, *and $c_2, c_2'$ are constants only depend on $\beta_1, \beta_2$, and the dataset.*

*Proof.* Consider $t = rN + s$ and the gradient at time $t$ is sampled from data with index $s$ in $r$-th epoch. Then we can decompose the error between $\mathbf{m}_r^s[k]$ and $\frac{1-\beta_1}{1-\beta_1^N} \sum_{j \in [N]} \beta_1^{(s,j)} \nabla \mathcal{L}_j(\mathbf{w}_r^0)[k]$ as

$$|\mathbf{m}_r^s[k] - \frac{1-\beta_1}{1-\beta_1^N} \sum_{j \in [N]} \beta_1^{(s,j)} \nabla \mathcal{L}_j(\mathbf{w}_r^0)[k]|$$

$$= |\sum_{\tau=0}^t \beta_1^\tau (1-\beta_1) \nabla \mathcal{L}_{i_{t-\tau}}(\mathbf{w}_{t-\tau})[k] - \frac{1-\beta_1}{1-\beta_1^N} \sum_{j \in [N]} \beta_1^{(s,j)} \nabla \mathcal{L}_j(\mathbf{w}_r^0)[k]|$$

$$\leq \underbrace{|\sum_{\tau=0}^t \beta_1^\tau (1-\beta_1) \nabla \mathcal{L}_{i_{t-\tau}}(\mathbf{w}_{t-\tau})[k] - \sum_{\tau=0}^t \beta_1^\tau (1-\beta_1) \nabla \mathcal{L}_{i_{t-\tau}}(\mathbf{w}_t)[k]|}_{(A): \text{ error from movement of weights}}$$

$$+ \underbrace{|\sum_{\tau=0}^t \beta_1^\tau (1-\beta_1) \nabla \mathcal{L}_{i_{t-\tau}}(\mathbf{w}_t)[k] - \sum_{\tau=0}^t \beta_1^\tau (1-\beta_1) \nabla \mathcal{L}_{i_{t-\tau}}(\mathbf{w}_r^0)[k]|}_{(B): \text{ error between } \mathbf{w}_t \text{ and } \mathbf{w}_r^0}$$

$$+ \underbrace{|\sum_{\tau=0}^t \beta_1^\tau (1-\beta_1) \nabla \mathcal{L}_{i_{t-\tau}}(\mathbf{w}_r^0)[k] - \frac{1-\beta_1}{1-\beta_1^N} \sum_{j \in [N]} \beta_1^{(s,j)} \nabla \mathcal{L}_j(\mathbf{w}_r^0)[k]|}_{(C): \text{ error from infinite-sum approximation}}.$$

Note that

$$(A) \leq \sum_{\tau=0}^t \beta_1^\tau (1-\beta_1) |\ell'(\mathbf{w}_{t-\tau}^\top \mathbf{x}_{i_{t-\tau}}) - \ell'(\mathbf{w}_t^\top \mathbf{x}_{i_{t-\tau}})| |\mathbf{x}_{i_{t-\tau}}[k]|$$

$$= \sum_{\tau=0}^t \beta_1^\tau (1-\beta_1) \left| \frac{\ell'(\mathbf{w}_{t-\tau}^\top \mathbf{x}_{i_{t-\tau}})}{\ell'(\mathbf{w}_t^\top \mathbf{x}_{i_{t-\tau}})} - 1 \right| |\ell'(\mathbf{w}_t^\top \mathbf{x}_{i_{t-\tau}})| |\mathbf{x}_{i_{t-\tau}}[k]|$$

$$\overset{(*)}{\leq} (1-\beta_1) \max_{j \in [N]} |\nabla \mathcal{L}_j(\mathbf{w}_t)[k]| \sum_{\tau=0}^t \beta_1^\tau (e^{\alpha D \sum_{\tau'=1}^\tau \eta_{t-\tau'}} - 1)|$$

$$\overset{(**)}{\leq} (1-\beta_1) c_2 \eta_t \max_{j \in [N]} |\nabla \mathcal{L}_j(\mathbf{w}_t)[k]|,$$

$$\overset{(***)}{\leq} (1-\beta_1) e^{\alpha N D \eta_{rN}} c_2 \eta_t \max_{j \in [N]} |\nabla \mathcal{L}_j(\mathbf{w}_r^0)[k]|$$

for some $c_2 > 0$ and $t > t_1$. Here, $(*)$ is from Lemma I.3 and

$$e^{|(\mathbf{w}_t - \mathbf{w}_{t-\tau})^\top \mathbf{x}_{i_{t-\tau}}|} - 1 \leq e^{\|\mathbf{w}_t - \mathbf{w}_{t-\tau}\|_\infty \|\mathbf{x}_{i_{t-\tau}}\|_1} - 1 \leq e^{\alpha D \sum_{\tau'=1}^\tau \eta_{t-\tau'}} - 1.$$

Also, $(**)$ is from Assumption 2.3, and $(***)$ is from

$$\max_{j \in [N]} |\nabla \mathcal{L}_j(\mathbf{w}_t)[k]| \leq \max_{j \in [N]} |\nabla \mathcal{L}_j(\mathbf{w}_r^0)[k]| \cdot \max_{j \in [N]} \left| \frac{\nabla \mathcal{L}_j(\mathbf{w}_t)[k]}{\nabla \mathcal{L}_j(\mathbf{w}_r^0)[k]} \right|$$

$$= \max_{j \in [N]} |\nabla \mathcal{L}_j(\mathbf{w}_r^0)[k]| \cdot \max_{j \in [N]} \left| \frac{\ell'(\mathbf{w}_t^\top \mathbf{x}_j)}{\ell'(\mathbf{w}_r^{0\top} \mathbf{x}_j)} \right|$$

$$\leq e^{\alpha N D \eta_{rN}} \max_{j \in [N]} |\nabla \mathcal{L}_j(\mathbf{w}_r^0)[k]|,$$

where the last inequality is from Lemma I.3 and

$$\max_{j \in [N]} \left| \frac{\ell'(\mathbf{w}_t^\top \mathbf{x}_j)}{\ell'(\mathbf{w}_r^{0\top} \mathbf{x}_j)} \right| \leq \max_{j \in [N]} e^{|(\mathbf{w}_t - \mathbf{w}_r^0)^\top \mathbf{x}_j|} \leq e^{\alpha N D \eta_{rN}}.$$

Also, observe that

$$(B) \le \sum_{\tau=0}^{t} \beta_1^\tau (1-\beta_1)|\ell'(\mathbf{w}_t^\top \mathbf{x}_{i_{t-\tau}}) - \ell'(\mathbf{w}_r^{0\top} \mathbf{x}_{i_{t-\tau}})||\mathbf{x}_{i_{t-\tau}}[k]|$$

$$= \sum_{\tau=0}^{t} \beta_1^\tau (1-\beta_1) \left| \frac{\ell'(\mathbf{w}_t^\top \mathbf{x}_{i_{t-\tau}})}{\ell'(\mathbf{w}_r^{0\top} \mathbf{x}_{i_{t-\tau}})} - 1 \right| |\ell'(\mathbf{w}_r^{0\top} \mathbf{x}_{i_{t-\tau}})||\mathbf{x}_{i_{t-\tau}}[k]|$$

$$\overset{(*)}{\le} (1-\beta_1) \max_{j \in [N]} |\nabla \mathcal{L}_j(\mathbf{w}_r^0)[k]|(e^{\alpha N D \eta_r N} - 1) \sum_{\tau=0}^{t} \beta_1^\tau$$

$$\overset{(**)}{\le} (e^{\alpha N D \eta_r N} - 1) \max_{j \in [N]} |\nabla \mathcal{L}_j(\mathbf{w}_r^0)[k]|,$$

where $(*)$ is from Lemma I.3 and

$$\left| \frac{\ell'(\mathbf{w}_t^\top \mathbf{x}_{i_{t-\tau}})}{\ell'(\mathbf{w}_r^{0\top} \mathbf{x}_{i_{t-\tau}})} - 1 \right| \le e^{|(\mathbf{w}_t - \mathbf{w}_r^0)^\top \mathbf{x}_{i_{t-\tau}}|} - 1 \le e^{\|\mathbf{w}_t - \mathbf{w}_r^0\|_\infty \|\mathbf{x}_{i_{t-\tau}}\|_1} \le e^{\alpha N D \eta_r N} - 1,$$

and $(**)$ is from $\sum_{\tau=0}^{t} \beta_1^\tau \le \frac{1}{1-\beta_1}$.

Furthermore,

$$(C) = \left| \sum_{\tau=0}^{t} \beta_1^\tau (1-\beta_1) \nabla \mathcal{L}_{i_{t-\tau}}(\mathbf{w}_r^0)[k] - \sum_{\tau=0}^{\infty} \beta_1^\tau (1-\beta_1) \nabla \mathcal{L}_{i_{t-\tau}}(\mathbf{w}_r^0)[k] \right|$$

$$\le \sum_{\tau=t+1}^{\infty} \beta_1^\tau (1-\beta_1) \left| \nabla \mathcal{L}_{i_{t-\tau}}(\mathbf{w}_r^0)[k] \right|$$

$$\le \beta_1^{t+1} \max_{j \in [N]} |\nabla \mathcal{L}_j(\mathbf{w}_r^0)[k]|.$$

Therefore, we can conclude that

$$|\mathbf{m}_r^s[k] - \frac{1-\beta_1}{1-\beta_1^N} \sum_{j \in [N]} \beta_1^{(s,j)} \nabla \mathcal{L}_j(\mathbf{w}_r^0)[k]|$$

$$\le \underbrace{\left( (1-\beta_1) e^{\alpha N D \eta_r N} c_2 \eta_t + (e^{\alpha N D \eta_r N} - 1) + \beta_1^{t+1} \right)}_{\triangleq \epsilon_{\mathbf{m}}(t)} \max_{j \in [N]} |\nabla \mathcal{L}_j(\mathbf{w}_r^0)[k]|.$$

Similarly,

$$|\mathbf{v}_r^s[k] - \frac{1-\beta_2}{1-\beta_2^N} \sum_{j \in [N]} \beta_2^{(s,j)} \nabla \mathcal{L}_j(\mathbf{w}_r^0)[k]^2|$$

$$= |\sum_{\tau=0}^{t} \beta_2^\tau (1-\beta_2) \nabla \mathcal{L}_{i_{t-\tau}}(\mathbf{w}_{t-\tau})[k]^2 - \frac{1-\beta_2}{1-\beta_2^N} \sum_{j \in [N]} \beta_2^{(s,j)} \nabla \mathcal{L}_j(\mathbf{w}_r^0)[k]^2|$$

$$\le \underbrace{|\sum_{\tau=0}^{t} \beta_2^\tau (1-\beta_2) \nabla \mathcal{L}_{i_{t-\tau}}(\mathbf{w}_{t-\tau})[k]^2 - \sum_{\tau=0}^{t} \beta_2^\tau (1-\beta_2) \nabla \mathcal{L}_{i_{t-\tau}}(\mathbf{w}_t)[k]^2|}_{(D): \text{ error from movement of weights}}$$

$$+ \underbrace{|\sum_{\tau=0}^{t} \beta_2^\tau (1-\beta_2) \nabla \mathcal{L}_{i_{t-\tau}}(\mathbf{w}_t)[k]^2 - \sum_{\tau=0}^{t} \beta_2^\tau (1-\beta_2) \nabla \mathcal{L}_{i_{t-\tau}}(\mathbf{w}_r^0)[k]^2|}_{(E): \text{ error between } \mathbf{w}_t \text{ and } \mathbf{w}_r^0}$$

$$+ \underbrace{|\sum_{\tau=0}^{t} \beta_2^\tau (1-\beta_2) \nabla \mathcal{L}_{i_{t-\tau}}(\mathbf{w}_r^0)[k]^2 - \frac{1-\beta_2}{1-\beta_2^N} \sum_{j \in [N]} \beta_2^{(s,j)} \nabla \mathcal{L}_j(\mathbf{w}_r^0)[k]^2|}_{(F): \text{ error from infinite-sum approximation}}.$$

Observe that

$$
\begin{aligned}
(D) &\leq \sum_{\tau=0}^{t} \beta_2^\tau (1-\beta_2) |\ell'(\mathbf{w}_{t-\tau}^\top \mathbf{x}_{i_{t-\tau}})^2 - \ell'(\mathbf{w}_t^\top \mathbf{x}_{i_{t-\tau}})^2| \|\mathbf{x}_{i_{t-\tau}}[k]\|^2 \\
&= \sum_{\tau=0}^{t} \beta_2^\tau (1-\beta_2) \left| \left(\frac{\ell'(\mathbf{w}_{t-\tau}^\top \mathbf{x}_{i_{t-\tau}})}{\ell'(\mathbf{w}_t^\top \mathbf{x}_{i_{t-\tau}})}\right)^2 - 1 \right| |\ell'(\mathbf{w}_t^\top \mathbf{x}_{i_{t-\tau}})|^2 |\mathbf{x}_{i_{t-\tau}}[k]|^2 \\
&\overset{(*)}{\leq} 3(1-\beta_2) \max_{j\in[N]} |\nabla \mathcal{L}_j(\mathbf{w}_t)[k]|^2 \sum_{\tau=0}^{t} \beta_2^\tau (e^{2\alpha D \sum_{\tau'=1}^{\tau} \eta_{t-\tau'}} - 1)| \\
&\overset{(**)}{\leq} 3(1-\beta_2) c_2' \eta_t \max_{j\in[N]} |\nabla \mathcal{L}_j(\mathbf{w}_t)[k]|^2, \\
&\overset{(***)}{\leq} 3(1-\beta_2) e^{2\alpha N D \eta_{rN}} c_2' \eta_t \max_{j\in[N]} |\nabla \mathcal{L}_j(\mathbf{w}_r^0)[k]|^2
\end{aligned}
$$

for some $c_2' > 0$ and $t > t_1'$. Here, $(*)$ is from Lemma I.4 and

$$
\left| \left(\frac{\ell'(\mathbf{w}_{t-\tau}^\top \mathbf{x}_{i_{t-\tau}})}{\ell'(\mathbf{w}_t^\top \mathbf{x}_{i_{t-\tau}})}\right)^2 - 1 \right| \leq 3(e^{2|(\mathbf{w}_t - \mathbf{w}_r^0)^\top \mathbf{x}_{i_{t-\tau}}|} - 1) \leq 3(e^{2\alpha D \sum_{\tau'=1}^{\tau} \eta_{t-\tau'}} - 1),
$$

$(**)$ is from Assumption 2.3, and $(***)$ can be derived similarly. Also, we get

$$
\begin{aligned}
(E) &\leq \sum_{\tau=0}^{t} \beta_2^\tau (1-\beta_2) |\ell'(\mathbf{w}_t^\top \mathbf{x}_{i_{t-\tau}})^2 - \ell'(\mathbf{w}_r^{0\top} \mathbf{x}_{i_{t-\tau}})^2| \|\mathbf{x}_{i_{t-\tau}}[k]\|^2 \\
&\leq 3(e^{2\alpha N D \eta_{rN}} - 1) \max_{j\in[N]} |\nabla \mathcal{L}_j(\mathbf{w}_r^0)[k]|^2, \\
(F) &= \left| \sum_{\tau=0}^{t} \beta_2^\tau (1-\beta_2) \nabla \mathcal{L}_{i_{t-\tau}}(\mathbf{w}_r^0)[k]^2 - \sum_{\tau=0}^{\infty} \beta_2^\tau (1-\beta_2) \nabla \mathcal{L}_{i_{t-\tau}}(\mathbf{w}_r^0)[k]^2 \right| \\
&\leq \sum_{\tau=t+1}^{\infty} \beta_2^\tau (1-\beta_2) \left| \nabla \mathcal{L}_{i_{t-\tau}}(\mathbf{w}_r^0)[k] \right|^2 \\
&\leq \beta_2^{t+1} \max_{j\in[N]} |\nabla \mathcal{L}_j(\mathbf{w}_r^0)[k]|^2,
\end{aligned}
$$

which can also be derived similarly to the previous part. Therefore, we can conclude that

$$
\begin{aligned}
&|\mathbf{v}_r^s[k] - \frac{1-\beta_2}{1-\beta_2^N} \sum_{j\in[N]} \beta_2^{(s,j)} \nabla \mathcal{L}_j(\mathbf{w}_r^0)[k]^2| \\
&\leq \underbrace{\left(3(1-\beta_2) e^{2\alpha N D \eta_{rN}} c_2' \eta_t + 3(e^{2\alpha N D \eta_{rN}} - 1) + \beta_2^{t+1}\right)}_{\triangleq \epsilon_\mathbf{v}(t)} \max_{j\in[N]} |\nabla \mathcal{L}_j(\mathbf{w}_r^0)[k]|^2.
\end{aligned}
$$

$\square$

Notice that $\epsilon_\mathbf{m}(t)$ and $\epsilon_\mathbf{v}(t)$ defined in Lemma D.2 converge to 0 as $t \to \infty$, implying that each coordinate of two momentum terms can be effectively approximated by a weighted sum of mini-batch gradients and gradient squares, which emphasizes the discrepancy with `Det-Adam` and `Inc-Adam`. We also mention that the bound depends on $\max_{j\in[N]} |\nabla \mathcal{L}_j(\mathbf{w}_r^0)[k]|$, which converges to 0 as $\mathcal{L}(\mathbf{w}_r^0) \to 0$. Such approaches provide tight bounds, which enables the asymptotic analysis of `Inc-Adam`.

**Proposition 2.5.** *Let* $\{\mathbf{w}_t\}_{t=0}^{\infty}$ *be the iterates of* `Inc-Adam` *with* $\beta_1 \leq \beta_2$. *Then, under Assumptions 2.2 and 2.3, the epoch-wise update* $\mathbf{w}_{r+1}^0 - \mathbf{w}_r^0$ *can be represented by*

$$
\mathbf{w}_{r+1}^0 - \mathbf{w}_r^0 = -\eta_{rN} \left( C_{inc}(\beta_1, \beta_2) \sum_{i\in[N]} \frac{\sum_{j\in[N]} \beta_1^{(i,j)} \nabla \mathcal{L}_j(\mathbf{w}_r^0)}{\sqrt{\sum_{j\in[N]} \beta_2^{(i,j)} \nabla \mathcal{L}_j(\mathbf{w}_r^0)^2}} + \boldsymbol{\epsilon}_r \right), \tag{2}
$$

where $\beta_1^{(i,j)} = \beta_1^{(i-j) \bmod N}$, $\beta_2^{(i,j)} = \beta_2^{(i-j) \bmod N}$, $C_{inc}(\beta_1, \beta_2) = \frac{1-\beta_1}{1-\beta_1^N}\sqrt{\frac{1-\beta_2^N}{1-\beta_2}}$ is a function of $\beta_1, \beta_2$, and $\lim_{r\to\infty} \boldsymbol{\epsilon}_r = \mathbf{0}$. If $\eta_t = (t+2)^{-a}$ for some $a \in (0,1]$, then $\|\boldsymbol{\epsilon}_r\|_\infty = \mathcal{O}(r^{-a/2})$.

*Proof.* Since both $\mathbf{v}_r^s[k]$ and $\frac{1-\beta_2}{1-\beta_2^N}\sum_{j\in[N]}\beta_2^{(s,j)}\nabla\mathcal{L}_j(\mathbf{w}_r^0)[k]^2$ are positive and $|a^2 - b^2| = |a - b||a+b| \geq |a-b|^2$ holds for two positive numbers $a$ and $b$, Lemma D.2 implies that

$$\left| \sqrt{\mathbf{v}_r^s[k]} - \sqrt{\frac{1-\beta_2}{1-\beta_2^N}}\sqrt{\sum_{j\in[N]}\beta_2^{(s,j)}\nabla\mathcal{L}_j(\mathbf{w}_r^0)[k]^2} \right| \leq \sqrt{\epsilon_{\mathbf{v}}(t)}\max_{j\in[N]}|\nabla\mathcal{L}_j(\mathbf{w}_r^0)[k]|.$$

Therefore, we can rewrite $\mathbf{m}_r^s[k]$ and $\sqrt{\mathbf{v}_r^s[k]}$ as

$$\mathbf{m}_r^s[k] = \underbrace{\frac{1-\beta_1}{1-\beta_1^N}\sum_{j\in[N]}\beta_1^{(s,j)}\nabla\mathcal{L}_j(\mathbf{w}_r^0)[k]}_{(a)} + \underbrace{\epsilon_{\mathbf{m}}'(t)\max_{j\in[N]}|\nabla\mathcal{L}_j(\mathbf{w}_r^0)[k]|}_{(\epsilon_1)},$$

$$\sqrt{\mathbf{v}_r^s[k]} = \underbrace{\sqrt{\frac{1-\beta_2}{1-\beta_2^N}}\sqrt{\sum_{j\in[N]}\beta_2^{(s,j)}\nabla\mathcal{L}_j(\mathbf{w}_r^0)[k]^2}}_{(b)} + \underbrace{\sqrt{\epsilon_{\mathbf{v}}'(t)}\max_{j\in[N]}|\nabla\mathcal{L}_j(\mathbf{w}_r^0)[k]|}_{(\epsilon_2)},$$

for some error terms $\epsilon_{\mathbf{m}}'(t), \epsilon_{\mathbf{v}}'(t)$ such that $|\epsilon_{\mathbf{m}}'(t)| \leq \epsilon_{\mathbf{m}}(t), |\epsilon_{\mathbf{v}}'(t)| \leq \epsilon_{\mathbf{v}}(t)$. Note that $\left|\frac{a+\epsilon_1}{b+\epsilon_2} - \frac{a}{b}\right| \leq \left|\frac{\epsilon_1}{b+\epsilon_2}\right| + \left|\frac{a}{b}\cdot\frac{\epsilon_2}{b+\epsilon_2}\right| \leq \left|\frac{\epsilon_1}{b}\right| + \left|\frac{a}{b}\cdot\frac{\epsilon_2}{b}\right|$ for positive numbers $\epsilon_1, \epsilon_2, b$. Thus, we can conclude that

$$\left| \frac{\mathbf{m}_r^s[k]}{\sqrt{\mathbf{v}_r^s[k]}} - \frac{(a)}{(b)} \right| \leq \left| \frac{(\epsilon_1)}{(b)} \right| + \left| \frac{(a)}{(b)}\cdot\frac{(\epsilon_2)}{(b)} \right| \to 0, \tag{9}$$

since

$$\left| \frac{(\epsilon_1)}{(b)} \right| \leq \frac{1}{\sqrt{\frac{1-\beta_2}{1-\beta_2^N}}\sqrt{\beta_2^N}}\epsilon_{\mathbf{m}}(t) \to 0,$$

$$\left| \frac{(a)}{(b)} \right| \leq \frac{\frac{1-\beta_1}{1-\beta_1^N}}{\sqrt{\frac{1-\beta_2}{1-\beta_2^N}}}\sqrt{N},$$

$$\left| \frac{(\epsilon_2)}{(b)} \right| \leq \frac{1}{\sqrt{\frac{1-\beta_2}{1-\beta_2^N}}\sqrt{\beta_2^N}}\sqrt{\epsilon_{\mathbf{v}}(t)} \to 0.$$

Now consider the epoch-wise update. From above results, we get

$$\mathbf{w}_{r+1}^0[k] - \mathbf{w}_r^0[k] = -\sum_{s=0}^{N-1}\eta_s\frac{\mathbf{m}_r^s[k]}{\sqrt{\mathbf{v}_r^s[k]}}$$

$$= -\sum_{s=0}^{N-1}\eta_{rN+s}\left( C_{\mathrm{inc}}(\beta_1,\beta_2)\frac{\sum_{j\in[N]}\beta_1^{(s,j)}\nabla\mathcal{L}_j(\mathbf{w}_r^0)[k]}{\sqrt{\sum_{j\in[N]}\beta_2^{(s,j)}\nabla\mathcal{L}_j(\mathbf{w}_r^0)[k]^2}} + \boldsymbol{\epsilon}_{rN+s}[k] \right), \tag{10}$$

for some $\boldsymbol{\epsilon}_t \to \mathbf{0}$. Since $\lim_{t\to\infty}\eta_t = 0$, the difference between $\eta_{rN+s}$ for different $s \in [N]$ converges to 0, which proves the claim.

Next, we consider the case $\eta_t = (t+2)^{-a}$ for some $a \in (0,1]$. Then it is clear that

$$\epsilon_{\mathbf{m}}(t) = (1-\beta_1)e^{\alpha ND\eta_{rN}}c_2\eta_t + (e^{\alpha ND\eta_{rN}} - 1) + \beta_1^{t+1} = \mathcal{O}(t^{-a}),$$

$$\epsilon_{\mathbf{v}}(t) = 3(1-\beta_2)e^{2\alpha ND\eta_{rN}}c_2'\eta_t + 3(e^{2\alpha ND\eta_{rN}} - 1) + \beta_2^{t+1} = \mathcal{O}(t^{-a}),$$

where $D = \max_{j \in [N]} \|\mathbf{x}_j\|_1$. Therefore, from Equation (9), we get

$$
\left| \frac{\mathbf{m}_r^s[k]}{\sqrt{\mathbf{v}_r^s[k]}} - C_{\text{inc}}(\beta_1, \beta_2) \frac{\sum_{j \in [N]} \beta_1^{(s,j)} \nabla \mathcal{L}_j(\mathbf{w}_r^0)[k]}{\sqrt{\sum_{j \in [N]} \beta_2^{(s,j)} \nabla \mathcal{L}_j(\mathbf{w}_r^0)[k]^2}} \right| = \mathcal{O}(t^{-a/2}),
$$

which implies $\epsilon_t[k] = \mathcal{O}(t^{-a/2})$ in Equation (10). Note that

$$
\sum_{s=0}^{N-1} \eta_{rN+s} \left( \underbrace{C_{\text{inc}}(\beta_1, \beta_2) \frac{\sum_{j \in [N]} \beta_1^{(s,j)} \nabla \mathcal{L}_j(\mathbf{w}_r^0)[k]}{\sqrt{\sum_{j \in [N]} \beta_2^{(s,j)} \nabla \mathcal{L}_j(\mathbf{w}_r^0)[k]^2}}}_{\triangleq p(s)} + \epsilon_{rN+s}[k] \right)
$$

$$
= \eta_{rN} \sum_{s=0}^{N-1} \left( p(s) + \underbrace{\frac{\eta_{rN+s} - \eta_{rN}}{\eta_{rN}} p(s) + \frac{\eta_{rN+s}}{\eta_{rN}} \epsilon_{rN+s}[k]}_{\triangleq \epsilon'_{rN+s}[k]} \right).
$$

Furthermore,

$$
\frac{\eta_{rN} - \eta_{(r+1)N}}{\eta_{rN}} = 1 - \left( 1 + \frac{N}{rN+2} \right)^{-a} = \mathcal{O}(r^{-1}),
$$

from Lemma I.7. Since $p(s)$ is upper bounded by a constant from CS inequality, we get $\epsilon'_{rN+s}[k] = \mathcal{O}(r^{-a/2})$, which ends the proof. $\square$

# E    MISSING PROOFS IN SECTION 3

In this section, we provide the omitted proofs in Section 3. We first introduce the proof of Corollary 3.2 describing how SR datasets eliminate coordinate-adaptivity of `Inc-Adam`. Then, we review previous literature on the limit direction of weighted GD and prove Theorem 3.3.

## E.1    PROOF OF COROLLARY 3.2

**Corollary 3.2.** *Consider* `Inc-Adam` *iterates* $\{\mathbf{w}_t\}_{t=0}^{\infty}$ *on SR data. Then, under Assumptions 2.2 and 2.3, the epoch-wise update* $\mathbf{w}_{r+1}^0 - \mathbf{w}_r^0$ *can be approximated by weighted normalized GD, i.e.,*

$$
\mathbf{w}_{r+1}^0 - \mathbf{w}_r^0 = -\eta_{rN} \left( \sum_{i \in [N]} \frac{a_i(r)}{\|\nabla \mathcal{L}(\mathbf{w}_r^0)\|_2} \nabla \mathcal{L}_i(\mathbf{w}_r^0) + \epsilon_r \right), \tag{3}
$$

*where* $\lim_{r \to \infty} \epsilon_r = \mathbf{0}$ *and* $c_1 \le a_i(r) \le c_2$ *for some positive constants* $c_1, c_2$ *only depending on* $\beta_1, \beta_2, \{\mathbf{x}_i\}_{i \in [N]}$. *If* $\eta_t = (t+2)^{-a}$ *for some* $a \in (0, 1]$, *then* $\|\epsilon_r\|_{\infty} = \mathcal{O}(r^{-a/2})$.

*Proof.* Given SR data $\{\mathbf{x}_i\}_{i\in[N]}$, let $x_i = |\mathbf{x}_i[0]|$. Notice that

$$\sum_{i\in[N]} \frac{\sum_{j\in[N]} \beta_1^{(i,j)} \nabla\mathcal{L}_j(\mathbf{w}_r^0)}{\sqrt{\sum_{j\in[N]} \beta_2^{(i,j)} \nabla\mathcal{L}_j(\mathbf{w}_r^0)^2}} = \sum_{i\in[N]} \frac{\sum_{j\in[N]} \beta_1^{(i,j)} \nabla\mathcal{L}_j(\mathbf{w}_r^0)}{\sqrt{\sum_{l\in[N]} \beta_2^{(i,l)} |\ell'(\langle\mathbf{w}_r^0, \mathbf{x}_l\rangle)|^2 x_l^2}}$$

$$= \sum_{i\in[N]} \sum_{j\in[N]} \frac{\beta_1^{(i,j)}}{\sqrt{\sum_{l\in[N]} \beta_2^{(i,l)} |\ell'(\langle\mathbf{w}_r^0, \mathbf{x}_l\rangle)|^2 x_l^2}} \nabla\mathcal{L}_j(\mathbf{w}_r^0)$$

$$= \sum_{j\in[N]} \left( \sum_{i\in[N]} \frac{\beta_1^{(i,j)}}{\sqrt{\sum_{l\in[N]} \beta_2^{(i,l)} |\ell'(\langle\mathbf{w}_r^0, \mathbf{x}_l\rangle)|^2 x_l^2}} \right) \nabla\mathcal{L}_j(\mathbf{w}_r^0)$$

$$= \sum_{j\in[N]} \underbrace{\left( \sum_{i\in[N]} \frac{\beta_1^{(i,j)} \|\nabla\mathcal{L}(\mathbf{w}_r^0)\|_2}{\sqrt{\sum_{l\in[N]} \beta_2^{(i,l)} |\ell'(\langle\mathbf{w}_r^0, \mathbf{x}_l\rangle)|^2 x_l^2}} \right)}_{a_j(r)} \frac{\nabla\mathcal{L}_j(\mathbf{w}_r^0)}{\|\nabla\mathcal{L}(\mathbf{w}_r^0)\|_2}.$$

Therefore, it is enough to show that $a_j(r)$ is bounded. Note that

$$a_j(r) \le \frac{N}{\sqrt{\beta_2^{N-1}}} \frac{\|\nabla\mathcal{L}(\mathbf{w}_r^0)\|_2}{\sqrt{\sum_{l\in[N]} |\ell'(\langle\mathbf{w}_r^0, \mathbf{x}_l\rangle)|^2 x_l^2}} = \frac{1}{\sqrt{\beta_2^{N-1}}} \frac{\|\sum_{l\in[N]} |\ell'(\langle\mathbf{w}_r^0, \mathbf{x}_l\rangle)| \mathbf{x}_l\|_2}{\sqrt{\sum_{l\in[N]} |\ell'(\langle\mathbf{w}_r^0, \mathbf{x}_l\rangle)|^2 x_l^2}}$$

$$\le \frac{\sqrt{d}}{\sqrt{\beta_2^{N-1}}} \frac{\sum_{l\in[N]} |\ell'(\langle\mathbf{w}_r^0, \mathbf{x}_l\rangle)| x_l}{\sqrt{\sum_{l\in[N]} |\ell'(\langle\mathbf{w}_r^0, \mathbf{x}_l\rangle)|^2 x_l^2}} \le \frac{\sqrt{dN}}{\sqrt{\beta_2^{N-1}}}.$$

To find lower bound of $a_j(r)$, we use Assumption 2.1. Take $\mathbf{v} \in \mathbb{R}^d$ such that $\|\mathbf{v}\|_2 = 1$ and $\mathbf{v}^\top \mathbf{x}_i > 0, \forall i \in [N]$. Let $\gamma \triangleq \min_{i\in[N]} \mathbf{v}^\top \mathbf{x}_i > 0$. Note that

$$(-\mathbf{v})^\top \nabla\mathcal{L}(\mathbf{w}_r^0) = \frac{1}{N} \sum_{l\in[N]} (-\ell'(\langle\mathbf{w}_r^0, \mathbf{x}_l\rangle)) \cdot \mathbf{v}^\top \mathbf{x}_i \ge \frac{\gamma}{N} \sum_{l\in[N]} |\ell'(\langle\mathbf{w}_r^0, \mathbf{x}_l\rangle)|,$$

and by CS inequality,

$$\|\nabla\mathcal{L}(\mathbf{w}_r^0)\|_2 = \|-\mathbf{v}\|_2 \|\nabla\mathcal{L}(\mathbf{w}_r^0)\|_2 \ge \langle -\mathbf{v}, \nabla\mathcal{L}(\mathbf{w}_r^0)\rangle \ge \frac{\gamma}{N} \sum_{l\in[N]} |\ell'(\langle\mathbf{w}_r^0, \mathbf{x}_l\rangle)|. \quad (11)$$

Therefore, we can conclude that

$$a_j(r) \ge N\beta_1^{N-1} \frac{\|\nabla\mathcal{L}(\mathbf{w}_r^0)\|_2}{\sqrt{\sum_{l\in[N]} |\ell'(\langle\mathbf{w}_r^0, \mathbf{x}_l\rangle)|^2 x_l^2}} \overset{(*)}{\ge} \gamma\beta_1^{N-1} \frac{\sum_{l\in[N]} |\ell'(\langle\mathbf{w}_r^0, \mathbf{x}_l\rangle)|}{\sqrt{\sum_{l\in[N]} |\ell'(\langle\mathbf{w}_r^0, \mathbf{x}_l\rangle)|^2 x_l^2}}$$

$$\ge \frac{\gamma\beta_1^{N-1}}{\max_{l\in[N]} x_l}$$

where $(*)$ is from Equation (11). Now we can take $c_1 = \frac{\gamma\beta_1^{N-1}}{\max_{l\in[N]} x_l}$ and $c_2 = \frac{\sqrt{dN}}{\sqrt{\beta_2^{N-1}}}$ only depending on $\beta_1, \beta_2, \{\mathbf{x}_i\}$. $\quad\square$

### E.2 PROOF OF THEOREM 3.3

**Related Work.** We now turn to the proof of Theorem 3.3, building upon the foundational work of Ji et al. (2020), who characterized the convergence direction of GD via its regularization path. Subsequent research has extended this characterization to weighted GD, which optimizes the weighted empirical risk $\mathcal{L}_{\mathbf{q}(t)}(\mathbf{w}) = \sum_{i\in[N]} q_i(t)\ell(\mathbf{w}^\top \mathbf{x}_i)$. Xu et al. (2021) proved that weighted GD converges to $\ell_2$-max-margin direction on the same linear classification task when the weights are fixed during training. This condition was later relaxed by Zhai et al. (2023), who demonstrated that the same convergence guarantee holds provided the weights converge to a limit, i.e., $\exists \lim_{t\to\infty} \mathbf{q}(t) = \hat{\mathbf{q}}$.

Our setting, however, introduces distinct technical challenges. First, the weights are bounded but not guaranteed to converge. The most relevant existing result is Theorem 7 in Zhai et al. (2023), which establishes the same limit direction but requires the stronger combined assumptions of lower-bounded weights, loss convergence, and directional convergence of the iterates. A further complication in our analysis is an additional error term, $\epsilon_r$ in Corollary 3.2, which must be carefully controlled. Our fine-grained analysis overcomes these issues by extending the methodology of Ji et al. (2020), enabling us to manage the error term under the sole, weaker assumption of loss convergence.

**Definition E.1.** Given $\boldsymbol{a} = (a_1, \cdots, a_N) \in \mathbb{R}^N$, we define $\boldsymbol{a}$-weighted loss as $\mathcal{L}^{\boldsymbol{a}}(\mathbf{w}) \triangleq \sum_{i \in [N]} a_i \mathcal{L}_i(\mathbf{w})$. We denote the regularized solution as $\bar{\mathbf{w}}^{\boldsymbol{a}}(B) \triangleq \arg \min_{\|\mathbf{w}\|_2 \leq B} \mathcal{L}^{\boldsymbol{a}}(\mathbf{w})$.

By introducing $\boldsymbol{a}$-weighted loss, we can regard weighted GD as vanilla GD with respect to weighted loss. To follow the line of Ji et al. (2020), we show that the regularization path converges in direction to $\ell_2$-max-margin solution, regardless of the choice of the weight vector $\boldsymbol{a}$ if it is bounded by two positive constants, and such convergence is uniform; we can take sufficiently large $B$ to be close the $\ell_2$ solution *for any* $\boldsymbol{a} \in [c_1, c_2]^N$.

**Lemma E.2** (Adaptation of Proposition 10 in Ji et al. (2020)). *Let* $\hat{\mathbf{u}} = \arg \max_{\|\mathbf{v}\|_2 \leq 1} \min_{i \in [N]} \langle \mathbf{v}, \mathbf{x}_i \rangle$ *be the (unique) $\ell_2$-max-margin solution and $c_1, c_2$ be two positive constants. Then, for any $\boldsymbol{a} \in [c_1, c_2]^N$,*

$$\lim_{B \to \infty} \frac{\bar{\mathbf{w}}^{\boldsymbol{a}}(B)}{B} = \hat{\mathbf{u}}.$$

*Furthermore, given $\epsilon > 0$, there exists $M(c_1, c_2, \epsilon, N) > 0$ only depending on $c_1, c_2, \epsilon, N$ such that $B > M$ implies $\|\frac{\bar{\mathbf{w}}^{\boldsymbol{a}}(B)}{B} - \hat{\mathbf{u}}\| < \epsilon$ for any $\boldsymbol{a} \in [c_1, c_2]^N$.*

*Proof.* We first have to show the uniqueness of $\ell_2$-max-margin solution. This proof was introduced by Ji et al. (2020, Proposition 10), but we provide it for completeness. Suppose that there exist two distinct unit vectors $\mathbf{u}_1$ and $\mathbf{u}_2$ such that both of them achieve the max-margin $\hat{\gamma}$. Take $\mathbf{u}_3 = \frac{\mathbf{u}_1 + \mathbf{u}_2}{2}$ as a middle point of $\mathbf{u}_1$ and $\mathbf{u}_2$. Then we get

$$\mathbf{u}_3^\top \mathbf{x}_i = \frac{1}{2}(\mathbf{u}_1^\top \mathbf{x}_i + \mathbf{u}_2^\top \mathbf{x}_i) \geq \hat{\gamma},$$

for all $i \in [N]$, which implies that $\min_{i \in [N]} \mathbf{u}_3^\top \mathbf{x}_i \geq \hat{\gamma}$. Since $\mathbf{u}_1 \neq \mathbf{u}_2$, we get $\|\mathbf{u}_3\| < 1$, implying that $\frac{\mathbf{u}_3}{\|\mathbf{u}_3\|}$ achieves a larger margin than $\hat{\gamma}$. This makes a contradiction.

Now we prove the main claim. Let $\hat{\gamma} = \min_{i \in [N]} \langle \hat{\mathbf{u}}, \mathbf{x}_i \rangle$ be the margin of $\hat{\mathbf{u}}$. Then, it satisfies

$$c_1 \ell(\min_{i \in [N]} \langle \bar{\mathbf{w}}^{\boldsymbol{a}}(B), \mathbf{x}_i \rangle) \leq \mathcal{L}^{\boldsymbol{a}}(\bar{\mathbf{w}}^{\boldsymbol{a}}(B)) \leq \mathcal{L}^{\boldsymbol{a}}(B\hat{\mathbf{u}}) \leq N c_2 \ell(B\hat{\gamma}). \tag{12}$$

For $\ell = \ell_{\exp}$, we get $\min_{i \in [N]} \langle \bar{\mathbf{w}}^{\boldsymbol{a}}(B), \mathbf{x}_i \rangle \geq B\hat{\gamma} - \log \frac{N c_2}{c_1}$, which implies

$$\min_{i \in [N]} \langle \frac{\bar{\mathbf{w}}^{\boldsymbol{a}}(B)}{B}, \mathbf{x}_i \rangle \geq \hat{\gamma} - \frac{1}{B} \log \frac{N c_2}{c_1}. \tag{13}$$

Since $\ell_2$-max-margin solution is unique, $\frac{\bar{\mathbf{w}}^{\boldsymbol{a}}(B)}{B}$ converges to $\hat{\mathbf{u}}$. Note that the lower bound in Equation (13) does not depend on $\boldsymbol{a} \in [c_1, c_2]^N$. Therefore, the choice of $M$ in Lemma E.2 only depends on $c_1, c_2, \epsilon, N$.

For $\ell = \ell_{\log}$, Equation (12) implies that $\ell(\min_{i \in [N]} \langle \bar{\mathbf{w}}^{\boldsymbol{a}}(B), \mathbf{x}_i \rangle) \leq \frac{N c_2}{c_1} \ell(B\hat{\gamma})$. Notice that $\frac{N c_2}{c_1} > 1$ and $\min_{i \in [N]} \langle \bar{\mathbf{w}}^{\boldsymbol{a}}(B), \mathbf{x}_i \rangle > 0, B\hat{\gamma} > 0$ hold for sufficiently large $B$ from Lemma I.2. From Lemma I.5, we get

$$\min_{i \in [N]} \langle \frac{\bar{\mathbf{w}}^{\boldsymbol{a}}(B)}{B}, \mathbf{x}_i \rangle \geq \hat{\gamma} - \frac{1}{B} \log(2^{\frac{N c_2}{c_1}} - 1).$$

Following the proof of the previous part, we can easily show that the statement also holds in this case. $\qquad\square$

**Lemma E.3** (Adaptation of Lemma 9 in Ji et al. (2020)). *Let $\alpha, c_1, c_2 > 0$ be given. Then, there exists $\rho(\alpha) > 0$ such that $\|\mathbf{w}\|_2 > \rho(\alpha) \Rightarrow \mathcal{L}^{\boldsymbol{a}}((1+\alpha)\|\mathbf{w}\|_2 \hat{\mathbf{u}}) \leq \mathcal{L}^{\boldsymbol{a}}(\mathbf{w})$ for any $\boldsymbol{a} \in [c_1, c_2]^N$.*

*Proof.* Let $\hat{\mathbf{u}}$ be the $\ell_2$-max-margin solution and $\hat{\gamma} = \max_{i \in [N]} \langle \hat{\mathbf{u}}, \mathbf{x}_i \rangle$ be its margin. From the uniform convergence in Lemma E.2, we can choose $\rho(\alpha)$ large enough so that

$$\|\mathbf{w}\|_2 > \rho(\alpha) \Rightarrow \left\| \frac{\bar{\mathbf{w}}^{\boldsymbol{a}}(\|\mathbf{w}\|_2)}{\|\mathbf{w}\|_2} - \hat{\mathbf{u}} \right\|_2 \leq \alpha\hat{\gamma},$$

*for any $\boldsymbol{a} \in [c_1, c_2]^N$.* For $1 \leq i \leq n$, we get

$$\begin{aligned}
\langle \bar{\mathbf{w}}^{\boldsymbol{a}}(\|\mathbf{w}\|_2), \mathbf{x}_i \rangle &= \langle \bar{\mathbf{w}}^{\boldsymbol{a}}(\|\mathbf{w}\|_2) - \|\mathbf{w}\|_2\hat{\mathbf{u}}, \mathbf{x}_i \rangle + \langle \|\mathbf{w}\|_2\hat{\mathbf{u}}, \mathbf{x}_i \rangle \\
&\leq \alpha\hat{\gamma}\|\mathbf{w}\|_2 + \langle \|\mathbf{w}\|_2\hat{\mathbf{u}}, \mathbf{x}_i \rangle \\
&\leq (1 + \alpha)\|\mathbf{w}\|_2\langle \hat{\mathbf{u}}, \mathbf{x}_i \rangle.
\end{aligned}$$

This implies that

$$\mathcal{L}^{\boldsymbol{a}}((1 + \alpha)\|\mathbf{w}\|_2\hat{\mathbf{u}}) \leq \mathcal{L}^{\boldsymbol{a}}(\bar{\mathbf{w}}^{\boldsymbol{a}}(\|\mathbf{w}\|_2)) \leq \mathcal{L}^{\boldsymbol{a}}(\mathbf{w}),$$

for any $\boldsymbol{a} \in [c_1, c_2]^N$. $\qquad\square$

**Theorem 3.3.** *Consider* Inc-Adam *iterates $\{\mathbf{w}_t\}_{t=0}^{\infty}$ with $\beta_1 \leq \beta_2$ on SR data under Assumptions 2.1 to 2.3. If (a) $\mathcal{L}(\mathbf{w}_t) \to 0$ as $t \to \infty$ and (b) $\eta_t = (t + 2)^{-a}$ for $a \in (2/3, 1]$, then it satisfies*

$$\lim_{t \to \infty} \frac{\mathbf{w}_t}{\|\mathbf{w}_t\|_2} = \hat{\mathbf{w}}_{\ell_2},$$

*where $\hat{\mathbf{w}}_{\ell_2}$ denotes the (unique) $\ell_2$-max-margin solution of SR data $\{\mathbf{x}_i\}_{i \in [N]}$.*

*Proof.* From Corollary 3.2, we can rewrite the update as

$$\begin{aligned}
\mathbf{w}_{r+1}^0 - \mathbf{w}_r^0 &= -\frac{\eta_{rN}}{\|\nabla\mathcal{L}(\mathbf{w}_r^0)\|_2} \sum_{i \in [N]} a_i(r)\nabla\mathcal{L}_i(\mathbf{w}_r^0) - \eta_{rN}\boldsymbol{\epsilon}_r \\
&= -\frac{\eta_{rN}}{\|\nabla\mathcal{L}(\mathbf{w}_r^0)\|_2}\nabla\mathcal{L}^{\boldsymbol{a}(r)}(\mathbf{w}_r^0) - \eta_{rN}\boldsymbol{\epsilon}_r,
\end{aligned}$$

where $c_1 \leq a_i(r) \leq c_2$ for some positive constants $c_1, c_2$ and $\lim_{r \to \infty} \boldsymbol{\epsilon}_r = \mathbf{0}$.

First, we show that $\lim_{r \to \infty} \frac{\mathbf{w}_r^0}{\|\mathbf{w}_r^0\|_2} = \hat{\mathbf{w}}_{\ell_2}$. Let $\epsilon > 0$ be given. Then, we can take $\alpha = \frac{\epsilon}{1-\epsilon}$ so that $\frac{1}{1+\alpha} = 1 - \epsilon$. Since $\|\mathbf{w}_t\|_2 \to \infty$, we can choose $r_0$ such that $t \geq r_0 N \implies \|\mathbf{w}_t\|_2 > \max\{\rho(\alpha), 1\}$, where $\rho(\alpha)$ is given by Lemma E.3. Then for any $r \geq r_0$, we get

$$\langle \nabla\mathcal{L}^{\boldsymbol{a}}(\mathbf{w}_r^0), \mathbf{w}_r^0 - (1 + \alpha)\|\mathbf{w}_r^0\|_2\hat{\mathbf{u}} \rangle \geq \mathcal{L}^{\boldsymbol{a}}(\mathbf{w}_r^0) - \mathcal{L}^{\boldsymbol{a}}((1 + \alpha)\|\mathbf{w}_r^0\|_2\hat{\mathbf{u}}) \geq 0,$$

which implies

$$\langle \nabla\mathcal{L}^{\boldsymbol{a}}(\mathbf{w}_r^0), \mathbf{w}_r^0 \rangle \geq (1 + \alpha)\|\mathbf{w}_r^0\|_2\langle \nabla\mathcal{L}^{\boldsymbol{a}}(\mathbf{w}_r^0), \hat{\mathbf{u}} \rangle.$$

Therefore, we get

$$\begin{aligned}
&\langle \mathbf{w}_{r+1}^0 - \mathbf{w}_r^0, \hat{\mathbf{u}} \rangle \\
&= \langle -\frac{\eta_{rN}}{\|\nabla\mathcal{L}(\mathbf{w}_r^0)\|_2}\nabla\mathcal{L}^{\boldsymbol{a}(r)}(\mathbf{w}_r^0), \hat{\mathbf{u}} \rangle + \langle -\eta_{rN}\boldsymbol{\epsilon}_r, \hat{\mathbf{u}} \rangle \\
&\geq \frac{1}{(1 + \alpha)\|\mathbf{w}_r^0\|_2}\langle -\frac{\eta_{rN}}{\|\nabla\mathcal{L}(\mathbf{w}_r^0)\|_2}\nabla\mathcal{L}^{\boldsymbol{a}(r)}(\mathbf{w}_r^0), \mathbf{w}_r^0 \rangle + \langle -\eta_{rN}\boldsymbol{\epsilon}_r, \hat{\mathbf{u}} \rangle \\
&= \frac{1}{(1 + \alpha)\|\mathbf{w}_r^0\|_2}\langle \mathbf{w}_{r+1}^0 - \mathbf{w}_r^0, \mathbf{w}_r^0 \rangle + \frac{1}{(1 + \alpha)\|\mathbf{w}_r^0\|_2}\langle \eta_{rN}c, \mathbf{w}_r^0 \rangle + \langle -\eta_{rN}\boldsymbol{\epsilon}_r, \hat{\mathbf{u}} \rangle \\
&= \frac{1}{(1 + \alpha)\|\mathbf{w}_r^0\|_2}\left( \frac{1}{2}\|\mathbf{w}_{r+1}^0\|_2^2 - \frac{1}{2}\|\mathbf{w}_r^0\|_2^2 - \frac{1}{2}\|\mathbf{w}_{r+1}^0 - \mathbf{w}_r^0\|_2^2 \right) + \langle -\eta_{rN}\boldsymbol{\epsilon}_r, \hat{\mathbf{u}} - \frac{\mathbf{w}_r^0}{(1 + \alpha)\|\mathbf{w}_r^0\|_2} \rangle \\
&\geq \frac{1}{(1 + \alpha)\|\mathbf{w}_r^0\|_2}\left( \frac{1}{2}\|\mathbf{w}_{r+1}^0\|_2^2 - \frac{1}{2}\|\mathbf{w}_r^0\|_2^2 - \frac{1}{2}\|\mathbf{w}_{r+1}^0 - \mathbf{w}_r^0\|_2^2 \right) - 2\eta_{rN}\|\boldsymbol{\epsilon}_r\|_2,
\end{aligned}$$

where the last inequality is from $\langle \eta_{rN}\boldsymbol{\epsilon}_r, \hat{\mathbf{u}} - \frac{\mathbf{w}_r^0}{(1+\alpha)\|\mathbf{w}_r^0\|_2} \rangle \leq \eta_{rN}\|\boldsymbol{\epsilon}_r\|_2 \left\| \hat{\mathbf{u}} - \frac{\mathbf{w}_r^0}{(1+\alpha)\|\mathbf{w}_r^0\|} \right\|_2 \leq 2\eta_{rN}\|\boldsymbol{\epsilon}_r\|_2$.

Note that

$$\frac{\frac{1}{2}\|\mathbf{w}_{r+1}^0\|_2^2 - \frac{1}{2}\|\mathbf{w}_r^0\|_2^2}{\|\mathbf{w}_r^0\|_2} \geq \|\mathbf{w}_{r+1}^0\|_2 - \|\mathbf{w}_r^0\|_2.$$

Furthermore,

$$\frac{\|\mathbf{w}_{r+1}^0 - \mathbf{w}_r^0\|_2^2}{2(1+\alpha)\|\mathbf{w}_r^0\|_2} \leq \frac{\|\mathbf{w}_{r+1}^0 - \mathbf{w}_r^0\|_2^2}{2} \leq \frac{1}{2}\left(\eta_{rN}^2 \frac{\|\nabla\mathcal{L}^{\boldsymbol{a}(r)}(\mathbf{w}_r^0)\|^2}{\|\nabla\mathcal{L}(\mathbf{w}_r^0)\|_2^2} + \eta_{rN}\|\boldsymbol{\epsilon}_r\|_2^2\right)$$

$$\leq c_3 r^{-2a},$$

for some $c_3 > 0$ and sufficiently large $r$, since $\eta_{rN} = \mathcal{O}(r^{-a})$, $\|\boldsymbol{\epsilon}_r\| = \mathcal{O}(r^{-a/2})$, and $\frac{\|\nabla\mathcal{L}^{\boldsymbol{a}(r)}(\mathbf{w}_r^0)\|^2}{\|\nabla\mathcal{L}(\mathbf{w}_r^0)\|^2}$ is upper bounded from

$$\frac{\|\nabla\mathcal{L}^{\boldsymbol{a}(r)}(\mathbf{w}_r^0)\|_2^2}{\|\nabla\mathcal{L}(\mathbf{w}_r^0)\|_2^2} \overset{(*)}{\leq} \frac{\left(c_2\sqrt{d}\max_{i\in[N]} x_i \sum_{i\in[N]} |\ell'(\langle\mathbf{w}_r^0,\mathbf{x}_l\rangle)|\right)^2}{\left(\frac{\gamma}{N}\sum_{i\in[N]} |\ell'(\langle\mathbf{w}_r^0,\mathbf{x}_i\rangle)|\right)^2} = \frac{c_2^2 d N^2 (\max_{i\in[N]} x_i)^2}{\gamma^2},$$

with $\gamma = \min_{i\in[N]}\langle\hat{\mathbf{w}}_{\ell_2},\mathbf{x}_i\rangle > 0$. Note that $(*)$ is from

$$\|\nabla\mathcal{L}(\mathbf{w}_r^0)\|_2^2 = \|\hat{\mathbf{w}}_{\ell_2}\|_2^2\|\nabla\mathcal{L}(\mathbf{w}_r^0)\|_2^2 \geq \langle\hat{\mathbf{w}}_{\ell_2}, \frac{1}{N}\sum_{i\in[N]}\ell'(\langle\mathbf{w}_r^0,\mathbf{x}_i\rangle)\mathbf{x}_i\rangle^2 \geq \left(\frac{\gamma}{N}\sum_{i\in[N]} |\ell'(\langle\mathbf{w}_r^0,\mathbf{x}_i\rangle)|\right)^2.$$

Therefore, we get

$$\langle\mathbf{w}_r^0 - \mathbf{w}_{r_0}^0, \hat{\mathbf{u}}\rangle \geq \frac{\|\mathbf{w}_r^0\|_2 - \|\mathbf{w}_{r_0}^0\|_2}{1+\alpha} - \sum_{s=r_0}^{r} c_3 s^{-2a} - 2\sum_{s=r_0}^{r} \eta_{sN}\|\boldsymbol{\epsilon}_s\|_2$$

$$\geq (1-\epsilon)(\|\mathbf{w}_r^0\|_2 - \|\mathbf{w}_{r_0}^0\|_2) - \underbrace{\left(\sum_{s=r_0}^{\infty} c_3 s^{-2a} + \sum_{s=r_0}^{\infty} c_4 s^{-\frac{3}{2}a}\right)}_{=c_5 < \infty},$$

since $\|\boldsymbol{\epsilon}_r\| = \mathcal{O}(r^{-a/2})$ and $a \in (2/3, 1]$. As a result, we can conclude that

$$\langle\frac{\mathbf{w}_r^0}{\|\mathbf{w}_r^0\|_2}, \hat{\mathbf{u}}\rangle \geq \frac{(1-\epsilon)(\|\mathbf{w}_r^0\|_2 - \|\mathbf{w}_{r_0}^0\|_2) + \langle\mathbf{w}_{r_0}^0, \hat{\mathbf{u}}\rangle + c_5}{\|\mathbf{w}_r\|_2},$$

which implies

$$\liminf_{r\to\infty}\langle\frac{\mathbf{w}_r^0}{\|\mathbf{w}_r^0\|_2}, \hat{\mathbf{u}}\rangle \geq 1 - \epsilon.$$

Since we choose $\epsilon > 0$ arbitrarily, we get $\lim_{r\to\infty}\frac{\mathbf{w}_r^0}{\|\mathbf{w}_r^0\|_2} = \hat{\mathbf{w}}_{\ell_2}$.

Second, we claim that $\lim_{t\to\infty}\frac{\mathbf{w}_t}{\|\mathbf{w}_t\|_2} = \hat{\mathbf{w}}_{\ell_2}$. It suffices to show that $\lim_{r\to\infty}\left\|\frac{\mathbf{w}_r^0}{\|\mathbf{w}_r^0\|_2} - \frac{\mathbf{w}_r^s}{\|\mathbf{w}_r^s\|_2}\right\|_2 = 0$ for all $s \in [N]$. Note that

$$\left\|\frac{\mathbf{w}_r^0}{\|\mathbf{w}_r^0\|_2} - \frac{\mathbf{w}_r^s}{\|\mathbf{w}_r^s\|_2}\right\|_2 \leq \left\|\frac{\mathbf{w}_r^0}{\|\mathbf{w}_r^0\|_2} - \frac{\mathbf{w}_r^0}{\|\mathbf{w}_r^s\|_2}\right\|_2 + \left\|\frac{\mathbf{w}_r^0}{\|\mathbf{w}_r^s\|_2} - \frac{\mathbf{w}_r^s}{\|\mathbf{w}_r^s\|_2}\right\|_2$$

$$\leq \frac{\|\mathbf{w}_r^s\|_2 - \|\mathbf{w}_r^0\|_2}{\|\mathbf{w}_r^s\|_2} + \frac{\|\mathbf{w}_r^s - \mathbf{w}_r^0\|_2}{\|\mathbf{w}_r^s\|_2}$$

$$\leq 2\frac{\|\mathbf{w}_r^s - \mathbf{w}_r^0\|_2}{\|\mathbf{w}_r^s\|_2} \to 0,$$

which ends the proof. □

# F    MISSING PROOFS IN SECTION 4

## F.1    PROOF OF PROPOSITION 4.1

**Proposition 4.1.** *Let $\{\mathbf{w}_t\}_{t=0}^{\infty}$ be the iterates of* Inc-Adam *with $\beta_1 \leq \beta_2$. Then, under Assumptions 2.2 and 2.3, the epoch-wise update $\mathbf{w}_{r+1}^0 - \mathbf{w}_r^0$ can be expressed as*

$$\mathbf{w}_{r+1}^0 - \mathbf{w}_r^0 = -\eta_{rN} \left( \sqrt{\frac{1 - \beta_2^N}{1 - \beta_2}} \frac{\nabla \mathcal{L}(\mathbf{w}_r^0)}{\sqrt{\sum_{i=1}^N \nabla \mathcal{L}_i(\mathbf{w}_r^0)^2}} + \boldsymbol{\epsilon}_{\beta_2}(r) \right),$$

*where $\limsup_{r \to \infty} \|\boldsymbol{\epsilon}_{\beta_2}(r)\|_{\infty} \leq \epsilon(\beta_2)$ and $\lim_{\beta_2 \to 1} \epsilon(\beta_2) = 0$.*

*Proof.* Note that

$$\sum_{i \in [N]} \frac{\sum_{j \in [N]} \beta_1^{(i,j)} \nabla \mathcal{L}_j(\mathbf{w}_r^0)[k]}{\sqrt{\sum_{j \in [N]} \nabla \mathcal{L}_j(\mathbf{w}_r^0)[k]^2}} = \frac{\sum_{j \in [N]} \left( \sum_{i \in [N]} \beta_1^{(i,j)} \nabla \mathcal{L}_j(\mathbf{w}_r^0)[k] \right)}{\sqrt{\sum_{j \in [N]} \nabla \mathcal{L}_j(\mathbf{w}_r^0)[k]^2}}$$

$$= \frac{1 - \beta_1^N}{1 - \beta_1} \frac{\nabla \mathcal{L}(\mathbf{w}_r^0)[k]}{\sqrt{\sum_{i=1}^N \nabla \mathcal{L}_i(\mathbf{w}_r^0)[k]^2}}.$$

Furthermore,

$$\left| \frac{\sum_{j \in [N]} \beta_1^{(i,j)} \nabla \mathcal{L}_j(\mathbf{w}_r^0)[k]}{\sqrt{\sum_{j \in [N]} \beta_2^{(i,j)} \nabla \mathcal{L}_j(\mathbf{w}_r^0)[k]^2}} - \frac{\sum_{j \in [N]} \beta_1^{(i,j)} \nabla \mathcal{L}_j(\mathbf{w}_r^0)[k]}{\sqrt{\sum_{j \in [N]} \nabla \mathcal{L}_j(\mathbf{w}_r^0)[k]^2}} \right|$$

$$\leq \left| \frac{\sum_{j \in [N]} \beta_1^{(i,j)} \nabla \mathcal{L}_j(\mathbf{w}_r^0)[k]}{\sqrt{\sum_{j \in [N]} \beta_2^{(i,j)} \nabla \mathcal{L}_j(\mathbf{w}_r^0)[k]^2}} \right| \left| 1 - \frac{\sqrt{\sum_{j \in [N]} \beta_2^{(i,j)} \nabla \mathcal{L}_j(\mathbf{w}_r^0)[k]^2}}{\sqrt{\sum_{j \in [N]} \nabla \mathcal{L}_j(\mathbf{w}_r^0)[k]^2}} \right|$$

$$\leq \sqrt{\sum_{j \in [N]} \frac{\beta_1^{(i,j)2}}{\beta_2^{(i,j)}} \left( 1 - \sqrt{\beta_2^{N-1}} \right)} \leq \underbrace{\sqrt{\sum_{j \in [N]} \frac{1}{\beta_2^{(i,j)}} \left( 1 - \sqrt{\beta_2^{N-1}} \right)}}_{\triangleq \epsilon(\beta_2)},$$

where $\lim_{\beta_2 \to 1} \epsilon(\beta_2) = 0$. Substituting to Equation (2), we get

$$\mathbf{w}_{r+1}^0[k] - \mathbf{w}_r^0[k] = -\eta_{rN} \left( C_{\text{inc}}(\beta_1, \beta_2) \frac{1 - \beta_1^N}{1 - \beta_1} \frac{\nabla \mathcal{L}(\mathbf{w}_r^0)[k]}{\sqrt{\sum_{i=1}^N \nabla \mathcal{L}_i(\mathbf{w}_r^0)[k]^2}} + \boldsymbol{\epsilon}_{\beta_2}(r)[k] \right)$$

$$= -\eta_{rN} \left( C_{\text{proxy}}(\beta_2) \frac{\nabla \mathcal{L}(\mathbf{w}_r^0)[k]}{\sqrt{\sum_{i=1}^N \nabla \mathcal{L}_i(\mathbf{w}_r^0)[k]^2}} + \boldsymbol{\epsilon}_{\beta_2}(r)[k] \right),$$

where $C_{\text{proxy}}(\beta_2) = \sqrt{\frac{1 - \beta_2^N}{1 - \beta_2}}$, $\limsup_{r \to \infty} \|\boldsymbol{\epsilon}_{\beta_2}(r)\|_{\infty} \leq N\epsilon(\beta_2)$, and $\lim_{\beta_2 \to 1} \epsilon(\beta_2) = 0$.    □

## F.2    PROOF OF PROPOSITION 4.3

To prove Proposition 4.3, we begin with identifying AdamProxy as normalized steepest descent with respect to an energy norm, where the inducing matrix depends on the current iterate and the dataset. The following lemma shows that the matrix is always non-degenerate; the energy norm is bounded above and below with respect to $\ell_2$-norm multiplied by two constants only depending on the dataset. This result takes a crucial role to make the convergence guarantee of AdamProxy.

**Lemma F.1.** *Consider* AdamProxy *iterates $\{\mathbf{w}_t\}$ under Assumptions 2.1 and 2.2. Then, it satisfies*

*(a)* $\mathrm{Prx}(\mathbf{w}) = \underset{\|\mathbf{v}\|_{\mathbf{P}(\mathbf{w})}=1}{\arg\min} \langle \nabla\mathcal{L}(\mathbf{w}), \mathbf{v} \rangle$, *where* $\tilde{\mathbf{P}}(\mathbf{w}) = \mathrm{diag}\left(\sqrt{\sum_{i\in[N]} \nabla\mathcal{L}_i(\mathbf{w})^2}\right)$ *and* $\mathbf{P}(\mathbf{w}) = \frac{1}{\|\nabla\mathcal{L}(\mathbf{w})\|^2_{\tilde{\mathbf{P}}^{-1}(\mathbf{w})}}\tilde{\mathbf{P}}(\mathbf{w})$.

*(b) There exist positive constants $c_1, c_2$ depending only on the dataset $\{\mathbf{x}_i\}_{i\in[N]}$ such that $c_1\|\mathbf{v}\|_2 \leq \|\mathbf{v}\|_{\mathbf{P}(\mathbf{w})} \leq c_2\|\mathbf{v}\|_2$ for all $\mathbf{v}, \mathbf{w} \in \mathbb{R}^d$.*

*Proof.* (a) Note that $\mathrm{Prx}(\mathbf{w}) = -\tilde{\mathbf{P}}(\mathbf{w})^{-1}\nabla\mathcal{L}(\mathbf{w}) = \arg\min_{\mathbf{v}} \langle \nabla\mathcal{L}(\mathbf{w}), \mathbf{v} \rangle + \frac{1}{2}\|\mathbf{v}\|^2_{\tilde{\mathbf{P}}(\mathbf{w})}$. Therefore, normalizing by $\|\nabla\mathcal{L}(\mathbf{w})\|^2_{\tilde{\mathbf{P}}^{-1}(\mathbf{w})}$, we get $\mathrm{Prx}(\mathbf{w}) = \underset{\|\mathbf{v}\|_{\mathbf{P}(\mathbf{w})}=1}{\arg\min} \langle \nabla\mathcal{L}(\mathbf{w}), \mathbf{v} \rangle$

(b) It is enough to show that every element of $\mathbf{P}(\mathbf{w})$ is bounded for some $c_1, c_2 > 0$. For simplicity, we denote $|\ell'(\mathbf{w}^\top \mathbf{x}_i)| = r_i$, $\min_{i\in[N],j\in[d]} |\mathbf{x}_i[j]| = B_1 > 0$ and $\max_{i\in[N],j\in[d]} |\mathbf{x}_i[j]| = B_2 > 0$.

Note that

$$\mathbf{P}(\mathbf{w})[k,k] = \sqrt{\sum_{i\in[N]} r_i^2 \mathbf{x}_i[k]^2} \times \frac{1}{\sum_{j\in[d]} \frac{\nabla\mathcal{L}(\mathbf{w})[j]^2}{\sqrt{\sum_{i\in[N]} r_i^2 \mathbf{x}_i[j]^2}}}$$

$$\geq B_1 \sqrt{\sum_{i\in[N]} r_i^2} \times \frac{1}{\sum_{j\in[d]} \frac{(\sum_{i\in[N]} r_i B_2)^2}{\sqrt{\sum_{i\in[N]} r_i^2 B_1^2}}}$$

$$= \frac{B_1^2}{B_2^2} \cdot \frac{1}{d} \frac{\sum_{i\in[N]} r_i^2}{(\sum_{i\in[N]} r_i)^2} \geq \frac{1}{Nd} \cdot \frac{B_1^2}{B_2^2}.$$

Let $\mathbf{v} \in \mathbb{R}^d$ s.t. $\|\mathbf{v}\|_2 = 1$ and $\mathbf{v}^\top \mathbf{x}_i > 0, \forall i \in [N]$ (since $\{\mathbf{x}_i\}$ is linearly separable). Let $\min_{i\in[N]} \mathbf{v}^\top \mathbf{x}_i = \gamma > 0$. Then, we get $\mathbf{v}^\top \nabla\mathcal{L}(\mathbf{w}) = \sum_{i\in[N]} r_i \mathbf{v}^\top \mathbf{x}_i \geq \gamma \sum_{i\in[N]} r_i$, which implies $\|\mathbf{v}\|^2_{\tilde{\mathbf{P}}(\mathbf{w})} \|\nabla\mathcal{L}(\mathbf{w})\|^2_{\tilde{\mathbf{P}}(\mathbf{w})^{-1}} \geq \langle \mathbf{v}, \nabla\mathcal{L}(\mathbf{w}) \rangle^2 \geq \gamma^2 \left(\sum_{i\in[N]} r_i\right)^2$

Note that $\|\mathbf{v}\|^2_{\tilde{\mathbf{P}}(\mathbf{w})} = \sum_{j\in[d]} \left(\sum_{i\in[N]} r_i^2 |\mathbf{x}_i[j]|^2 \cdot \mathbf{v}[j]^2\right) \leq dB_2 \sqrt{\sum_{i\in[N]} r_i^2}$. To wrap up, we get

$$\|\nabla\mathcal{L}(\mathbf{w})\|^2_{\tilde{\mathbf{P}}(\mathbf{w})^{-1}} \geq \frac{\gamma^2}{dB_2} \frac{(\sum_{i\in[N]} r_i)^2}{\sqrt{\sum_{i\in[N]} r_i^2}},$$

and therefore,

$$\mathbf{P}(\mathbf{w})[k,k] = \frac{\sqrt{\sum_{i\in[N]} r_i^2 \mathbf{x}_i[k]^2}}{\|\nabla\mathcal{L}(\mathbf{w})\|^2_{\tilde{\mathbf{P}}(\mathbf{w})^{-1}}} \leq \sqrt{\sum_{i\in[N]} r_i^2 \mathbf{x}_i[k]^2} \frac{dB_2}{\gamma^2} \frac{\sqrt{\sum_{i\in[N]} r_i^2}}{(\sum_{i\in[N]} r_i)^2} \leq \frac{dB_2^2}{\gamma^2}.$$

As a result, we can conclude that

$$\frac{B_1^2}{dB_2^2 N} \|\mathbf{v}\| \leq \|\mathbf{v}\|_{\mathbf{P}(\mathbf{w})} \leq \frac{dB_2^2}{\gamma^2} \|\mathbf{v}\|, \quad \forall \mathbf{v}, \mathbf{w} \in \mathbb{R}^d,$$

and take $c_1 = \frac{B_1^2}{dB_2^2 N}$ and $c_2 = \frac{dB_2^2}{\gamma^2}$.

$\square$

**Proposition 4.3** (Loss convergence). *Under Assumptions 2.1 and 2.2, there exists a positive constant $\eta > 0$ depending only on the dataset $\{\mathbf{x}_i\}_{i\in[N]}$, such that if the learning rate schedule satisfies $\eta_t \leq \eta$ and $\sum_{t=0}^{\infty} \eta_t = \infty$, then* AdamProxy *iterates minimize the loss, i.e., $\lim_{t\to\infty} \mathcal{L}(\mathbf{w}_t) = 0$.*

*Proof.* First, we start with the descent lemma for `AdamProxy`, following the standard techniques in the analysis of normalized steepest descent.

Let $D = \sup_{\mathbf{w} \in \mathbb{R}^d} \max_{i \in [N]} \|\mathbf{x}_i\|_{\mathbf{P}^{-1}(\mathbf{w})}$. Notice that $D \leq c_2 \max_{i \in [N]} \|\mathbf{x}_i\|_2 < \infty$ by Lemma F.1. Also, we define

$$\gamma_{\mathbf{w}} = \max_{\|\mathbf{v}\|_{\mathbf{P}(\mathbf{w})} \leq 1} \min_{i \in [N]} \mathbf{v}^\top \mathbf{x}_i$$

be the $\|\cdot\|_{\mathbf{P}(\mathbf{w})}$-max-margin. Also notice that $\bar{\gamma} \triangleq \sup_{\mathbf{w} \in \mathbb{R}^d} \gamma_{\mathbf{w}} < \infty$, since

$$\max_{\|\mathbf{v}\|_{\mathbf{P}(\mathbf{w})} \leq 1} \min_{i \in [N]} \mathbf{v}^\top \mathbf{x}_i \leq \max_{\|\mathbf{v}\|_2 \leq \frac{1}{c_1}} \min_{i \in [N]} \mathbf{v}^\top \mathbf{x}_i$$

for any $\mathbf{w} \in \mathbb{R}^d$ by Lemma F.1. Then, we get

$$\mathcal{L}(\mathbf{w}_{t+1}) = \mathcal{L}(\mathbf{w}_t) + \eta_t \langle \nabla \mathcal{L}(\mathbf{w}_t), \mathrm{Prx}(\mathbf{w}_t) \rangle + \frac{\eta_t^2}{2} \mathrm{Prx}(\mathbf{w}_t)^\top \nabla^2 \mathcal{L}(\mathbf{w}_t + \beta(\mathbf{w}_{t+1} - \mathbf{w}_t)) \mathrm{Prx}(\mathbf{w}_t)$$

$$\overset{(*)}{\leq} \mathcal{L}(\mathbf{w}_t) - \eta_t \|\nabla \mathcal{L}(\mathbf{w}_t)\|_{\mathbf{P}^{-1}(\mathbf{w}_t)} + \frac{\eta_t^2 D^2}{2} \sup\{\mathcal{G}(\mathbf{w}_t), \mathcal{G}(\mathbf{w}_{t+1})\}$$

$$\overset{(**)}{\leq} \mathcal{L}(\mathbf{w}_t) - \eta_t \|\nabla \mathcal{L}(\mathbf{w}_t)\|_{\mathbf{P}^{-1}(\mathbf{w}_t)} + \frac{\eta_t^2 D^2 e^{\eta_0 D}}{2} \mathcal{G}(\mathbf{w}_t)$$

$$\overset{(***)}{\leq} \mathcal{L}(\mathbf{w}_t) - \left(\eta_t - \frac{\eta_t^2 D^2 e^{\eta_0 D}}{2} \gamma_{\mathbf{w}_t}\right) \|\nabla \mathcal{L}(\mathbf{w}_t)\|_{\mathbf{P}^{-1}(\mathbf{w}_t)}$$

$$\leq \mathcal{L}(\mathbf{w}_t) - \frac{\eta_t}{2} \|\nabla \mathcal{L}(\mathbf{w}_t)\|_{\mathbf{P}^{-1}(\mathbf{w}_t)},$$

for $\eta_t \leq \frac{1}{\bar{\gamma} D^2 e^{\eta_0 D}} \triangleq \eta$. Note that $(*)$ is from

$$\mathrm{Prx}(\mathbf{w}_t)^\top \nabla^2 \mathcal{L}(\mathbf{w}) \mathrm{Prx}(\mathbf{w}_t) = \frac{1}{N} \sum_{i \in [N]} \ell''(\mathbf{w})(\mathrm{Prx}(\mathbf{w}_t)^\top \mathbf{x}_i)^2$$

$$\leq \frac{1}{N} \sum_{i \in [N]} \ell''(\mathbf{w}) \|\mathrm{Prx}(\mathbf{w}_t)\|_\infty^2 \|\mathbf{x}_i\|_1^2 \leq D^2 \mathcal{G}(\mathbf{w}),$$

where the last inequality is from Lemma I.1, and $(**), (***)$ are also from Lemma I.1. Telescoping this inequality, we get

$$\frac{1}{2} \sum_{t=t_0}^T \eta_t \|\nabla \mathcal{L}(\mathbf{w}_t)\|_{\mathbf{P}^{-1}(\mathbf{w}_t)} \leq \mathcal{L}(\mathbf{w}_{t_0}) - \mathcal{L}(\mathbf{w}_T) \leq \mathcal{L}(\mathbf{w}_{t_0}),$$

which implies $\sum_{t=t_0}^\infty \eta_t \|\nabla \mathcal{L}(\mathbf{w}_t)\|_{\mathbf{P}^{-1}(\mathbf{w}_t)} < \infty$. Since $\sum_{t=t_0}^T \eta_t = \infty$, we get $\liminf_{t \to \infty} \|\nabla \mathcal{L}(\mathbf{w}_t)\|_{\mathbf{P}^{-1}(\mathbf{w}_t)} = 0$. From Lemma F.1, we get $\liminf_{t \to \infty} \|\nabla \mathcal{L}(\mathbf{w}_t)\|_2 = 0$, also implying $\liminf_{t \to \infty} \mathcal{L}(\mathbf{w}_t) = 0$. Since $\mathcal{L}(\mathbf{w}_t)$ is monotonically decreasing, we get $\mathcal{L}(\mathbf{w}_t) \to 0$. $\square$

### F.3 PROOF OF LEMMA 4.5

**Intuition.** Before we provide a rigorous proof of Lemma 4.5, we first demonstrate its intuitive explanation motivated by Soudry et al. (2018). For simplicity, assume $\ell = \ell_{\exp}$ and let $\mathbf{w}_t = g(t)\hat{\mathbf{w}} + \boldsymbol{\rho}(t)$ where $g(t) = \|\mathbf{w}_t\|_2 \to \infty$, $\boldsymbol{\rho}(t) \in \mathbb{R}^d$, and $\frac{1}{g(t)}\boldsymbol{\rho}(t) \to \mathbf{0}$. Then, the mini-batch gradient can be represented by

$$\nabla \mathcal{L}_i(\mathbf{w}) = -\exp(-\mathbf{w}^\top \mathbf{x}_i)\mathbf{x}_i = -\exp(-g(t)\hat{\mathbf{w}}^\top \mathbf{x}_i)\exp(-\boldsymbol{\rho}(t)^\top \mathbf{x}_i)\mathbf{x}_i.$$

As $g(t) \to \infty$, the coefficient exponentially decays to 0. It implies that only terms with the smallest $\hat{\mathbf{w}}^\top \mathbf{x}_i$ will contribute to the update of `AdamProxy`. Therefore, the limit direction $\hat{\mathbf{w}}$ will be described by $\frac{\sum_{i \in [N]} c_i \mathbf{x}_i}{\sqrt{\sum_{i \in [N]} c_i^2 \mathbf{x}_i^2}}$ where $c_i$ is the contribution of the $i$-th sample to the update and it vanishes for $i \notin S$ where $S = \arg\min_{i \in [N]} \hat{\mathbf{w}}^\top \mathbf{x}_i$.

Building upon this intuition, we first establish the following technical lemma, characterizing limit points of a sequence in a form of `AdamProxy`.

**Lemma F.2.** *Let $(\boldsymbol{a}(t))_{t\geq 0}$ be a sequence of real vectors in $\mathbb{R}_{>0}^N$ and $\{\mathbf{x}_i\}_{i\in S} \subseteq \mathbb{R}^d$ be the dataset with nonzero entries for an index set $S \subseteq [N]$. Suppose that $\mathbf{b}_t = \frac{\sum_{i\in S} a_i(t)\mathbf{x}_i}{\sqrt{\sum_{i\in S} a_i(t)^2 \mathbf{x}_i^2}}$ satisfies $\|\mathbf{b}_t\|_2 \geq C > 0$ for all $t \geq 0$. Then every limit point of $\frac{\mathbf{b}_t}{\|\mathbf{b}_t\|_2}$ is positively proportional to $\frac{\sum_{i\in[N]} c_i\mathbf{x}_i}{\sqrt{\sum_{i\in[N]} c_i^2 \mathbf{x}_i^2}}$ for some $\mathbf{c} \in \Delta^{N-1}$ satisfying $c_i = 0$ for $i \notin S$.*

*Proof.* Define a function $F : \Delta^{|S|-1} \to \mathbb{R}^d$ as

$$F(\mathbf{d}) = \frac{\sum_{i\in S} d_i\mathbf{x}_i}{\sqrt{\sum_{i\in S} d_i^2 \mathbf{x}_i^2}}.$$

Since $\{\mathbf{x}_i\}_{i\in S}$ has nonzero entries, $F$ is continuous. Let $A = \{\mathbf{d} \in \Delta^{|S|-1} : \|F(\mathbf{d})\|_2 \geq C\}$. Since $F$ is continuous, $A$ is a closed subset of $\Delta^{|S|-1}$. Furthermore, since $\|\boldsymbol{\delta}_t\|_2 \geq C$ for all $t \geq 0$, $\{\boldsymbol{a}(t)\}_{t\geq 0} \subseteq A$.

Now let $\hat{\boldsymbol{\delta}}$ be a limit point of $\frac{\boldsymbol{\delta}_t}{\|\boldsymbol{\delta}_t\|_2}$. Define function $G : A \subseteq \Delta^{|S|-1} \to \mathbb{R}^d$ as

$$G(\mathbf{d}) = \frac{1}{\left\| \frac{\sum_{i\in S} d_i\mathbf{x}_i}{\sqrt{\sum_{i\in S} d_i^2 \mathbf{x}_i^2}} \right\|_2} \cdot \frac{\sum_{i\in S} d_i\mathbf{x}_i}{\sqrt{\sum_{i\in S} d_i^2 \mathbf{x}_i^2}}.$$

Notice that $G$ is continuous on $A$ and $\hat{\boldsymbol{\delta}} = \lim_{t\to\infty} G(\boldsymbol{a}(t))$. Since $A$ is bounded and closed, Bolzano-Weierstrass Theorem tells us that there exists a subsequence $\boldsymbol{a}(t_n)$ such that $\exists \lim_{n\to\infty} \boldsymbol{a}(t_n) = \mathbf{c} \in A$. Therefore, we get

$$\hat{\boldsymbol{\delta}} = \lim_{n\to\infty} G(\boldsymbol{a}(t_n)) = G(\lim_{n\to\infty} \boldsymbol{a}(t_n)) = G(\mathbf{c}).$$

Hence, the limit point $\hat{\boldsymbol{\delta}}$ is proportional to $\frac{\sum_{i\in S} c_i\mathbf{x}_i}{\sqrt{\sum_{i\in S} c_i^2 \mathbf{x}_i^2}}$. Then we regard $\mathbf{c} \in \Delta^{N-1}$ by taking $c_i = 0$ for $i \notin S$. $\qquad\square$

**Lemma 4.5.** *Under Assumptions 2.1, 2.2 and 4.4, there exists $\mathbf{c} = (c_0, \cdots, c_{N-1}) \in \Delta^{N-1}$ such that the limit direction $\hat{\mathbf{w}}$ of* `AdamProxy` *satisfies*

$$\hat{\mathbf{w}} \propto \frac{\sum_{i\in[N]} c_i\mathbf{x}_i}{\sqrt{\sum_{i\in[N]} c_i^2 \mathbf{x}_i^2}}, \tag{5}$$

*and $c_i = 0$ for $i \notin S$, where $S = \arg\min_{i\in[N]} \hat{\mathbf{w}}^\top \mathbf{x}_i$ is the index set of support vectors of $\hat{\mathbf{w}}$.*

*Proof.* We start with the case of $\ell = \ell_{\exp}$. First step is to characterize $\hat{\boldsymbol{\delta}}$, the limit direction of $\boldsymbol{\delta}_t$. To begin with, we introduce some new notations.

· From Assumption 4.4, let $\mathbf{w}_t = g(t)\hat{\mathbf{w}} + \boldsymbol{\rho}(t)$ where $g(t) = \|\mathbf{w}_t\|_2 \to \infty$, $\boldsymbol{\rho}(t) \in \mathbb{R}^d$, and $\frac{1}{g(t)}\boldsymbol{\rho}(t) \to \mathbf{0}$.

· Let $\gamma = \min_i \langle \mathbf{x}_i, \hat{\mathbf{w}} \rangle, \bar{\gamma}_i = \langle \mathbf{x}_i, \hat{\mathbf{w}} \rangle, \bar{\gamma} = \min_{i\notin S} \langle \mathbf{x}_i, \hat{\mathbf{w}} \rangle$. Then it satisfies $S = \{i \in [N] : \langle \mathbf{x}_i, \hat{\mathbf{w}} \rangle = \gamma\}$. Here, note that $\bar{\gamma} > \gamma > 0$.

· Let $\boldsymbol{\alpha}(t) \in \mathbb{R}^N$ be $\alpha_i(t) = \exp(-\boldsymbol{\rho}(t)^\top \mathbf{x}_i)$.

· Let $B_0 = \max_i \|\mathbf{x}_i\|_2, B_1 = \min_{i\in[N], j\in[d]} |\mathbf{x}_i[j]| > 0$, and $B_2 = \max_{i\in[N], j\in[d]} |\mathbf{x}_i[j]|$.

Since $\|\boldsymbol{\rho}(t)\|/g(t) \to 0$ and $\gamma, \bar{\gamma} > 0$, there exist $t_{\epsilon_1}, t_{\epsilon_2} > 0$ such that

$$\boldsymbol{\rho}(t)^\top \mathbf{x}_i \leq \|\boldsymbol{\rho}(t)\|_2 B_0 \leq \epsilon_1 \gamma g(t), \ \forall t > t_{\epsilon_1}, \forall i \in [N],$$
$$\boldsymbol{\rho}(t)^\top \mathbf{x}_i \geq -\|\boldsymbol{\rho}(t)\|_2 B_0 \geq -\epsilon_2 \bar{\gamma} g(t), \ \forall t > t_{\epsilon_2}, \forall i \in [N],$$

for all $\epsilon_1, \epsilon_2 > 0$. Then, we can decompose dominant and residual terms in the update rule.

$$\boldsymbol{\delta}_t = \frac{\sum_{i \in S} \exp(-\gamma g(t)) \exp(-\boldsymbol{\rho}(t)^\top \mathbf{x}_i) \mathbf{x}_i}{\sqrt{\sum_{i \in [N]} \exp(-2\bar{\gamma}_i g(t)) \exp(-2\boldsymbol{\rho}(t)^\top \mathbf{x}_i) \mathbf{x}_i^2}} + \frac{\sum_{i \in S^\complement} \exp(-\bar{\gamma}_i g(t)) \exp(-\boldsymbol{\rho}(t)^\top \mathbf{x}_i) \mathbf{x}_i}{\sqrt{\sum_{i \in [N]} \exp(-2\bar{\gamma}_i g(t)) \exp(-2\boldsymbol{\rho}(t)^\top \mathbf{x}_i) \mathbf{x}_i^2}}$$

$$\triangleq \mathbf{d}(t) + \mathbf{r}(t).$$

To investigate the limit direction of $\boldsymbol{\delta}_t$, we first show that $\mathbf{d}(t)$ dominates $\mathbf{r}(t)$, i.e., $\lim_{t \to \infty} \frac{\|\mathbf{r}(t)\|_2}{\|\mathbf{d}(t)\|_2} = 0$. Let $\mathbf{M}_t = \mathrm{diag}\left(\sqrt{\sum_{i \in [N]} \exp(-2\bar{\gamma}_i g(t)) \exp(-2\boldsymbol{\rho}(t)^\top \mathbf{x}_i) \mathbf{x}_i^2}\right)$. Notice that

$$\|\mathbf{M}_t \hat{\mathbf{w}}\|_2 \|\mathbf{d}(t)\|_2 \geq \langle \mathbf{M}_t \hat{\mathbf{w}}, \mathbf{d}(t) \rangle = \gamma \sum_{i \in S} \exp(-\gamma g(t)) \exp(-\boldsymbol{\rho}(t)^\top \mathbf{x}_i).$$

Since the diagonals of $\mathbf{M}_t$ are upper bounded by $B_2 \sqrt{\sum_{i \in [N]} \exp(-2\bar{\gamma}_i g(t)) \exp(-2\boldsymbol{\rho}(t)^\top \mathbf{x}_i)}$, we get

$$\|\mathbf{d}(t)\|_2 \geq \frac{\gamma \sum_{i \in S} \exp(-\gamma g(t)) \exp(-\boldsymbol{\rho}(t)^\top \mathbf{x}_i)}{B_2 \sqrt{\sum_{i \in [N]} \exp(-2\bar{\gamma}_i g(t)) \exp(-2\boldsymbol{\rho}(t)^\top \mathbf{x}_i)}}.$$

Also, notice that

$$\|\mathbf{r}(t)\|_2 \leq \frac{B_2 \sum_{i \in S} \exp(-\gamma g(t)) \exp(-\boldsymbol{\rho}(t)^\top \mathbf{x}_i)}{B_1 \sqrt{\sum_{i \in [N]} \exp(-2\bar{\gamma}_i g(t)) \exp(-2\boldsymbol{\rho}(t)^\top \mathbf{x}_i)}}.$$

From the following inequalities

$$\sum_{i \in S} \exp(-\gamma g(t)) \exp(-\boldsymbol{\rho}(t)^\top \mathbf{x}_i) \geq \exp(-\gamma g(t)) \exp(-\epsilon_1 \gamma g(t))$$

$$= \exp(-(1 + \epsilon_1) \gamma g(t)),$$

$$\sum_{i \in S^\complement} \exp(-\bar{\gamma}_i g(t)) \exp(-\boldsymbol{\rho}(t)^\top \mathbf{x}_i) \leq N \exp(-\bar{\gamma} g(t)) \exp(\epsilon_2 \bar{\gamma} g(t))$$

$$= N \exp(-(1 - \epsilon_2) \bar{\gamma} g(t)),$$

we conclude that

$$\frac{\|\mathbf{r}(t)\|_2}{\|\mathbf{d}(t)\|_2} = \frac{B_2^2}{\gamma B_1} \frac{\sum_{i \in S^\complement} \exp(-\gamma g(t)) \exp(-\boldsymbol{\rho}(t)^\top \mathbf{x}_i)}{\sum_{i \in S} \exp(-\gamma g(t)) \exp(-\boldsymbol{\rho}(t)^\top \mathbf{x}_i)}$$

$$\leq \frac{N B_2^2}{\gamma B_1} \exp(-\frac{1}{2}(\bar{\gamma} - \gamma) g(t)) \to 0.$$

Next, we claim that every limit point of $\frac{\mathbf{d}(t)}{\|\mathbf{d}(t)\|_2}$ is positively proportional to $\frac{\sum_{i \in [N]} c_i \mathbf{x}_i}{\sqrt{\sum_{i \in [N]} c_i^2 \mathbf{x}_i^2}}$ for some $\mathbf{c} = (c_0, \cdots, c_{N-1}) \in \Delta^{N-1}$ satisfying $c_i = 0$ for $i \notin S$. Notice that

$$\mathbf{d}(t)[k] = \frac{\sum_{i \in S} \exp(-\gamma g(t)) \exp(-\boldsymbol{\rho}(t)^\top \mathbf{x}_i) \mathbf{x}_i[k]}{\sqrt{\sum_{i \in [N]} \exp(-2\bar{\gamma}_i g(t)) \exp(-2\boldsymbol{\rho}(t)^\top \mathbf{x}_i) \mathbf{x}_i^2[k]}}$$

$$= \frac{\sum_{i \in S} \exp(-\gamma g(t)) \exp(-\boldsymbol{\rho}(t)^\top \mathbf{x}_i) \mathbf{x}_i[k]}{\sqrt{\sum_{i \in S} \exp(-2\gamma g(t)) \exp(-2\boldsymbol{\rho}(t)^\top \mathbf{x}_i) \mathbf{x}_i^2[k] + \sum_{i \in S^\complement} \exp(-2\bar{\gamma}_i g(t)) \exp(-2\boldsymbol{\rho}(t)^\top \mathbf{x}_i) \mathbf{x}_i^2[k]}}$$

$$= \frac{\sum_{i \in S} \exp(-\gamma g(t)) \exp(-\boldsymbol{\rho}(t)^\top \mathbf{x}_i) \mathbf{x}_i[k]}{\sqrt{\sum_{i \in S} \exp(-2\gamma g(t)) \exp(-2\boldsymbol{\rho}(t)^\top \mathbf{x}_i) \mathbf{x}_i^2[k]}} \frac{1}{\sqrt{1 + \frac{\sum_{i \in S^\complement} \exp(-2\bar{\gamma}_i g(t)) \exp(-2\boldsymbol{\rho}(t)^\top \mathbf{x}_i) \mathbf{x}_i^2[k]}{\sum_{i \in S} \exp(-2\gamma g(t)) \exp(-2\boldsymbol{\rho}(t)^\top \mathbf{x}_i) \mathbf{x}_i^2[k]}}}.$$

Let $\mathbf{b}_t = \frac{\sum_{i \in S} \exp(-\gamma g(t)) \exp(-\boldsymbol{\rho}(t)^\top \mathbf{x}_i) \mathbf{x}_i}{\sqrt{\sum_{i \in S} \exp(-2\gamma g(t)) \exp(-2\boldsymbol{\rho}(t)^\top \mathbf{x}_i) \mathbf{x}_i^2}} = \frac{\sum_{i \in S} \exp(-\boldsymbol{\rho}(t)^\top \mathbf{x}_i) \mathbf{x}_i}{\sqrt{\sum_{i \in S} \exp(-2\boldsymbol{\rho}(t)^\top \mathbf{x}_i) \mathbf{x}_i^2}}$. Since

$$\frac{\sum_{i \in S^\complement} \exp(-2\bar{\gamma}_i g(t)) \exp(-2\boldsymbol{\rho}(t)^\top \mathbf{x}_i) \mathbf{x}_i^2[k]}{\sum_{i \in S} \exp(-2\gamma g(t)) \exp(-2\boldsymbol{\rho}(t)^\top \mathbf{x}_i) \mathbf{x}_i^2[k]} \to 0,$$

every limit point of $\frac{\mathbf{d}(t)}{\|\mathbf{d}(t)\|_2}$ is represented by a limit point of $\frac{\mathbf{b}_t}{\|\mathbf{b}_t\|_2}$. Notice that $\mathbf{b}_t$ is an update of AdamProxy under the dataset $\{\mathbf{x}_i\}_{i \in S}$, which implies $\|\mathbf{b}_t\|_2$ is lower bounded by a positive constant from Lemma F.1. Therefore, Lemma F.2 proves the claim.

Hence, we can characterize $\hat{\boldsymbol{\delta}}$ as

$$\hat{\boldsymbol{\delta}} = \lim_{t \to \infty} \frac{\boldsymbol{\delta}_t}{\|\boldsymbol{\delta}_t\|_2} = \lim_{t \to \infty} \frac{\mathbf{d}(t) + \mathbf{r}(t)}{\|\mathbf{d}(t) + \mathbf{r}(t)\|_2}$$

$$= \lim_{t \to \infty} \frac{\mathbf{d}(t)}{\|\mathbf{d}(t) + \mathbf{r}(t)\|_2} + \lim_{t \to \infty} \frac{\mathbf{r}(t)}{\|\mathbf{d}(t) + \mathbf{r}(t)\|_2}$$

$$= \lim_{t \to \infty} \frac{\mathbf{d}(t)}{\|\mathbf{d}(t)\|_2} \propto \frac{\sum_{i \in [N]} c_i \mathbf{x}_i}{\sqrt{\sum_{i \in [N]} c_i^2 \mathbf{x}_i^2}},$$

for some $\mathbf{c} \in \Delta^{N-1}$ satisfying $c_i = 0$ for $i \notin S$.

Second step is to connect the limiting behavior of $\boldsymbol{\delta}_t$ to the limit direction $\hat{\mathbf{w}}$ using Stolz-Cesaro theorem. From the first step, we can represent

$$\boldsymbol{\delta}_t = h(t)\hat{\boldsymbol{\delta}} + \boldsymbol{\sigma}(t),$$

where $h(t) = \|\boldsymbol{\delta}_t\|_2$ and $\frac{1}{h(t)}\boldsymbol{\sigma}(t) \to 0$. Notice that $\mathbf{w}_t - \mathbf{w}_0 = \sum_{s=0}^{t-1} \eta_s h(s)(\hat{\boldsymbol{\delta}} + \frac{1}{h(s)}\boldsymbol{\sigma}(t))$. Since $\hat{\boldsymbol{\delta}} + \frac{1}{h(s)}\boldsymbol{\sigma}(t)$ is bounded, we get $\sum_{s=0}^{t-1} \eta_s h(s) \to \infty$. Then we take

$$\boldsymbol{a}_t = \mathbf{w}_t - \mathbf{w}_0 = \sum_{s=0}^{t-1} \eta_s h(s)(\hat{\boldsymbol{\delta}} + \frac{1}{h(s)}\boldsymbol{\sigma}(t))$$

$$b_t = \sum_{s=0}^{t-1} \eta_s h(s).$$

Then, $\{b_t\}_{t=1}^\infty$ is strictly monotone and diverging. Also, $\lim_{t \to \infty} \frac{\boldsymbol{a}_{t+1} - \boldsymbol{a}_t}{b_{t+1} - b_t} = \hat{\boldsymbol{\delta}}$. Then, by Stolz-Cesaro theorem, we get

$$\lim_{t \to \infty} \frac{\boldsymbol{a}_t}{b_t} = \hat{\boldsymbol{\delta}}.$$

This implies $\mathbf{w}_t = b_t \hat{\boldsymbol{\delta}} + \boldsymbol{\tau}(t)$ where $\frac{\boldsymbol{\tau}(t)}{b_t} \to 0$. Also notice that $\mathbf{w}_t = g(t)\hat{\mathbf{w}} + \boldsymbol{\rho}(t)$. Dividing by $g(t)$, we get

$$\hat{\mathbf{w}} = \lim_{t \to \infty} \frac{g(t)\hat{\mathbf{w}} + \boldsymbol{\rho}(t)}{g(t)} = \lim_{t \to \infty} \frac{b_t}{g(t)}\left(\hat{\boldsymbol{\delta}} + \frac{\boldsymbol{\tau}(t)}{b_t}\right).$$

Since $\ell_2$ norm is continuous, we get

$$1 = \|\hat{\mathbf{w}}\|_2 = \lim_{t \to \infty} \frac{b_t}{g(t)}\left\|\hat{\boldsymbol{\delta}} + \frac{\boldsymbol{\tau}(t)}{b_t}\right\|_2 = \lim_{t \to \infty} \frac{b_t}{g(t)},$$

which implies $\hat{\mathbf{w}} = \hat{\boldsymbol{\delta}}$.

Then we move on to the case of $\ell = \ell_{\log}$. This kind of extension is possible since the logistic loss has a similar tail behavior of the exponential loss, following the line of Soudry et al. (2018). We adopt the same notation with previous part, and we decompose dominant and residual terms as follows:

$$\boldsymbol{\delta}_t = \frac{\sum_{i \in S} |\ell'(\gamma g(t) + \boldsymbol{\rho}(t)^\top \mathbf{x}_i)| \mathbf{x}_i}{\sqrt{\sum_{i \in [N]} |\ell'(\bar{\gamma}_i g(t) + \boldsymbol{\rho}(t)^\top \mathbf{x}_i)|^2 \mathbf{x}_i^2}} + \frac{\sum_{i \in S^\complement} |\ell'(\bar{\gamma}_i g(t) + \boldsymbol{\rho}(t)^\top \mathbf{x}_i)| \mathbf{x}_i}{\sqrt{\sum_{i \in [N]} |\ell'(\bar{\gamma}_i g(t) + \boldsymbol{\rho}(t)^\top \mathbf{x}_i)|^2 \mathbf{x}_i^2}}$$

$$\triangleq \mathbf{d}(t) + \mathbf{r}(t).$$

Notice that $\lim_{z\to\infty} \frac{|\ell'_{\log}(z)|}{|\ell'_{\exp}(z)|} = \lim_{z\to\infty} \frac{1}{1+e^{-z}} = 1$. Therefore, the limit behavior of $\mathbf{d}(t)$ and $\mathbf{r}(t)$ is identical to the previous $\ell = \ell_{\exp}$ case. This implies the same proof also holds for the logistic loss, which ends the proof. $\qquad\square$

### F.4  PROOF OF THEOREM 4.8

**Theorem 4.8.** *Under Assumptions 2.1 and 4.7, $P_{Adam}(\mathbf{c})$ admits unique primal and dual solutions, so that $\mathbf{p}(\mathbf{c})$ and $\mathbf{d}(\mathbf{c})$ can be regarded as vector-valued functions. Moreover, under Assumptions 2.1, 2.2, 4.4 and 4.7, the following hold:*

(a) *$\mathbf{p} : \Delta^{N-1} \to \mathbb{R}^d$ is continuous.*

(b) *$\mathbf{d} : \Delta^{N-1} \to \mathbb{R}^N_{\geq 0}\backslash\{\mathbf{0}\}$ is continuous. Consequently, the map $T(\mathbf{c}) \triangleq \frac{\mathbf{d}(\mathbf{c})}{\|\mathbf{d}(\mathbf{c})\|_1}$ is continuous.*

(c) *The map $T : \Delta^{N-1} \to \Delta^{N-1}$ admits at least one fixed point.*

(d) *There exists $\mathbf{c}^* \in \{\mathbf{c} \in \Delta^{N-1} : T(\mathbf{c}) = \mathbf{c}\}$ such that the convergence direction $\hat{\mathbf{w}}$ of* AdamProxy *is proportional to $\mathbf{p}(\mathbf{c}^*)$.*

*Proof.* We first show that $P_{Adam}(\mathbf{c})$ has a unique solution and $\mathbf{p}(\mathbf{c})$ can be identified as a vector-valued function. Since $\mathbf{M}(\mathbf{c})$ is positive definite for every $\mathbf{c} \in \Delta^{N-1}$, $\frac{1}{2}\|\mathbf{w}\|_{\mathbf{M}(\mathbf{c})}$ is strictly convex. Since the feasible set is convex, there exists a unique optimal solution of $P_{Adam}(\mathbf{c})$ and we can redefine $\mathbf{p}(\mathbf{c})$ as a vector-valued function.

Since the inequality constraints are linear, $P_{Adam}(\mathbf{c})$ satisfies Slater's condition, which implies that there exists a dual solution. From Assumption 4.7, such dual solution is unique.

(a) Let $f(\mathbf{w}, \mathbf{c}) = \frac{1}{2}\|\mathbf{w}\|_{\mathbf{M}(\mathbf{c})}$ be the objective function of $P_{Adam}(\mathbf{c})$ and $F = \{\mathbf{w} \in \mathbb{R}^d : \mathbf{w}^\top\mathbf{x}_i - 1 \geq 0, \forall i \in [N]\}$ be the feasible set. It is clear that such $f$ is continuous on $\mathbf{w}$ and $\mathbf{c}$. Let $\bar{\mathbf{c}} \in \Delta^{N-1}$ and assume $\mathbf{p}$ is not continuous on $\bar{\mathbf{c}}$. Then there exists $\{\mathbf{c}_k\} \subset \Delta^{N-1}$ such that $\lim_{k\to\infty} \mathbf{c}_k = \bar{\mathbf{c}}$ but $\|\mathbf{p}(\mathbf{c}_k) - \mathbf{p}(\bar{\mathbf{c}})\|_2 \geq \epsilon$ for some $\epsilon > 0$. We denote $\mathbf{w}_k = \mathbf{p}(\mathbf{c}_k)$ and $\bar{\mathbf{w}} = \mathbf{p}(\bar{\mathbf{c}})$.

First, construct $\{\mathbf{u}_k\} \subset F$ such that $\lim_{k\to\infty} \mathbf{u}_k = \bar{\mathbf{w}}$. Then we get a natural relationship between $\mathbf{w}_k$ and $\mathbf{u}_k$ as

$$\frac{1}{2}\mathbf{w}_k^\top \mathbf{M}(\mathbf{c}_k)\mathbf{w}_k \leq \frac{1}{2}\mathbf{u}_k^\top \mathbf{M}(\mathbf{c}_k)\mathbf{u}_k.$$

Second, consider the case when $\{\mathbf{w}_k\}$ is bounded. Then we can take a subsequence $\mathbf{w}_{k_n} \to \mathbf{w}_0$. Since $\{\mathbf{w}_{k_n}\} \subset F$ and $F$ is closed, we get $\mathbf{w}_0 \in F$. Also, since $f$ is continuous, $f(\mathbf{w}_{k_n}, \mathbf{c}_{k_n}) \to f(\mathbf{w}_0, \bar{\mathbf{c}})$. Therefore,

$$f(\mathbf{w}_{k_n}, \mathbf{c}_{k_n}) \leq f(\bar{\mathbf{w}}, \mathbf{c}_{k_n}) \xrightarrow[n\to\infty]{} f(\mathbf{w}_0, \bar{\mathbf{c}}) \leq f(\bar{\mathbf{w}}, \bar{\mathbf{c}}),$$

which implies $\mathbf{w}_0 = \bar{\mathbf{w}}$. This makes a contradiction to $\|\mathbf{p}(\mathbf{c}_k) - \mathbf{p}(\bar{\mathbf{c}})\|_2 = \|\mathbf{w}_k - \bar{\mathbf{w}}\|_2 \geq \epsilon$.

Lastly, consider the case when $\{\mathbf{w}_k\}$ is not bounded. By taking a subsequence, we can assume that $\|\mathbf{w}_k\|_2 \to \infty$ without loss of generality. Define $\mathbf{v}_k = \frac{\mathbf{w}_k}{\|\mathbf{w}_k\|_2}$. Since $\mathbf{v}_k$ is bounded, we can take a convergent subsequence and consider $\lim_{k\to\infty} \mathbf{v}_k = \bar{\mathbf{v}}$ without loss of generality. Then,

$$\frac{1}{2}\mathbf{w}_k^\top \mathbf{M}(\mathbf{c}_k)\mathbf{w}_k \leq \frac{1}{2}\mathbf{u}_k^\top \mathbf{M}(\mathbf{c}_k)\mathbf{u}_k \Rightarrow \frac{1}{2}\mathbf{v}_k^\top \mathbf{M}(\mathbf{c}_k)\mathbf{v}_k \leq \frac{1}{2}\left(\frac{\mathbf{u}_k}{\|\mathbf{w}_k\|_2}\right)^\top \mathbf{M}(\mathbf{c}_k)\left(\frac{\mathbf{u}_k}{\|\mathbf{w}_k\|_2}\right).$$

Since $f$ is continuous and $\{\mathbf{u}_k\}$ is bounded, we get

$$\frac{1}{2}\bar{\mathbf{v}}^\top \mathbf{M}(\bar{\mathbf{c}})\bar{\mathbf{v}} = f(\bar{\mathbf{v}}, \bar{\mathbf{c}}) = \lim_{k\to\infty} f(\mathbf{v}_k, \mathbf{c}_k) = \lim_{k\to\infty} \frac{1}{2}\mathbf{v}_k^\top \mathbf{M}(\mathbf{c}_k)\mathbf{v}_k$$

$$\leq \limsup_{k\to\infty} \frac{1}{2}\left(\frac{\mathbf{u}_k}{\|\mathbf{w}_k\|}\right)^\top \mathbf{M}(\mathbf{c}_k)\left(\frac{\mathbf{u}_k}{\|\mathbf{w}_k\|}\right) = 0.$$

Note that $\mathbf{M}(\bar{\mathbf{c}})$ is positive definite and $\frac{1}{2}\bar{\mathbf{v}}^\top \mathbf{M}(\bar{\mathbf{c}})\bar{\mathbf{v}} = 0$ implies $\bar{\mathbf{v}} = 0$, which makes a contradiction.

(b) Let $\mathbf{c}_0 \in \Delta^{N-1}$ be given and take $\mathbf{w}^* = \mathbf{p}(\mathbf{c}_0)$. From KKT conditions of $P_{\text{Adam}}(\mathbf{c}_0)$, the dual solution $\mathbf{d}(\mathbf{c}_0)$ is given by

$$\mathbf{M}(\mathbf{c}_0)\mathbf{w}^* = \sum_{i \in S(\mathbf{w}^*)} d_i(\mathbf{c}_0)\mathbf{x}_i$$

and such $d_i(\mathbf{c}_0) \geq 0$ is uniquely determined since $\{\mathbf{x}_i\}_{i \in S(\mathbf{w}^*)}$ is a set of linearly independent vectors by Assumption 4.7.

Now we claim that $\mathbf{d}(\mathbf{c})$ is continuous at $\mathbf{c} = \mathbf{c}_0$. Notice that $\min_{i \notin S(\mathbf{w}^*)} \mathbf{w}^{*\top}\mathbf{x}_i > 1$. Since $\mathbf{p}$ is continuous at $\mathbf{c}_0$, there exists $\delta > 0$ such that $\mathbf{p}(\mathbf{c})^\top \mathbf{x}_i - 1 > 0$ for $i \notin S(\mathbf{w}^*)$ and $\mathbf{c} \in \Delta^{N-1} \cap B_\delta(\mathbf{c}_0)$. Therefore, $S(\mathbf{p}(\mathbf{c})) \subseteq S(\mathbf{w}^*)$ on $\mathbf{c} \in \Delta^{N-1} \cap B_\delta(\mathbf{c}_0)$.

Let $\mathbf{X}$ be a matrix whose columns are the support vectors of $\mathbf{w}^*$. On $\mathbf{c} \in \Delta^{N-1} \cap B_\delta(\mathbf{c}_0)$, KKT conditions tells us that

$$\mathbf{M}(\mathbf{c})\mathbf{p}(\mathbf{c}) = \sum_{i \in S(\mathbf{p}(\mathbf{c}))} d_i(\mathbf{c})\mathbf{x}_i \overset{(*)}{=} \sum_{i \in S(\mathbf{w}^*)} d_i(\mathbf{c})\mathbf{x}_i = \mathbf{X}\mathbf{d}(\mathbf{c})$$

$$\overset{(**)}{\Leftrightarrow} \mathbf{d}(\mathbf{c}) = (\mathbf{X}^\top|_{\text{im}\,\mathbf{X}^\top})^{-1}\mathbf{M}(\mathbf{c})\mathbf{p}(\mathbf{c}),$$

where $(*)$ is from $S(\mathbf{p}(\mathbf{c})) \subseteq S(\mathbf{w}^*)$ and $(**)$ is from the linear independence of columns of $\mathbf{X}$. Notice that $\mathbf{M}(\mathbf{c})$ and $\mathbf{w}^*(\mathbf{c})$ are continuous on $\mathbf{c} = \mathbf{c}_0$, which implies that $\mathbf{d}(\mathbf{c})$ is continuous on $\mathbf{c} = \mathbf{c}_0$.

Since at least one of the dual solutions is strictly positive, $\mathbf{d}$ is a continuous map from $\Delta^{N-1}$ to $\mathbb{R}^N_{\geq 0}\backslash\{\mathbf{0}\}$. This implies that $T$ is continuous, since $\mathbf{d} \mapsto \frac{\mathbf{d}}{\sum_{i \in [N]} d_i}$ is continuous on $\mathbb{R}^N_{\geq 0}\backslash\{\mathbf{0}\}$.

(c) Since $\Delta^{N-1}$ is a nonempty convex compact subset of $\mathbb{R}^N$, there exists a fixed point of $T$ by Brouwer fixed-point theorem.

(d) From Lemma 4.5, there exists $\mathbf{c}^* \in \Delta^{N-1}$ such that $\hat{\mathbf{w}} \propto \frac{\sum_{i=1}^N c_i^* \mathbf{x}_i}{\sqrt{\sum_{i=1}^N c_i^{*2}\mathbf{x}_i^2}}$ with $c_i^* = 0$ for $i \notin S'$ where $S' = \arg\min_{i \in [N]} \hat{\mathbf{w}}^\top \mathbf{x}_i$. Then we take $\hat{\mathbf{w}} = \frac{\sum_{i \in S} kc_i^* \mathbf{x}_i}{\sqrt{\sum_{i \in S} c_i^{*2}\mathbf{x}_i^2}}$ for some $k > 0$. We claim that such $\mathbf{c}^*$ becomes a fixed point of $T$ and $\hat{\mathbf{w}} \propto \mathbf{p}(\mathbf{c}^*)$.

Consider the optimization problem $P_{\text{Adam}}(\mathbf{c}^*)$ and its unique primal solution $\mathbf{w}^* = \mathbf{p}(\mathbf{c}^*)$. Notice that $\min_{i \in [N]} \hat{\mathbf{w}}^\top \mathbf{x}_i = \gamma > 0$ since `AdamProxy` minimizes the loss. Therefore, $\mathbf{w}^* = \frac{1}{\gamma}\hat{\mathbf{w}}$ and $d_i(\mathbf{c}^*) = \frac{kc_i^*}{\gamma}$ satisfy the following KKT conditions

$$\mathbf{M}(\mathbf{c}^*)\mathbf{w}^* = \sum_{i \in S^*} d_i \mathbf{x}_i, d_i \geq 0,$$

$$\mathbf{w}^{*\top}\mathbf{x}_i - 1 \geq 0, \forall i \in [N],$$

where $S^* = \{i \in [N] : \mathbf{w}^{*\top}\mathbf{x}_i - 1 = 0\}$ is the index set of support vectors of $\mathbf{w}^*$. This implies that $T(\mathbf{c}^*) = \mathbf{c}^*$ and $\hat{\mathbf{w}} = \gamma\mathbf{w}^* \propto \mathbf{w}^* = \mathbf{p}(\mathbf{c}^*)$, which proves the claim.

$\square$

## F.5 DETAILED CALCULATIONS OF EXAMPLE 4.11

Consider $N = d$ and $\{\mathbf{x}_i\}_{i \in [d]} \subseteq \mathbb{R}^d$ where $\mathbf{x}_i = x_i \mathbf{e}_i + \delta \sum_{j \neq i} \mathbf{e}_j$ for some $0 < \delta$ and $0 < x_0 < \cdots < x_{d-1}$. $\ell_\infty$-max-margin problem is given by

$$\min \|\mathbf{w}\|_\infty \text{ subject to } \mathbf{w}^\top \mathbf{x}_i \geq 1, \forall i \in [N].$$

(For the convenience of calculation, we use the objective $\|\mathbf{w}\|_\infty$ rather than $\frac{1}{2}\|\mathbf{w}\|_\infty^2$.) Its KKT conditions are given by

$$\partial\|\mathbf{w}\|_\infty \ni \sum_{i\in[N]} \lambda_i \mathbf{x}_i,$$

$$\sum_{i\in[N]} \lambda_i(\mathbf{w}^\top\mathbf{x}_i - 1) = 0,$$

$$\lambda_i \geq 0, \ \mathbf{w}^\top\mathbf{x}_i - 1 \geq 0, \forall i \in [N].$$

Note that $\mathbf{w}^* = \left(\frac{1}{x_0+(d-1)\delta}, \cdots, \frac{1}{x_0+(d-1)\delta}\right) \in \mathbb{R}^d$ and $\boldsymbol{\lambda}^* = \left(\frac{1}{x_0+(d-1)\delta}, 0, \cdots, 0\right) \in \mathbb{R}^d$ satisfy the KKT conditions since

$$\partial\|\mathbf{w}\|_\infty\Big|_{\mathbf{w}=\mathbf{w}^*} = \Delta^{d-1} \ni \frac{1}{x_0+(d-1)\delta}\mathbf{x}_0 = \sum_{i\in[N]} \lambda_i^*\mathbf{x}_i,$$

$$\sum_{i\in[N]} \lambda_i^*(\mathbf{w}^{*\top}\mathbf{x}_i - 1) = \lambda_1^*\left(\frac{x_0+(d-1)\delta}{x_0+(d-1)\delta} - 1\right) = 0,$$

$$\lambda_i^* \geq 0, \mathbf{w}^{*\top}\mathbf{x}_i - 1 \geq 0, \forall i \in [N].$$

Now we show that $\mathbf{c}^* = (1, 0, \cdots, 0) \in \Delta^{d-1}$ is a fixed point of $T$ in Theorem 4.8 and $\mathbf{w}^* = \mathbf{p}(\mathbf{c}^*)$. Note that for $k = \frac{1}{x_0+(d-1)\delta} > 0$, it satisfies

$$\mathbf{M}(\mathbf{c}^*)\mathbf{w}^* = \mathrm{diag}(x_0, \delta, \cdots, \delta)\mathbf{w}^* = k\mathbf{x}_0 = k\sum_{i\in[N]} c_i^*\mathbf{x}_i$$

$$\sum_{i\in[N]} c_i^*(\mathbf{w}^{*\top}\mathbf{x}_i - 1) = 0,$$

$$c_i^* \geq 0, \mathbf{w}^{*\top}\mathbf{x}_i - 1 \geq 0, \forall i \in [N],$$

which implies $T(\mathbf{c}^*) = \mathbf{c}^*$ and $\mathbf{w}^* = \mathbf{p}(\mathbf{c}^*)$.

## G   MISSING PROOFS IN SECTION 5

---

**Algorithm 4** `Inc-Signum`

---

**Hyperparams:** Learning rate schedule $\{\eta_t\}_{t=0}^{T-1}$, momentum parameter $\beta \in [0,1)$, batch size $b$
**Input:** Initial weight $\mathbf{w}_0$, dataset $\{\mathbf{x}_i\}_{i\in[N]}$
 1: Initialize momentum $\mathbf{m}_{-1} = \mathbf{0}$
 2: **for** $t = 0, 1, 2, \ldots, T-1$ **do**
 3:    $\mathcal{B}_t \leftarrow \{(t \cdot b + i) \pmod N\}_{i=0}^{b-1}$
 4:    $\mathbf{g}_t \leftarrow \nabla\mathcal{L}_{\mathcal{B}_t}(\mathbf{w}_t) = \frac{1}{b}\sum_{i\in\mathcal{B}_t}\ell'(\mathbf{w}_t^\top\mathbf{x}_i)\mathbf{x}_i$
 5:    $\mathbf{m}_t \leftarrow \beta\mathbf{m}_{t-1} + (1-\beta)\mathbf{g}_t$
 6:    $\mathbf{w}_{t+1} \leftarrow \mathbf{w}_t - \eta_t\,\mathrm{sign}(\mathbf{m}_t)$
 7: **end for**
 8: **return** $\mathbf{w}_T$

---

**Related Work.**   Our proof of Theorem 5.1 builds on standard techniques from the analysis of the implicit bias of normalized steepest descent on linearly separable data (Gunasekar et al., 2018a; Zhang et al., 2024a; Fan et al., 2025). The most closely related result is due to Fan et al. (2025), who showed that full-batch Signum converges in direction to the maximum $\ell_\infty$-margin solution. Theorem 5.1 extends this result to the mini-batch setting, establishing that the mini-batch variant of `Inc-Signum` (Algorithm 4) also converges in direction to the maximum $\ell_\infty$-margin solution, provided the momentum parameter is chosen sufficiently close to 1.

**Technical Contribution.** The key technical contribution enabling the mini-batch analysis is Lemma G.2. Importantly, requiring momentum parameter $\beta$ close to 1 is not merely a technical convenience but intrinsic to the mini-batch setting ($b < N$), as formalized in Lemma G.2 and supported empirically in Figure 10 of Appendix B.

**Implicit Bias of SignSGD.** We note that as an extreme case, `Inc-Signum` with $\beta = 0$ and a batch size of 1 (i.e., SignSGD) has a simple implicit bias: its iterates converge in direction to $\sum_{i \in [N]} \text{sign}(\mathbf{x}_i)$, which corresponds to neither the $\ell_2$- nor the $\ell_\infty$-max-margin solution.

**Notation.** We introduce additional notation to analyze `Inc-Signum` (Algorithm 4) with arbitrary mini-batch size $b$. Let $\mathcal{B}_t \subseteq [N]$ denote the set of indices in the mini-batch sampled at iteration $t$. The corresponding mini-batch loss $\mathcal{L}_{\mathcal{B}_t}(\mathbf{w})$ is defined as

$$\mathcal{L}_{\mathcal{B}_t}(\mathbf{w}) \triangleq \frac{1}{|\mathcal{B}_t|} \sum_{i \in \mathcal{B}_t} \ell(\mathbf{w}^\top \mathbf{x}_i).$$

We define the maximum normalized $\ell_\infty$-margin as

$$\gamma_\infty \triangleq \max_{\|\mathbf{w}\|_\infty \leq 1} \min_{i \in [N]} \mathbf{w}^\top \mathbf{x}_i > 0,$$

and again introduce the proxy $\mathcal{G} : \mathbb{R}^d \to \mathbb{R}$ defined as

$$\mathcal{G}(\mathbf{w}) \triangleq -\frac{1}{N} \sum_{i \in [N]} \ell'(\mathbf{w}^\top \mathbf{x}_i).$$

As before, we consider $\ell$ to be either the logistic loss $\ell_{\log}(z) = \log(1 + \exp(-z))$ or the exponential loss $\ell_{\exp}(z) = \exp(-z)$. Finally, let $D$ be an upper bound on the $\ell_1$-norm of the data, i.e., $\|\mathbf{x}_i\|_1 \leq D$ for all $i \in [N]$.

**Lemma G.1** (Descent inequality). `Inc-Signum` *iterates* $\{\mathbf{w}_t\}$ *satisfy*

$$\mathcal{L}(\mathbf{w}_{t+1}) \leq \mathcal{L}(\mathbf{w}_t) - \eta_t \langle \nabla \mathcal{L}(\mathbf{w}_t), \Delta_t \rangle + C_H \eta_t^2 \mathcal{G}(\mathbf{w}_t), \quad \Delta_t := \text{sign}(\mathbf{m}_t),$$

*where* $C_H = \frac{1}{2} D^2 e^{\eta_0 D}$.

*Proof.* By Taylor's theorem,

$$\mathcal{L}(\mathbf{w}_{t+1}) = \mathcal{L}(\mathbf{w}_t - \eta_t \Delta_t) = \mathcal{L}(\mathbf{w}_t) - \eta_t \langle \nabla \mathcal{L}(\mathbf{w}_t), \Delta_t \rangle + \frac{1}{2} \eta_t^2 \Delta_t^\top \nabla^2 \mathcal{L}(\mathbf{w}_t - \zeta \eta_t \Delta_t) \Delta_t,$$

for some $\zeta \in (0, 1)$. Note that for any $\mathbf{w} \in \mathbb{R}^d$,

$$\Delta_t^\top \nabla^2 \mathcal{L}(\mathbf{w}) \Delta_t = \frac{1}{N} \sum_{i \in [N]} \ell''(\mathbf{w}^\top \mathbf{x}_i)(\Delta_t^\top \mathbf{x}_i)^2 \leq \frac{1}{N} \sum_{i \in [N]} \ell''(\mathbf{w}^\top \mathbf{x}_i) \|\Delta_t\|_\infty^2 \|\mathbf{x}_i\|_1^2 \leq D^2 \mathcal{G}(\mathbf{w}),$$

where we used $\mathcal{G}(\mathbf{w}) \geq \frac{1}{N} \sum_{i \in [N]} \ell''(\mathbf{w}^\top \mathbf{x}_i)$ from Lemma I.1. Then,

$$\mathcal{L}(\mathbf{w}_{t+1}) \leq \mathcal{L}(\mathbf{w}_t) - \eta_t \langle \nabla \mathcal{L}(\mathbf{w}_t), \Delta_t \rangle + \frac{1}{2} \eta_t^2 \Delta_t^\top \nabla^2 \mathcal{L}(\mathbf{w}_t - \zeta \eta_t \Delta_t) \Delta_t$$

$$\leq \mathcal{L}(\mathbf{w}_t) - \eta_t \langle \nabla \mathcal{L}(\mathbf{w}_t), \Delta_t \rangle + \frac{1}{2} \eta_t^2 D^2 \mathcal{G}(\mathbf{w}_t - \zeta \eta_t \Delta_t)$$

$$\leq \mathcal{L}(\mathbf{w}_t) - \eta_t \langle \nabla \mathcal{L}(\mathbf{w}_t), \Delta_t \rangle + \frac{1}{2} \eta_t^2 D^2 e^{\eta_t D} \mathcal{G}(\mathbf{w}),$$

where we used $\mathcal{G}(\mathbf{w}') \leq e^{D\|\mathbf{w}' - \mathbf{w}\|_\infty} \mathcal{G}(\mathbf{w})$ for all $\mathbf{w}, \mathbf{w}'$ from Lemma I.1. Finally, choosing $C_H := \frac{1}{2} D^2 e^{\eta_0 D}$, we obtain the desired inequality. $\square$

**Lemma G.2** (EMA misalignment). *We denote* $\mathbf{e}_t := \mathbf{m}_t - \nabla L(\mathbf{w}_t)$. *Suppose that* $\beta \in (\frac{N-b}{N}, 1)$. *Then, there exists* $t_0 \in \mathbb{N}$ *such that for all* $t \geq t_0$,

$$\|\mathbf{e}_t\|_1 = \|\mathbf{m}_t - \nabla \mathcal{L}(\mathbf{w}_t)\|_1 \leq \left[ (1 - \beta) D \frac{N}{b} \left( \frac{N}{b} - 1 \right) + C_1 \eta_t + C_2 \beta^t \right] \mathcal{G}(\mathbf{w}_t)$$

*where* $C_1, C_2 > 0$ *are constants determined by* $\beta, N, b,$ *and* $D$.

*Proof.* The momentum $\mathbf{m}_t$ can be written as:

$$\mathbf{m}_t = (1 - \beta) \sum_{\tau=0}^{t} \beta^\tau \mathbf{g}_{t-\tau} = (1 - \beta) \sum_{\tau=0}^{t} \beta^\tau \nabla \mathcal{L}_{\mathcal{B}_{t-\tau}}(\mathbf{w}_{t-\tau}),$$

and the full-batch gradient $\nabla \mathcal{L}(\mathbf{w}_t)$ can be written as:

$$\nabla \mathcal{L}(\mathbf{w}_t) = \beta^{t+1} \nabla L(\mathbf{w}_t) + (1 - \beta) \sum_{\tau=0}^{t} \beta^\tau \nabla \mathcal{L}(\mathbf{w}_t),$$

Consequently, the misalignment $\mathbf{e}_t = \mathbf{m}_t - \nabla \mathcal{L}(\mathbf{w}_t)$ can be decomposed as:

$$\begin{aligned}
\mathbf{e}_t &= (1 - \beta) \sum_{\tau=0}^{t} \beta^\tau (\nabla \mathcal{L}_{\mathcal{B}_{t-\tau}}(\mathbf{w}_{t-\tau}) - \nabla \mathcal{L}_{\mathcal{B}_{t-\tau}}(\mathbf{w}_t)) \\
&\quad + (1 - \beta) \sum_{\tau=0}^{t} \beta^\tau (\nabla \mathcal{L}_{\mathcal{B}_{t-\tau}}(\mathbf{w}_t) - \nabla \mathcal{L}(\mathbf{w}_t)) \\
&\quad - \beta^{t+1} \nabla \mathcal{L}(\mathbf{w}_t),
\end{aligned}$$

and thus

$$\begin{aligned}
\|\mathbf{e}_t\|_1 &= \underbrace{\left\| (1 - \beta) \sum_{\tau=0}^{t} \beta^\tau (\nabla \mathcal{L}_{\mathcal{B}_{t-\tau}}(\mathbf{w}_{t-\tau}) - \nabla \mathcal{L}_{\mathcal{B}_{t-\tau}}(\mathbf{w}_t)) \right\|_1}_{\triangleq \text{(A)}} \\
&\quad + \underbrace{\left\| (1 - \beta) \sum_{\tau=0}^{t} \beta^\tau (\nabla \mathcal{L}_{\mathcal{B}_{t-\tau}}(\mathbf{w}_t) - \nabla \mathcal{L}(\mathbf{w}_t)) \right\|_1}_{\triangleq \text{(B)}} \\
&\quad + \underbrace{\left\| \beta^{t+1} \nabla \mathcal{L}(\mathbf{w}_t) \right\|_1}_{\triangleq \text{(C)}}.
\end{aligned}$$

We upper bound each term separately.

First, the term (A) represents the misalignment by the weight movement, which can be bounded as:

$$\begin{aligned}
\text{(A)} &= \left\| (1 - \beta) \sum_{\tau=0}^{t} \beta^\tau (\nabla \mathcal{L}_{\mathcal{B}_{t-\tau}}(\mathbf{w}_{t-\tau}) - \nabla \mathcal{L}_{\mathcal{B}_{t-\tau}}(\mathbf{w}_t)) \right\|_1 \\
&\leq (1 - \beta) \sum_{\tau=0}^{t} \beta^\tau \|\nabla \mathcal{L}_{\mathcal{B}_{t-\tau}}(\mathbf{w}_{t-\tau}) - \nabla \mathcal{L}_{\mathcal{B}_{t-\tau}}(\mathbf{w}_t)\|_1 \\
&= (1 - \beta) \sum_{\tau=0}^{t} \beta^\tau \left\| \frac{1}{b} \sum_{i \in \mathcal{B}_{t-\tau}} (\ell'(\mathbf{w}_{t-\tau}^\top \mathbf{x}_i) - \ell'(\mathbf{w}_t^\top \mathbf{x}_i)) \mathbf{x}_i \right\|_1 \\
&\leq (1 - \beta) \sum_{\tau=0}^{t} \beta^\tau \frac{D}{b} \sum_{i \in \mathcal{B}_{t-\tau}} |\ell'(\mathbf{w}_{t-\tau}^\top \mathbf{x}_i) - \ell'(\mathbf{w}_t^\top \mathbf{x}_i)| \\
&\leq \frac{(1 - \beta)D}{b} \sum_{\tau=0}^{t} \beta^\tau \sum_{i \in \mathcal{B}_{t-\tau}} |\ell'(\mathbf{w}_t^\top \mathbf{x}_i)| \left| \frac{\ell'(\mathbf{w}_{t-\tau}^\top \mathbf{x}_i)}{\ell'(\mathbf{w}_t^\top \mathbf{x}_i)} - 1 \right| \\
&\leq \frac{(1 - \beta)DN}{b} \mathcal{G}(\mathbf{w}_t) \sum_{\tau=0}^{t} \beta^\tau \sum_{i \in \mathcal{B}_{t-\tau}} \left| \frac{\ell'(\mathbf{w}_{t-\tau}^\top \mathbf{x}_i)}{\ell'(\mathbf{w}_t^\top \mathbf{x}_i)} - 1 \right|,
\end{aligned}$$

where we used $N\mathcal{G}(\mathbf{w}) = -\sum_{i \in [N]} \ell'(\mathbf{w}^\top \mathbf{x}_i) = \sum_{i \in [N]} |\ell'(\mathbf{w}^\top \mathbf{x}_i)| \geq \max_{i \in [N]} |\ell'(\mathbf{w}^\top \mathbf{x}_i)|$ in the last inequality. For all $i \in [N]$,

$$\left| \frac{\ell'(\mathbf{w}_{t-\tau}^\top \mathbf{x}_i)}{\ell'(\mathbf{w}_t^\top \mathbf{x}_i)} - 1 \right| \leq e^{|(\mathbf{w}_t - \mathbf{w}_{t-\tau})^\top \mathbf{x}_i|} - 1 \leq e^{\|\mathbf{w}_t - \mathbf{w}_{t-\tau}\|_\infty \|\mathbf{x}_i\|_1} - 1 \leq e^{D \sum_{\tau'=1}^{\tau} \eta_{t-\tau'}} - 1.$$

By Assumption 2.3, there exists $t_0 \in \mathbb{N}$ and constant $c_1 > 0$ determined by $\beta$ and $D$ such that $\sum_{\tau=0}^{t} \beta^\tau (e^{D \sum_{\tau'=1}^{\tau} \eta_{t-\tau'}} - 1) \le c_1 \eta_t$ for all $t \ge t_0$. Then, for all $t \ge t_0$, we have

$$
\begin{aligned}
(A) &\le \frac{(1-\beta)DN}{b} \mathcal{G}(\mathbf{w}_t) \sum_{\tau=0}^{t} \beta^\tau b(e^{D \sum_{\tau'=1}^{\tau} \eta_{t-\tau'}} - 1) \\
&= (1-\beta)DN\mathcal{G}(\mathbf{w}_t) \sum_{\tau=0}^{t} \beta^\tau e^{D \sum_{\tau'=1}^{\tau} \eta_{t-\tau'}} - 1 \\
&\le (1-\beta)DNc_1\eta_t \mathcal{G}(\mathbf{w}_t).
\end{aligned}
$$

Second, the term (B) represents the misalignment by mini-batch updates. Denote the number of mini-batches in a single epoch as $m := \frac{N}{b}$. Since $\mathcal{B}_t = \{(t \cdot b + i) \pmod{N}\}_{i=0}^{b-1}$, note that $\mathcal{B}_i = \mathcal{B}_j$ if and only if $i \equiv j \pmod{m}$. Now, the term (B) can be upper bounded as

$$
\begin{aligned}
(B) &= \left\| (1-\beta) \sum_{\tau=0}^{t} \beta^\tau (\nabla \mathcal{L}_{\mathcal{B}_{t-\tau}}(\mathbf{w}_t) - \nabla \mathcal{L}(\mathbf{w}_t)) \right\|_1 \\
&= \left\| (1-\beta) \sum_{\tau=0}^{t} \beta^\tau \left[ \nabla \mathcal{L}_{\mathcal{B}_{t-\tau}}(\mathbf{w}_t) - \frac{1}{m} \sum_{j=1}^{m} \nabla \mathcal{L}_{\mathcal{B}_j}(\mathbf{w}_t) \right] \right\|_1 \\
&= \left\| (1-\beta) \sum_{j=1}^{m} \left( \sum_{\tau \le t: (t-\tau) \equiv j \pmod{m}} \beta^\tau - \frac{1}{m} \sum_{\tau=0}^{t} \beta^\tau \right) \nabla \mathcal{L}_{\mathcal{B}_j}(\mathbf{w}_t) \right\|_1 \\
&\le (1-\beta)m \cdot \max_{j \in [m]} \left| \sum_{\tau \le t: (t-\tau) \equiv j \pmod{m}} \beta^\tau - \frac{1}{m} \sum_{\tau=0}^{t} \beta^\tau \right| \cdot \max_{j \in [m]} \|\nabla \mathcal{L}_{\mathcal{B}_j}(\mathbf{w}_t)\|_1 \\
&\le (1-\beta)Dm^2\mathcal{G}(\mathbf{w}_t) \cdot \max_{j \in [m]} \left| \sum_{\tau \le t: (t-\tau) \equiv j \pmod{m}} \beta^\tau - \frac{1}{m} \sum_{\tau=0}^{t} \beta^\tau \right|,
\end{aligned}
$$

where the last inequality holds since

$$
\max_{j \in [m]} \|\nabla \mathcal{L}_{\mathcal{B}_j}(\mathbf{w})\|_1 = \frac{1}{b} \max_{j \in [m]} \left\| \sum_{i \in \mathcal{B}_j} \ell'(\mathbf{w}^\top \mathbf{x}_i)\mathbf{x}_i \right\|_1 \le \frac{1}{b} \sum_{i=1}^{N} |\ell'(\mathbf{w}^\top \mathbf{x}_i)| \cdot D = \frac{DN}{b}\mathcal{G}(\mathbf{w}) = Dm\mathcal{G}(\mathbf{w}),
$$

for all $\mathbf{w} \in \mathbb{R}^d$.

It remains to upper bound $\max_{j \in [m]} \left| \sum_{\tau \leq t: (t-\tau) \equiv j \pmod{m}} \beta^\tau - \frac{1}{m} \sum_{\tau=0}^{t} \beta^\tau \right|$. Fix arbitrary $j \in [m]$. Note that

$$(1-\beta) \left( \sum_{\tau \leq t: (t-\tau) \equiv j \pmod{m}} \beta^\tau - \frac{1}{m} \sum_{\tau=0}^{t} \beta^\tau \right)$$

$$\leq (1-\beta) \sum_{k=0}^{\lfloor \frac{t}{m} \rfloor} \beta^{mk} - (1-\beta)\frac{1}{m} \sum_{\tau=0}^{t} \beta^\tau$$

$$= (1-\beta) \sum_{k=0}^{\lfloor \frac{t}{m} \rfloor} \beta^{mk} - (1-\beta) \sum_{k=0}^{\lfloor \frac{t}{m} \rfloor - 1} \left( \frac{1}{m} \beta^{mk} \sum_{\tau=0}^{m-1} \beta^\tau \right) - (1-\beta)\frac{1}{m} \sum_{\tau=m(\lfloor \frac{t}{m} \rfloor - 1)+1}^{t} \beta^\tau$$

$$\leq (1-\beta)\beta^{m\lfloor \frac{t}{m} \rfloor} + \sum_{k=0}^{\lfloor \frac{t}{m} \rfloor - 1} \beta^{mk} \left[ (1-\beta) - \frac{1}{m}(1-\beta^m) \right]$$

$$\overset{(*)}{\leq} (1-\beta)\beta^{t-m} + \sum_{k=0}^{\lfloor \frac{t}{m} \rfloor - 1} \beta^{mk} \frac{(m-1)(1-\beta)^2}{2}$$

$$\leq (1-\beta)\beta^{t-m} + \frac{1}{1-\beta^m} \cdot \frac{(m-1)(1-\beta)^2}{2}$$

$$\overset{(**)}{\leq} (1-\beta)\beta^{t-m} + \frac{2}{m(1-\beta)} \cdot \frac{(m-1)(1-\beta)^2}{2}$$

$$= (1-\beta)\beta^{t-m} + \frac{m-1}{m}(1-\beta),$$

where the inequalities $(*)$ and $(**)$ hold since $(1-\epsilon)^m \leq 1 - m\epsilon + \frac{m(m-1)}{2}\epsilon^2 \leq 1 - \frac{m}{2}\epsilon$ for all $0 \leq \epsilon \leq \frac{1}{m-1}$ and choose $\epsilon = 1 - \beta$.

Similarly, we have

$$(1-\beta)\left(\frac{1}{m}\sum_{\tau=0}^{t}\beta^\tau - \sum_{\tau\le t:\,(t-\tau)\equiv j\pmod{m}}\beta^\tau\right)$$

$$\le (1-\beta)\frac{1}{m}\sum_{\tau=0}^{t}\beta^\tau - (1-\beta)\sum_{k=0}^{\lfloor\frac{t+1}{m}\rfloor-1}\beta^{m(k+1)-1}$$

$$= (1-\beta)\sum_{k=0}^{\lfloor\frac{t+1}{m}\rfloor-1}\left(\frac{1}{m}\beta^{mk}\sum_{\tau=0}^{m-1}\beta^\tau\right) + (1-\beta)\frac{1}{m}\sum_{\tau=m\lfloor\frac{t+1}{m}\rfloor}^{t}\beta^\tau - (1-\beta)\sum_{k=0}^{\lfloor\frac{t+1}{m}\rfloor-1}\beta^{m(k+1)-1}$$

$$\le (1-\beta)\frac{1}{m}\sum_{\tau=t-m+2}^{t}\beta^\tau + \sum_{k=0}^{\lfloor\frac{t+1}{m}\rfloor-1}\beta^{mk}\left[\frac{1}{m}(1-\beta^m) - (1-\beta)\beta^{m-1}\right]$$

$$= \frac{1}{m}\beta^{t-m+2}(1-\beta^{m-1}) + \sum_{k=0}^{\lfloor\frac{t+1}{m}\rfloor-1}\beta^{mk}\left[\frac{1}{m}(1-\beta^m) - (1-\beta)\beta^{m-1}\right]$$

$$\le \frac{1}{m}\beta^{t-m+2}(1-\beta^{m-1}) + \sum_{k=0}^{\lfloor\frac{t+1}{m}\rfloor-1}\beta^{mk}\frac{(m-1)(1-\beta)^2}{2}$$

$$\le \frac{1}{m}\beta^{t-m+2}(1-\beta^{m-1}) + \frac{1}{1-\beta^m}\cdot\frac{(m-1)(1-\beta)^2}{2}$$

$$\le (1-\beta)\beta^{t-m} + \frac{m-1}{m}(1-\beta).$$

Combining the bounds, we get

$$(\text{B}) \le (1-\beta)Dm(\beta^{t-m}m + m - 1)\mathcal{G}(\mathbf{w}_t).$$

Finally,

$$(\text{C}) = \|\beta^{t+1}\nabla\mathcal{L}(\mathbf{w}_t)\|_1 \le \beta^{t+1}D\mathcal{G}(\mathbf{w}_t).$$

Therefore, we conclude

$$\|\mathbf{e}\|_1 \le \left[(1-\beta)Dm(m-1) + C_1\eta_t + C_2\beta^t\right]\mathcal{G}(\mathbf{w}_t)$$

where $C_1, C_2 > 0$ are constants determined by $\beta$, $m$, and $D$. $\qquad\square$

**Corollary G.3.** *Suppose that $\beta \in (\frac{N-b}{N}, 1)$. Then, there exists $t_0 \in \mathbb{N}$ such that for all $t \ge t_0$,* Inc-Signum *iterates $\{\mathbf{w}_t\}$ satisfy*

$$\mathcal{L}(\mathbf{w}_{t+1}) \le \mathcal{L}(\mathbf{w}_t) - \eta_t(\gamma_\infty - 2(1-\beta)D\tfrac{N}{b}(\tfrac{N}{b}-1) - (2C_1 + C_H)\eta_t - 2C_2\beta^t)\mathcal{G}(\mathbf{w}_t),$$

*where $C_H, C_1, C_2 > 0$ are constants in Lemmas G.1 and G.2.*

*Proof.* By Lemma I.1, we get

$$\begin{aligned}
\langle\nabla\mathcal{L}(\mathbf{w}_t), \Delta_t\rangle &= \langle\mathbf{m}_t, \Delta_t\rangle - \langle\mathbf{e}_t, \Delta_t\rangle \\
&\ge \|\mathbf{m}_t\|_1 - \|\mathbf{e}_t\|_1\|\Delta_t\|_\infty \\
&\ge (\|\nabla\mathcal{L}(\mathbf{w}_t)\|_1 - \|\mathbf{e}_t\|_1) - \|\mathbf{e}_t\|_1 \\
&= \|\nabla\mathcal{L}(\mathbf{w}_t)\|_1 - 2\|\mathbf{e}_t\|_1 \\
&\ge \gamma_\infty\mathcal{G}(\mathbf{w}_t) - 2\|\mathbf{e}_t\|_1.
\end{aligned}$$

Now using Lemma G.1 and Lemma G.2, we conclude

$$\begin{aligned}
\mathcal{L}(\mathbf{w}_{t+1}) &\le \mathcal{L}(\mathbf{w}_t) - \eta_t\langle\nabla\mathcal{L}(\mathbf{w}_t), \Delta_t\rangle + C_H\eta_t^2\mathcal{G}(\mathbf{w}_t) \\
&\le \mathcal{L}(\mathbf{w}_t) - \eta_t(\gamma_\infty\mathcal{G}(\mathbf{w}_t) - 2\|\mathbf{e}_t\|_1) + C_H\eta_t^2\mathcal{G}(\mathbf{w}_t) \\
&\le \mathcal{L}(\mathbf{w}_t) - \eta_t(\gamma_\infty - 2(1-\beta)D\tfrac{N}{b}(\tfrac{N}{b}-1) - (2C_1 + C_H)\eta_t - 2C_2\beta^t)\mathcal{G}(\mathbf{w}_t),
\end{aligned}$$

which ends the proof. $\qquad\square$

**Proposition G.4** (Loss convergence). *Suppose that $\beta \in (1 - \frac{\gamma_\infty}{4C_0}, 1)$ if $b < N$ and $\beta \in (0, 1)$ if $b = N$, where $C_0 := D\frac{N}{b}(\frac{N}{b} - 1)$. Then, $\mathcal{L}(\mathbf{w}_t) \to 0$ as $t \to \infty$.*

*Proof.* Note that $\beta \in (\frac{N-b}{N}, 1)$ since $\gamma_\infty = \max_{\|\mathbf{w}\|_\infty \leq 1} \min_{i \in [N]} \mathbf{w}^\top \mathbf{x}_i \leq D$. By Corollary G.3, there exists $t_0 \in \mathbb{N}$ such that for all $t \geq t_0$,

$$\eta_t(\gamma_\infty - 2C_0(1-\beta) - (2C_1 + C_H)\eta_t - 2C_2\beta^t)\mathcal{G}(\mathbf{w}_t) \leq \mathcal{L}(\mathbf{w}_t) - \mathcal{L}(\mathbf{w}_{t+1}).$$

Since $\eta_t, \beta^t \to 0$ as $t \to \infty$, there exists $t_1 \geq t_0$ such that for all $t \geq t_1$,

$$(2C_1 + C_H)\eta_t + 2C_2\beta^t < \frac{\gamma_\infty}{4}.$$

Then,

$$\frac{\gamma_\infty}{4}\sum_{t=t_1}^\infty \eta_t\mathcal{G}(\mathbf{w}_t) \leq \sum_{t=t_1}^\infty \eta_t(\gamma_\infty - 2C_0(1-\beta) - (2C_1+C_H)\eta_t - 2C_2\beta^t)\mathcal{G}(\mathbf{w}_t) \leq \sum_{t=t_1}^\infty \mathcal{L}(\mathbf{w}_t) - \mathcal{L}(\mathbf{w}_{t+1}) < \infty.$$

Thus, $\sum_{t=t_0}^\infty \eta_t\mathcal{G}(\mathbf{w}_t) < \infty$ and since $\sum_{t=t_0}^\infty \eta_t = \infty$, this implies $\mathcal{G}(\mathbf{w}_t) \to 0$ and therefore $\mathcal{L}(\mathbf{w}_t) \to 0$ as $t \to \infty$. $\qquad\square$

**Proposition G.5** (Unnormalized margin lower bound). *Suppose that $\beta \in (1 - \frac{\gamma_\infty}{4C_0}, 1)$ if $b < N$ and $\beta \in (0, 1)$ if $b = N$, where $C_0 := D\frac{N}{b}(\frac{N}{b} - 1)$. Then, there exists $t_s \in \mathbb{N}$ such that for all $t \geq t_s$,*

$$\min_{i \in [N]} \mathbf{w}^\top \mathbf{x}_i \leq (\gamma_\infty - 2C_0(1-\beta))\sum_{\tau=t_s}^{t-1} \eta_\tau \frac{\mathcal{G}(\mathbf{w}_\tau)}{\mathcal{L}(\mathbf{w}_\tau)} - (2C_1 + C_H)\sum_{\tau=t_s}^{t-1} \eta_\tau^2 - \frac{2C_2\eta_0}{1-\beta},$$

*where $C_0 := D\frac{N}{b}(\frac{N}{b} - 1)$ and $C_H, C_1, C_2 > 0$ are constants in Lemmas G.1 and G.2.*

*Proof.* By Proposition G.4, there exists time step $t_s \in \mathbb{N}$ such that $\mathcal{L}(\mathbf{w}_t) \leq \frac{\log 2}{N}$ for all $t \geq t_s$. Then, $\ell(\mathbf{w}_t^\top \mathbf{x}_i) \leq \frac{1}{N}\mathcal{L}(\mathbf{w}_t) \leq \log 2 < 1$, and thus $\min_{i \in [N]} \mathbf{w}_t^\top \mathbf{x}_i \geq 0$ for all $t \geq t_s$. Then, for all $t \geq t_s$,

$$\exp(-\min_{i \in [N]} \mathbf{w}_t^\top \mathbf{x}_i) = \max_{i \in [N]} \exp(-\mathbf{w}_t^\top \mathbf{x}_i) \leq \frac{1}{\log 2}\max_{i \in [N]} \log(1 + \exp(-\mathbf{w}^\top \mathbf{x}_i)) \leq \frac{N\mathcal{L}(\mathbf{w}_t)}{\log 2},$$

for logistic loss, and $\exp(-\min_{i \in [N]} \mathbf{w}_t^\top \mathbf{x}_i) \leq N\mathcal{L}(\mathbf{w}_t) \leq \frac{N\mathcal{L}(\mathbf{w}_t)}{\log 2}$ for exponential loss.

Using Corollary G.3 and $\mathcal{G}(\mathbf{w}) \leq \mathcal{L}(\mathbf{w})$ from Lemma I.1, we get

$$\mathcal{L}(\mathbf{w}_t) \leq \mathcal{L}(\mathbf{w}_{t-1})\left(1 - (\gamma_\infty - 2C_0(1-\beta))\eta_{t-1}\frac{\mathcal{G}(\mathbf{w}_{t-1})}{\mathcal{L}(\mathbf{w}_{t-1})} + (2C_1 + C_H)\eta_{t-1}^2 + 2C_2\beta^{t-1}\eta_{t-1}\right)$$

$$\leq \mathcal{L}(\mathbf{w}_{t-1})\exp\left(-(\gamma_\infty - 2C_0(1-\beta))\eta_{t-1}\frac{\mathcal{G}(\mathbf{w}_{t-1})}{\mathcal{L}(\mathbf{w}_{t-1})} + (2C_1 + C_H)\eta_{t-1}^2 + 2C_2\beta^{t-1}\eta_{t-1}\right)$$

$$\leq \mathcal{L}(\mathbf{w}_{t_s})\exp\left(-(\gamma_\infty - 2C_0(1-\beta))\sum_{\tau=t_s}^{t-1}\eta_\tau\frac{\mathcal{G}(\mathbf{w}_\tau)}{\mathcal{L}(\mathbf{w}_\tau)} + (2C_1 + C_H)\sum_{\tau=t_s}^{t-1}\eta_\tau^2 + 2C_2\sum_{\tau=t_s}^{t-1}\beta^\tau\eta_\tau\right)$$

$$\leq \frac{\log 2}{N}\exp\left(-(\gamma_\infty - 2C_0(1-\beta))\sum_{\tau=t_s}^{t-1}\eta_\tau\frac{\mathcal{G}(\mathbf{w}_\tau)}{\mathcal{L}(\mathbf{w}_\tau)} + (2C_1 + C_H)\sum_{\tau=t_s}^{t-1}\eta_\tau^2 + \frac{2C_2\eta_0}{1-\beta}\right).$$

Thus, we get

$$\exp(-\min_{i \in [N]} \mathbf{w}_t^\top \mathbf{x}_i) \leq \frac{N\mathcal{L}(\mathbf{w}_t)}{\log 2}$$

$$\leq \exp\left(-(\gamma_\infty - 2C_0(1-\beta))\sum_{\tau=t_s}^{t-1}\eta_\tau\frac{\mathcal{G}(\mathbf{w}_\tau)}{\mathcal{L}(\mathbf{w}_\tau)} + (2C_1 + C_H)\sum_{\tau=t_s}^{t-1}\eta_\tau^2 + \frac{2C_2\eta_0}{1-\beta}\right),$$

which gives the desired inequality. $\qquad\square$

**Theorem 5.1.** *Let $\delta > 0$. Then there exists $\epsilon > 0$ such that the iterates $\{\mathbf{w}_t\}_{t=0}^{\infty}$ of* `Inc-Signum` *(Algorithm 4) with batch size $b$ and momentum $\beta \in (1 - \epsilon, 1)$, under Assumptions 2.1 and 2.3, satisfy*

$$\liminf_{t \to \infty} \frac{\min_{i \in [N]} \mathbf{x}_i^{\top} \mathbf{w}_t}{\|\mathbf{w}_t\|_{\infty}} \geq \gamma_{\infty} - \delta, \tag{7}$$

*where*

$$\gamma_{\infty} \triangleq \max_{\|\mathbf{w}\|_{\infty} \leq 1} \min_{i \in [N]} \mathbf{w}^{\top} \mathbf{x}_i, \quad D \triangleq \max_{i \in [N]} \|\mathbf{x}_i\|_1,$$

*and*

$$\epsilon = \frac{1}{2D \cdot \frac{N}{b}(\frac{N}{b} - 1)} \min\{\delta, \tfrac{\gamma_{\infty}}{2}\} \quad \text{if } b < N, \qquad \epsilon = 1 \quad \text{if } b = N.$$

*Proof.* Let $C_0 := D\frac{N}{b}(\frac{N}{b} - 1)$ so that $\epsilon := \min\{\frac{\delta}{2C_0}, \frac{\gamma_{\infty}}{4C_0}\}$ if $b < N$ and $\epsilon := 1$ if $b = N$. Note that $C_0 = 0$ if $b = N$. Suppose that $\beta \in (1 - \epsilon, 1)$.

Let $t_0$ be a time step that satisfy Corollary G.3. By Proposition G.4, there exists $t^{\star} \geq t_0$ such that $(2C_1 + C_H)\eta_t + 2C_2\beta^t < \frac{\gamma_{\infty}}{8}$ and $\mathcal{L}(\mathbf{w}_t) \leq \frac{\log 2}{N}$ for all $t \geq t^{\star}$. Then, for each $t \geq t^{\star}$, we get $\frac{\mathcal{G}(\mathbf{w}_t)}{\mathcal{L}(\mathbf{w}_t)} \geq 1 - \frac{N\mathcal{L}(\mathbf{w}_t)}{2} \geq \frac{1}{2}$. By Corollary G.3, for all $t \geq t^{\star}$,

$$\mathcal{L}(\mathbf{w}_t) \leq \mathcal{L}(\mathbf{w}_{t-1}) \left(1 - (\gamma_{\infty} - 2C_0(1 - \beta))\eta_{t-1}\frac{\mathcal{G}(\mathbf{w}_{t-1})}{\mathcal{L}(\mathbf{w}_{t-1})} + (2C_1 + C_H)\eta_{t-1}^2 + 2C_2\beta^{t-1}\eta_{t-1}\right)$$

$$\leq \mathcal{L}(\mathbf{w}_{t-1}) \left(1 - \frac{1}{4}\gamma_{\infty}\eta_{t-1} + \frac{1}{8}\gamma_{\infty}\eta_{t-1}\right)$$

$$\leq \mathcal{L}(\mathbf{w}_{t-1}) \exp\left(-\frac{1}{8}\gamma_{\infty}\eta_{t-1}\right)$$

$$\leq \mathcal{L}(\mathbf{w}_{t^{\star}}) \exp\left(-\frac{\gamma_{\infty}}{8}\sum_{\tau=t^{\star}}^{t-1}\eta_{\tau}\right)$$

$$\leq \frac{\log 2}{N} \exp\left(-\frac{\gamma_{\infty}}{8}\sum_{\tau=t^{\star}}^{t-1}\eta_{\tau}\right).$$

Consequently, by Lemma I.1, we have

$$\frac{\mathcal{G}(\mathbf{w}_t)}{\mathcal{L}(\mathbf{w}_t)} \geq 1 - \frac{N\mathcal{L}(\mathbf{w}_t)}{2} \geq 1 - \exp\left(-\frac{\gamma_{\infty}}{8}\sum_{\tau=t^{\star}}^{t-1}\eta_{\tau}\right),$$

for all $t \geq t^{\star}$.

Finally, using Proposition G.5, we get

$$\gamma_{\infty} - 2C_0(1 - \beta) - \frac{\min_{i \in [N]} \mathbf{w}_t^{\top} \mathbf{x}_i}{\|\mathbf{w}_t\|_{\infty}}$$

$$\leq \frac{(\gamma_{\infty} - 2C_0(1 - \beta))\left(\|\mathbf{w}_0\| + \sum_{\tau=0}^{t^{\star}-1}\eta_{\tau} + \sum_{\tau=t^{\star}}^{t}\eta_{\tau}e^{-\frac{\gamma_{\infty}}{8}\sum_{\tau=t^{\star}}^{t-1}\eta_{\tau}}\right) + (2C_1 + C_H)\sum_{\tau=t^{\star}}^{t-1}\eta_{\tau}^2 + \frac{2C_2\eta_0}{1-\beta}}{\|\mathbf{w}_0\| + \sum_{\tau=0}^{t-1}\eta_{\tau}}$$

$$= \mathcal{O}\left(\frac{\sum_{\tau=0}^{t^{\star}-1}\eta_{\tau} + \sum_{\tau=t^{\star}}^{t}\eta_{\tau}e^{-\frac{\gamma_{\infty}}{8}\sum_{\tau=t^{\star}}^{t-1}\eta_{\tau}} + \sum_{\tau=t^{\star}}^{t-1}\eta_{\tau}^2}{\sum_{\tau=0}^{t-1}\eta_{\tau}}\right)$$

Therefore, we conclude

$$\liminf_{t \to \infty} \frac{\min_{i \in [N]} \mathbf{w}_t^{\top} \mathbf{x}_i}{\|\mathbf{w}_t\|_{\infty}} \geq \gamma_{\infty} - 2C_0(1 - \beta) \geq \gamma - \delta.$$

$\square$

## H    MISSING PROOFS IN APPENDIX A

**Lemma A.1.** *Suppose that (a) $\mathcal{L}(\mathbf{w}_r) \to 0$ and (b) $\mathbf{w}_r = \|\mathbf{w}_r\|_2 \hat{\mathbf{w}} + \boldsymbol{\rho}(r)$ for some $\hat{\mathbf{w}}$ with $\exists \lim_{r\to} \boldsymbol{\rho}(r)$. Then, under Assumptions 2.1 and 2.2, there exists $\mathbf{c} = (c_0, \cdots, c_{N-1}) \in \Delta^{N-1}$ such that the limit direction $\hat{\mathbf{w}}$ of* Inc-Adam *with $\beta_1 = 0$ satisfies*

$$\hat{\mathbf{w}} \propto \sum_{i\in[N]} \frac{c_i \mathbf{x}_i}{\sqrt{\sum_{j\in[N]} \beta_2^{(i,j)} c_j^2 \mathbf{x}_j^2}}, \tag{8}$$

*and $c_i = 0$ for $i \notin S$, where $S = \arg\min_{i\in[N]} \hat{\mathbf{w}}^\top \mathbf{x}_i$ is the index set of support vectors of $\hat{\mathbf{w}}$.*

*Proof.* We start with the case of $\ell = \ell_{\exp}$. First step is to characterize $\hat{\boldsymbol{\delta}}$, the limit of $\boldsymbol{\delta}_r$. Notice that (b) is a strictly stronger assumption than Assumption 4.4 and it simplifies the analysis, while maintaining the intuition that the terms of support vectors dominate the update direction. Let $\lim_{r\to\infty} \boldsymbol{\rho}(r) = \hat{\boldsymbol{\rho}}$. We recall previous notations as $\gamma = \min_i \langle \mathbf{x}_i, \hat{\mathbf{w}} \rangle, \bar{\gamma}_i = \langle \mathbf{x}_i, \hat{\mathbf{w}} \rangle, \bar{\gamma} = \min_{i\notin S} \langle \mathbf{x}_i, \hat{\mathbf{w}} \rangle$. Then it satisfies $S = \{i \in [N] : \langle \mathbf{x}_i, \hat{\mathbf{w}} \rangle = \gamma\}$ and $\bar{\gamma} > \gamma > 0$. We can decompose dominant and residual terms in the update rule as follows.

$$\boldsymbol{\delta}_r = \sum_{i\in S} \frac{\exp(-\gamma g(r)) \exp(-\boldsymbol{\rho}(r)^\top \mathbf{x}_i) \mathbf{x}_i}{\sqrt{\sum_{j\in[N]} \beta_2^{(i,j)} \exp(-2\bar{\gamma}_j g(r)) \exp(-2\boldsymbol{\rho}(r)^\top \mathbf{x}_j) \mathbf{x}_j^2}}$$

$$+ \sum_{i\in S^\complement} \frac{\exp(-\bar{\gamma}_i g(r)) \exp(-\boldsymbol{\rho}(r)^\top \mathbf{x}_i) \mathbf{x}_i}{\sqrt{\sum_{j\in[N]} \beta_2^{(i,j)} \exp(-2\bar{\gamma}_j g(r)) \exp(-2\boldsymbol{\rho}(r)^\top \mathbf{x}_j) \mathbf{x}_j^2}} + \boldsymbol{\epsilon}_r$$

$$= \sum_{i\in S} \frac{\exp(-\boldsymbol{\rho}(r)^\top \mathbf{x}_i) \mathbf{x}_i}{\sqrt{\sum_{j\in[N]} \beta_2^{(i,j)} \exp(-2(\bar{\gamma}_j - \gamma) g(r)) \exp(-2\boldsymbol{\rho}(r)^\top \mathbf{x}_j) \mathbf{x}_j^2}}$$

$$+ \sum_{i\in S^\complement} \frac{\exp(-(\bar{\gamma}_j - \gamma) g(r)) \exp(-\boldsymbol{\rho}(r)^\top \mathbf{x}_i) \mathbf{x}_i}{\sqrt{\sum_{j\in[N]} \beta_2^{(i,j)} \exp(-2(\bar{\gamma}_j - \gamma) g(r)) \exp(-2\boldsymbol{\rho}(r)^\top \mathbf{x}_j) \mathbf{x}_j^2}} + \boldsymbol{\epsilon}_r$$

$$\triangleq \mathbf{d}(r) + \mathbf{r}(r) + \boldsymbol{\epsilon}_r.$$

Since $\bar{\gamma}_j > \gamma$ and $g(r) \to \infty$, $\mathbf{r}(r)$ converges to 0. Therefore, we get

$$\hat{\boldsymbol{\delta}} \triangleq \lim_{r\to\infty} \boldsymbol{\delta}_r = \lim_{r\to\infty} \mathbf{d}(r) = \lim_{r\to\infty} \sum_{i\in S} \frac{\exp(-\boldsymbol{\rho}(r)^\top \mathbf{x}_i) \mathbf{x}_i}{\sqrt{\sum_{j\in S} \beta_2^{(i,j)} \exp(-2\boldsymbol{\rho}(r)^\top \mathbf{x}_j) \mathbf{x}_j^2}}$$

$$= \sum_{i\in S} \frac{\exp(-\hat{\boldsymbol{\rho}}^\top \mathbf{x}_i) \mathbf{x}_i}{\sqrt{\sum_{j\in S} \beta_2^{(i,j)} \exp(-2\hat{\boldsymbol{\rho}}^\top \mathbf{x}_j) \mathbf{x}_j^2}}$$

$$= \sum_{i\in[N]} \frac{c_i \mathbf{x}_i}{\sqrt{\sum_{j\in[N]} \beta_2^{(i,j)} c_j^2 \mathbf{x}_j^2}},$$

for some $\mathbf{c} \in \Delta^{N-1}$ satisfying $c_i = 0$ for $i \notin S$. Using the same technique based on Stolz-Cesaro theorem, we can also deduce that $\hat{\mathbf{w}} = \hat{\boldsymbol{\delta}}$. Since we can extend this result to $\ell = \ell_{\log}$ following the proof of Lemma 4.5, the statement is proved. $\square$

## I    TECHNICAL LEMMAS

### I.1    PROXY FUNCTION

**Lemma I.1** (Proxy function). *The proxy function $\mathcal{G}$ satisfy the following properties: for any given weights $\mathbf{w}, \mathbf{w}' \in \mathbb{R}^d$ and any norm $\|\cdot\|$,*

*(a)* $\gamma_{\|\cdot\|} \mathcal{G}(\mathbf{w}) \leq \|\nabla \mathcal{L}(\mathbf{w})\|_* \leq D\mathcal{G}(\mathbf{w})$, *where* $D = \max_{i\in[N]} \|\mathbf{x}_i\|_*$ *and* $\gamma_{\|\cdot\|} = \max_{\|\mathbf{w}\|\leq 1} \min_{i\in[N]} \mathbf{w}^\top \mathbf{x}_i$ *is the $\|\cdot\|$-normalized max margin,*

*(b)* $1 - \frac{N\mathcal{L}(\mathbf{w})}{2} \leq \frac{\mathcal{G}(\mathbf{w})}{\mathcal{L}(\mathbf{w})} \leq 1$,

*(c)* $\mathcal{G}(\mathbf{w}) \geq \frac{1}{N} \sum_{i \in [N]} \ell''(\mathbf{w}^\top \mathbf{x}_i)$,

*(d)* $\mathcal{G}(\mathbf{w}') \leq e^{B\|\mathbf{w}' - \mathbf{w}\|} \mathcal{G}(\mathbf{w})$, *where* $D = \max_{i \in [N]} \|\mathbf{x}_i\|_*$.

*Proof.* This lemma (or a similar variant) is proved in Zhang et al. (2024a) and Fan et al. (2025). Below, we provide a proof for completeness.

(a) First, by duality we get

$$\|\nabla \mathcal{L}(\mathbf{w})\|_* = \max_{\|\mathbf{g}\| \leq 1} \langle \mathbf{g}, -\nabla \mathcal{L}(\mathbf{w}) \rangle \geq \max_{\|\mathbf{g}\| \leq 1} -\frac{1}{N} \sum_{i \in [N]} \ell'(\mathbf{w}^\top \mathbf{x}_i) \mathbf{g}^\top \mathbf{x}_i$$
$$\geq \mathcal{G}(\mathbf{w}) \max_{\|\mathbf{g}\| \leq 1} \min_{i \in [N]} \mathbf{g}^\top \mathbf{x}_i$$
$$= \gamma_{\|\cdot\|} \mathcal{G}(\mathbf{w}).$$

Second, we can obtain the lower bound as

$$\|\nabla \mathcal{L}(\mathbf{w})\|_* = \| -\frac{1}{N} \sum_{i \in [N]} \ell'(\mathbf{w}^\top \mathbf{x}_i) \mathbf{x}_i\|_* \leq -\frac{1}{N} \sum_{i \in [N]} \ell'(\mathbf{w}^\top \mathbf{x}_i) \|\mathbf{x}_i\|_* \leq D\mathcal{G}(\mathbf{w}).$$

(b) For exponential loss, $\frac{\mathcal{G}(\mathbf{w})}{\mathcal{L}(\mathbf{w})} = 1$. For logistic loss, the lower bound $\frac{\mathcal{G}(\mathbf{w})}{\mathcal{L}(\mathbf{w})} \geq 1 - \frac{N\mathcal{L}(\mathbf{w})}{2}$ follows from Zhang et al. (2024a, Lemma C.7). The upper bound follows from the elementary inequality $-\ell'_{\log}(z) = \frac{\exp(-z)}{1 + \exp(-z)} \leq \log(1 + \exp(-z)) = \ell_{\log}(z)$ for all $z \in \mathbb{R}$.

(c) For exponential loss, the equality holds. For logistic loss, the elementary inequality $-\ell'_{\log}(z) = \frac{\exp(-z)}{1 + \exp(-z)} \geq \frac{\exp(-z)}{(1 + \exp(-z))^2} = \ell''_{\log}(z)$ for all $z \in \mathbb{R}$, which results in

$$\mathcal{G}(\mathbf{w}) = -\frac{1}{N} \sum_{i \in [N]} \ell'(\mathbf{w}^\top \mathbf{x}_i) \geq \frac{1}{N} \sum_{i \in [N]} \ell''(\mathbf{w}^\top \mathbf{x}_i).$$

(d) First, for exponential loss, $-\ell'_{\exp}(z') = -\exp(z - z')\ell'_{\exp}(z) \leq -\exp(|z' - z|)\ell'_{\exp}(z)$, and for logistic loss, $-\ell'_{\log}(z') = \frac{\exp(z) + 1}{\exp(z') + 1} \ell'_{\log}(z) \leq -\exp(|z' - z|)\ell'_{\log}(z)$ hold for any $z, z' \in \mathbb{R}$. By duality, we get

$$\mathcal{G}(\mathbf{w}') = -\frac{1}{N} \sum_{i \in [N]} \ell'(\mathbf{w}'^\top \mathbf{x}_i) = -\frac{1}{N} \sum_{i \in [N]} \ell'(\mathbf{w}^\top \mathbf{x}_i + (\mathbf{w}' - \mathbf{w})^\top \mathbf{x}_i)$$
$$\leq -\frac{1}{N} \sum_{i \in [N]} \ell'(\mathbf{w}^\top \mathbf{x}_i) \exp(|(\mathbf{w}' - \mathbf{w})^\top \mathbf{x}_i|)$$
$$\leq -\frac{1}{N} \sum_{i \in [N]} \ell'(\mathbf{w}^\top \mathbf{x}_i) \exp(\|\mathbf{w}' - \mathbf{w}\| \|\mathbf{x}_i\|_*)$$
$$\leq -\frac{1}{N} \sum_{i \in [N]} \ell'(\mathbf{w}^\top \mathbf{x}_i) \exp(D\|\mathbf{w}' - \mathbf{w}\|)$$
$$= e^{D\|\mathbf{w}' - \mathbf{w}\|} \mathcal{G}(\mathbf{w}).$$

$\square$

## I.2 PROPERTIES OF LOSS FUNCTIONS

**Lemma I.2** (Lemma C.4 in Zhang et al. (2024a)). *For* $\ell \in \{\ell_{exp}, \ell_{log}\}$, *either* $\mathcal{G}(\mathbf{w}) < \frac{1}{2n}$ *or* $\mathcal{L}(\mathbf{w}) < \frac{\log 2}{n}$ *implies* $\mathbf{w}^\top \mathbf{x}_i > 0$ *for all* $i \in [N]$.

**Lemma I.3** (Lemma C.5 in Zhang et al. (2024a)). *For $\ell \in \{\ell_{exp}, \ell_{log}\}$ and any $z_1, z_2 \in \mathbb{R}$, we have*

$$\left| \frac{\ell'(z_1)}{\ell'(z_2)} - 1 \right| \leq e^{|z_1 - z_2|} - 1.$$

**Lemma I.4** (Lemma C.6 in Zhang et al. (2024a)). *For $\ell \in \{\ell_{exp}, \ell_{log}\}$ and any $z_1, z_2, z_3, z_4 \in \mathbb{R}$, we have*

$$\left| \frac{\ell'(z_1)\ell'(z_3)}{\ell'(z_2)\ell'(z_4)} - 1 \right| \leq \left( e^{|z_1 - z_2|} - 1 \right) + \left( e^{|z_3 - z_4|} - 1 \right) + \left( e^{|z_1 + z_3 - z_2 - z_4|} - 1 \right).$$

**Lemma I.5.** *For $a > 1$ and $z_1, z_2 > 0$, if $\ell_{log}(z_1) \leq a\ell_{log}(z_2)$, then $z_1 \geq z_2 - \log(2^a - 1)$.*

*Proof.* Note that

$$\log(1 + e^{-z_1}) \leq a \log(1 + e^{-z_2}) \implies e^{-z_1} \leq (1 + e^{-z_2})^a - 1,$$

and define $f(x) = \frac{(1+x)^a - 1}{x}$. Since $f$ is an increasing function on the interval $(0, 1)$, we get $\sup_{x \in (0,1)} f(x) = f(1) = 2^a - 1$. This implies $(1 + x)^a - 1 \leq (2^a - 1)x$ for $x \in (0, 1)$. Since $z_1, z_2 > 0$, it satisfies $e^{-z_1}, e^{-z_2} \in (0, 1)$. Therefore, we get

$$e^{-z_1} \leq (1 + e^{-z_2})^a - 1 \leq (2^a - 1)e^{-z_2}.$$

By taking the natural logarithm of both sides, we get the desired inequality. $\qquad\square$

### I.3 AUXILIARY RESULTS

**Lemma I.6** (Lemma C.1 in Zhang et al. (2024a)). *The learning rate $\eta_t = (t + 2)^{-a}$ with $a \in (0, 1]$ satisfies Assumption 2.3.*

**Lemma I.7** (Bernoulli's Inequality). *(a) If $r \geq 1$ and $x \geq -1$, then $(1 + x)^r \geq 1 + rx$.*

*(b) If $0 \leq r \leq 1$ and $x \geq -1$, then $(1 + x)^r \leq 1 + rx$.*

**Lemma I.8** (Stolz-Cesaro Theorem). *Let $(a_n)_{n \geq 1}$ and $(b_n)_{n \geq 1}$ be the two sequences of real numbers. Assume that $(b_n)_{n \geq 1}$ is strictly monotone and divergent sequence and the following limit exists:*

$$\lim_{n \to \infty} \frac{a_{n+1} - a_n}{b_{n+1} - b_n} = l.$$

*Then it satisfies that*

$$\lim_{n \to \infty} \frac{a_n}{b_n} = l.$$

**Lemma I.9** (Brouwer Fixed-point Theorem). *Every continuous function from a nonempty convex compact subset of $\mathbb{R}^d$ to itself has a fixed point.*

