# OpenReview forum: "Implicit Bias of Per-sample Adam on Separable Data: Departure from the Full-batch Regime"
_ICLR.cc/2026/Conference — ICLR 2026 Poster_

### Official Review · Reviewer_yBsj · 2025-10-16

**Soundness:** 3
**Presentation:** 2
**Contribution:** 2
**Rating:** 4
**Confidence:** 5

**Summary:**

In this paper, the authors study the implicit bias of Adam under stochastic mini-batch settings. Under certain assumptions, authors propose that stochastic Adam has an implicit bias toward a maximum margin solution w.r.t. a specific margin, determined by a data-dependent diagonal matrix. This result is vastly different from the previous work of implicit bias of full-batch Adam [1], and of great importance to the understanding of Adam.

[1] Zhang et al. The implicit bias of Adam on separable data. NeurIPS 2024

**Strengths:**

1. Considering that the real practice of Adam is usually mini-batch instead of full-batch, studying the implicit bias of stochastic Adam is of great importance.
2. The conclusions of this paper are fairly complete and abundant, covering the stochastic Adam and Signum.

**Weaknesses:**

1. **Strong technical assumptions**: The critical assumption that $\beta_2\to 1$ essentially makes the denominator of Adam's updates remain invariant for each data. From the theoretical perspective, such a simplification directly renders that stochastic Adam has the same updating direction as a specific preconditioned full-batch GD, which appears to be the direct reason for the derived implicit bias. The central question, however, is whether this assumption is justifiable for drawing conclusions about the true implicit bias of stochastic Adam in practical settings. This is particularly controversial because, even under the simple incremental setting, for any fixed $\beta_2 < 1$, the term $\beta_2^N$ decays dramatically with a large sample size $N$. Consequently, toy experiments with a minimal $N=10$ are insufficient and fail to substantiate the generalizability of the claimed conclusion. In addition, the assumption that Adamproxy has a convergent direction is also unsatisfactory. To some extent, combined with the connection between Adamproxy and preconditioned GD, the final conclusion is somewhat trivial, following the intuitive idea that only support vectors will contribute to the updates like [2]. I know that a similar assumption is also utilized in [3], but [3] does not study the simple linear classification settings, and hence acceptable.  Furthermore, many assumptions could be improved. For example, the assumption (a) in Theorem 3.3. should be rigorously established. In addition, could the author explain why the theoretical derivation for general data has to be built upon Adamproxy, instead of Adam, given that the remainder will converge to 0?

2. **Lack of convergence rate**: The mathematical derivation and calculations do not provide an exact convergence rate for the directional convergence.

3. **Potential issues in the proof**: In the proof of Proposition 4.3, authors finally conclude that $\\|\nabla L(w_t)\\|\_{P^{-1}(w_t)} \to 0$ since $\sum \eta_t \\|\nabla L(w_t)\\|\_{P^{-1}(w_t)}$ is bounded. However, the boundedness of $\sum a_n b_n$ and the divergence of $\sum a_n$ do not, in general, guarantee that $b_n \to 0$ (There exist many counter-examples such that $\lim\sup b_n >0$).  Authors should provide more details on how they derive such a conclusion.

[2]. Soudry et al. The implicit bias of gradient descent on separable data. JMLR.

[3]. Xie and Li. Adam exploits $\ell_\infty$-geometry of loss landscape via coordinate-wise adaptivity. ICML 2024.

**Questions:**

1. In fact, both the construction of the GR data and the assumption that $\beta_2 \to 0$ for general data ensure a critical point: within each epoch, all updates $w_{t+1} - w_t$ corresponding to different data points share the same adaptive term—namely, the denominator (please correct me if this observation is inaccurate). Based on this property, we can interchange the order of summations so that all data points contribute equally to the total update of one epoch. I am curious whether this property still holds if we consider randomly shuffling the data within each epoch, rather than processing them in a fixed order.
2. Moreover, I suggest that the authors revise their presentation regarding the implicit bias on the GR data, at least in the abstract. This result heavily depends on the specific construction of the data distribution, and I am somewhat skeptical whether it can be legitimately described as an implicit bias of Adam. Therefore, stating the $\ell_2$-margin conclusion in the abstract without explicitly mentioning its strong reliance on the particularly trivial data setup could be misleading, especially for readers who do not examine the full details.

---

> ### Author Response · Authors · 2025-11-23
> **Response to Reviewer yBsj (1/5)**
>
> We appreciate your thoughtful feedback. We would like to address your concerns and questions below. We look forward to discuss further if anything is unclear.
>
> ### W1: Strong Technical Assumptions
> > ... The central question, however, is whether this assumption is justifiable for drawing conclusions about the true implicit bias of stochastic Adam in practical settings. This is particularly controversial because, even under the simple incremental setting, for any fixed $\beta_2 <1$, the term $\beta_2^N$ decays dramatically with a large sample size $N$. Consequently, toy experiments with a minimal $N=10$ are insufficient and fail to substantiate the generalizability of the claimed conclusion. ...
>
> We acknowledge that our analysis of AdamProxy is restricted to the high-$\beta_2$ regime, and the _good enough_ choice of $\beta_2$ may depend on the sample size $N$. As discussed in Appendix A.2., extending the analysis for fixed $\beta_2<1$ setting requires techniques beyond our dual fixed-point framework. We view this as an important direction for future work.
>
> To evaluate the practical relevance of our approximation, we conducted additional experiments examining the extent to which AdamProxy reliably approximates Per-Sample Adam (including Incremental, Random Reshuffling, and With-Replacement Sampling) for moderate $\beta_2$. Specifically, we fix $(\beta_1, \beta_2)=(0.9, 0.95)$, and sample the dataset from Gaussian distribution changing the number of data points as $N \in \\{10, 20, 30, 40, 50\\}$ with $d=50$ and $N \in \\{100, 200, 300, 400, 500\\}$ with $d=500$. We repeat this for $(\beta_1, \beta_2)=(0.9, 0.99)$ on the latter datasets. We run Incremental, Random Reshuffling, With-Replacement Adam with LR schedule $\eta_t=\mathcal{O}(t^{-0.5})$ until the loss becomes lower than 1e-20 and calculate the final cosine similarity between the iterate and the fixed-point solution.
>
> **Table 1: Cosine similarity between the iterate and the fixed-point solution, $(\beta_1, \beta_2)=(0.9, 0.95)$, Gaussian data with $d=50$**
> $N$|10|20|30|40|50|
> -|:-:|:-:|:-:|:-:|:-:|
> Incremental|1.000|1.000|0.999|0.997|0.992
> Random Reshuffling|1.000|0.999|0.988|0.996|0.995
> With-Replacement|1.000|0.996|0.988|0.990|0.987
>
> **Table 2: Cosine similarity between the iterate and the fixed-point solution, $(\beta_1, \beta_2)=(0.9, 0.95)$, Gaussian data with $d=500$**
> $N$|100|200|300|400|500|
> -|:-:|:-:|:-:|:-:|:-:|
> Incremental|0.996|0.987|0.978|0.970|0.975
> Random Reshuffling|0.998|0.995|0.991|0.986|0.986
> With-Replacement|0.997|0.993|0.988|0.982|0.982
>
> **Table 3: Cosine similarity between the iterate and the fixed-point solution, $(\beta_1, \beta_2)=(0.9, 0.99)$, Gaussian data with $d=500$**
> $N$|100|200|300|400|500|
> -|:-:|:-:|:-:|:-:|:-:|
> Incremental|1.000|0.999|0.997|0.995|0.994
> Random Reshuffling|1.000|0.999|0.999|0.998|0.998
> With-Replacement|1.000|0.999|0.998|0.997|0.997
>
>
> Table 1 shows that our approximation via AdamProxy matches Incremental Adam and other Adam variants, especially when $N$ is relatively smaller than the effective window size $\frac{1}{1-\beta_2}$ (in this case, when $N\leq20$), with cosine similarity larger than 0.999. Furthermore, even for the relatively large $N>20$, our fixed-point analysis is predictive enough with cosine similarity $\approx 0.99$, justifying our $\beta_2\rightarrow 1$ approximation for the pratical choice of $(\beta_1, \beta_2)=(0.9, 0.95)$.
>
> When $N$ becomes even larger, the gap between the fixed-point solution and the converging direction increases. In Table 2, we observe that the cosine similarity is near 0.98 when $N=500$, demonstrating that the dynamics of Inc-Adam may go beyond the one of AdamProxy when $N$ becomes larger. This gap can be closed by taking $\beta_2=0.99$, which is also a common choice in practice (see Table 3).
>
> We expect that the good enough choice of $\beta_2$, where our approximation effectively holds, depends on the sample size $N$. To investigate the limiting behavior of Incremental Adam beyond the $\beta_2 \rightarrow 1$ regime, it requires the analysis for moderate $\beta_2<1$. Extending our theory to general $\beta_2$ is a promising direction for the complete analysis of per-sample Adam, while we leave it as a future work.
>
> Meanwhile, we want to mention that our AdamProxy type of approach under high-$\beta_2$ approximation is particularly reflective of _large-batch_ training setup in practice, where the effective horizon of EMA is larger than the number of mini-batches in a single epoch. In this light, our $\beta_2 \rightarrow 1$ approach offers valuable insights for future work analyzing adaptive methods in large-batch settings.

---

> ### Author Response · Authors · 2025-11-23
> **Response to Reviewer yBsj (2/5)**
>
> > ... In addition, the assumption that Adamproxy has a convergent direction is also unsatisfactory.
>
> We acknowledge that our analysis relies on directional convergence (Assumption 4.4), whereas Zhang et al. (2024) directly proved the convergence of full-batch Adam to the $\ell_\infty$-max-margin solution. However, we note that AdamProxy’s limiting behavior is inherently _data-dependent_, unlike the full-batch Adam setting analyzed by Zhang et al.
>
> Zhang et al. use a margin-gap argument: they prove that the gap between the $\ell_\infty$-max-margin and the normalized margin of the iterate vanishes. This approach is possible because full-batch Adam always converges to the $\ell_\infty$-max-margin classifier, independent of the dataset geometry.
>
> For AdamProxy, by contrast, the limiting direction depends on the dataset through a data-dependent preconditioner. This makes a margin-gap argument inapplicable unless one already knows which margin geometry the algorithm will converge to. Thus, we analyze AdamProxy under the directional convergence assumption to identify and characterize the limit direction in the first place. This assumption isolates the geometric structure of the limit without concerning an independent question of convergence itself.
>
> We view our results as establishing a foundational starting point for the implicit bias of adaptive optimizers in the mini-batch setup. Assumptions analogous to ours were also common in early implicit bias work. For example, Gunasekar et al. (2018) assumed loss convergence and directional convergence, which were later improved by follow-up works. Ji and Telgarsky (2020) additionally showed that gradient flow converges in direction for homogeneous networks under logistic loss, while extending such convergence guarantees to discrete-time gradient descent remains open. We expect that future work, building upon our framework, will relax or even remove Assumption 4.4.
>
>
> Zhang et al., "The implicit bias of adam on separable data", NeurIPS 2024
>
> Ji and Telgarsky, "Directional convergence and alignment in deep learning", NeurIPS 2020
>
> Gunasekar et al., "Implicit bias of gradient descent on linear convolutional networks", NeurIPS 2018.
>
>
> > To some extent, combined with the connection between Adamproxy and preconditioned GD, the final conclusion is somewhat trivial, following the intuitive idea that only support vectors will contribute to the updates like [2]. ...
>
> While AdamProxy can indeed be viewed as a form of preconditioned gradient descent, the key difference is that its preconditioner depends on the current iterate and evolves over training. Therefore, existing implicit bias analyses for preconditioned GD with fixed preconditioners do not apply.
>
> Lemma 4.5 illustrates this challenge: the limiting direction $\hat{w}$ is characterized by Eq. (5), where the coefficient $c_i$ and the set of indices of support vectors $S$ _depend on_ $\hat{w}$. Then we adopt our parametric optimization problem and dual fixed-point formulation to describe this solution (Theorem 4.8), which are the core novel ideas in our work. We highlight that this formulation is _nontrivial_ in the sense that it characterizes a data-adaptive converging direction; on SR data and shifted-diagonal data, the fixed-point solutions correspond to $\ell_2$- and $\ell_\infty$-max margin solution, respectively.

---

> ### Author Response · Authors · 2025-11-23
> **Response to Reviewer yBsj (3/5)**
>
> > Furthermore, many assumptions could be improved. For example, the assumption (a) in Theorem 3.3. should be rigorously established.
>
> We appreciate your suggestions on possible improvements in our result, while there are some technical challeges to directly prove the assumption (a) in Theorem 3.3 (loss convergence, $L(w_t) \rightarrow 0$). A typical approach to show that GD minimizes the loss is to show _descent lemma_; use Taylor 2nd order expansion and show the loss decreases monotonically.
>
> In our setting of weighted GD, the update rule can be regarded as GD with respect to a weighted loss $L^{\mathbf{a}(t)}(w)=\sum a_i(t) L_i(w)$ and learning rates $\eta_t$. By denoting $L^{\mathbf{a}(t)}(w)=L_t(w)$ and applying Taylor's expansion, we can get the following one-step descent lemma under small enough $\eta_t$.
>
> $$L_t(w_{t+1})-L_t(w_t)\leq-\frac{\eta_t}{2}\\|\nabla L_t(w_t)\\|_2^2$$
>
> However, since the loss $L_t$ changes over time $t$, we cannot guarantee the monotonic decrease of the weighted (or original) loss. Without such monotonicity, we can only deduce that $\lim \inf L(w_t)=0$ under mild assumptions (details in the next paragraph). Therefore, removing the assumption of loss convergence requires additional methodologies beyond our framework, which we leave as a future direction.
>
> To support our argument, we present some partial results showing $\lim\inf L(w_t)=0$ under mild assumptions. We recall that the weight $a_i(t)$ is bounded by constants $c,C>0$, i.e., $c\leq a_i(t)\leq C$. For notational convenience, let $L_C(w)=C\cdot\sum_{i \in [N]}L_i(w)$. Using the one-step bound and the convexity, we can deduce that
>
> $$\\|w_{t+1}-w\\|\_2^2 \leq \\|w_t-w\\|\_2^2+2\eta_t (L_t(w)-L_t(w_{t+1})).$$
>
> Assume that $\lim \inf L_t(w_t)=\epsilon>0$. Then we get $\lim \inf L_t(w_{t+1}) > \lim \inf \frac{c}{C}L_{t+1}(w_{t+1}) = \frac{c \epsilon}{C}$. Take $\bar{w}$ with $L_C(\bar{w}) < \frac{c \epsilon}{2C}$. By substituting to above inequality, we get
>
> $$\\|w_{t+1}-\bar{w}\\|\_2^2\leq\\|w_t-\bar{w}\\|\_2^2+2 \eta_t(L_t(\bar{w})-L_t(w_{t+1}))<\\|w_t-\bar{w}\\|\_2^2-\frac{c\epsilon}{C}\eta_t,$$
>
> for sufficiently large $t$. If $\sum\eta_t=\infty$, then it makes contradiction by telescoping above inequality to get $\\|w_{t+1}-\bar{w}\\|_2\rightarrow-\infty$. This proves that $\lim\inf L_t(w_t)=0$.
>
> > In addition, could the author explain why the theoretical derivation for general data has to be built upon Adamproxy, instead of Adam, given that the remainder will converge to 0?
>
> While Proposition 2.5. suggests a more _general_ proxy with arbitrary $\beta_1, \beta_2$ for Adam, analyzing such update rule is technically challenging. Even for RMSprop, where $\beta_1$ is chosen to be 0, analyzing its converging direction lies beyond our framework (more in Appendix A.2.). We therefore introduce AdamProxy, where we consider the limit of $\beta_2 \rightarrow 1$ and characterize its limiting behavior.

---

> ### Author Response · Authors · 2025-11-23
> **Response to Reviewer yBsj (4/5)**
>
> ### W2: Lack of Convergence Rate
> > The mathematical derivation and calculations do not provide an exact convergence rate for the directional convergence
>
> We acknolwedge that our paper mainly focuses on identifying implicit bias in an _asymptotic_ sense, rather than showing convergence rate of directional convergence. We describe why such an asymptotic analysis was inevitable in our analysis.
>
> Zhang et al. (2024) investigated the implicit bias of full-batch Adam on the same logistic regression setup, with exact convergence rate on the margin gap $\left|\min_{i \in [N]}\frac{\langle w_t, y_i x_i \rangle}{\|w_t\|_\infty} - \gamma \right|$. Their approach was based on _margin-based_ analysis, where they use descent lemma and directly bound the margin of current iterate. Originated from Gunasekar et al, the margin-based analysis has been adopted to prove the implicit bias of various optimizers (e.g. Fan et al.), including Section 5 in our paper.
>
> However, in case of our results, such a direct characterization is technically challenging. For Section 3 (SR data), the update of weighted GD is described by GD of weighted loss $L^a(w)$. Although we can show a similar descent lemma for the weighted loss $L^a(w)$, the coefficients $a(t)$ vary over time step $t$ and we cannot get a descent lemma for the _original_ loss $L(w)$. We therefore analyzed the converging direction with additional assumption of loss convergence, $L(w_t) \rightarrow 0$, which causes an asymptotic result.
>
> For Section 4 (AdamProxy), the limit direction is not described as a max-margin problem with an explicit form. Thus, we cannot apply previous margin-based analysis for AdamProxy (see our response to W1).
>
> For Section 5 (Signum), our analysis is based on margin-based approach, yielding a convergence-rate-type result (see Appendix G) as follows.
>
> $$\gamma_\infty - \frac{\min_{i\in [N]}\mathbf{w}\_t^\top \mathbf{x}\_i}{\|\mathbf{w}\_t\|\_\infty} \leq \mathcal O \left( \frac{\sum_{\tau = 0}^{t^\star-1}\eta_{\tau} + \sum_{\tau=t^\star}^t \eta_\tau e^{-\frac{\gamma_\infty}{8} \sum_{\tau=t^\star}^{t-1} \eta_\tau}+\sum_{\tau=t^\star}^{t-1} \eta_\tau^2}{ \sum_{\tau = 0}^{t-1}\eta_{\tau}} \right) + 2C_0(1-\beta)
> $$
>
> However, it is different from typical _convergence rate_, since the margin gap is upper-bounded with the constant $2C_0(1-\beta)$, which is caused by the misalignment between the full-batch gradient and the EMA of mini-batch gradients $\mathbf{m}\_t$. Therefore, we choose to present the asymptotic rate to emphasize that $\ell_\infty$-normalized margin of Signum iterates converges to $\ell_\infty$-max-margin with a constant margin gap, where the gap vanishes as $\beta$ goes to 1.
>
> Gunasekar et al., "Characterizing Implicit Bias in Terms of Optimization Geometry", ICML 2018
>
> Zhang et al., "The implicit bias of adam on separable data", NeurIPS 2024
>
> Fan et al., "Implicit Bias of Spectral Descent and Muon on Multiclass Separable Data", 2025.
>
> ### W3: Potential issues in the proof
> > In the proof of Proposition 4.3, authors finally conclude that $\\|\nabla L(w_t)\\|\_{P^{-1}(w_t)} \rightarrow 0$ since $\sum \eta_t \\|\nabla L(w_t)\\|_{P^{-1}(w_t)}$ is bounded. However, the boundedness of $\sum a_n b_n$ and the divergence of $\sum a_n$ do not, in general, guarantee that $b_n \rightarrow 0$ (There exist many counter-examples such that $\lim\sup b_n > 0$).
>
> Thank you for pointing out some logical gaps in our proof. We acknowledge that the explanation is not complete and the omitted steps are crucial to understand the conclusion. We revised the manuscript to incorporate the following steps.
>
> The key underlying point is that the loss $L(w_t)$ is **monotonically decreasing** and it is enough to show that $\lim \inf L(w_t)=0$ to prove the loss convergence. While the boundedness of $\sum \eta_t \\|\nabla L(w_t)\\|\_{P^{-1}(w_t)}$ and the divergence of $\sum \eta_t$ do not guarantee $\\|\nabla L(w_t)\\|\_{P^{-1}(w_t)} \rightarrow 0$, we can instead deduce that $\lim \inf \\|\nabla L(w_t)\\|\_{P^{-1}(w_t)}= 0$. Since $\\|\cdot \\|\_{P^{-1}(w_t)} \geq C\\|\cdot\\|_2$ for some constant $C>0$, we get $\lim\inf L(w_t)=0$, which proves the loss convergence.

---

> ### Author Response · Authors · 2025-11-23
> **Response to Reviewer yBsj (5/5)**
>
> ### Q1: Extension to Random-Shuffling
> > ... we can interchange the order of summations so that all data points contribute equally to the total update of one epoch. I am curious whether this property still holds if we consider randomly shuffling the data within each epoch, rather than processing them in a fixed order.
>
> First, our derivation to AdamProxy relies on Proposition 2.5, which is only satisfied when the order of summations is fixed throughout the training (i.e., incremental sampling setup). This implies that directly extending our results to random-shuffling setup requires additional analysis.
>
> For example, we consider $N=3$ data points with indices 1, 2, and 3, and provide a high-level intuition on why the fixed ordering is crucial. First, assume that the iterate $w_t$ does not change over time. Then for the mini-batch gradient $g_t$ at time step $t$, we get $g_1=g_4=\cdots, g_2=g_5=\cdots, g_3=g_6=\cdots$. Also, by taking $\beta_2 \rightarrow 1$, assume that the second momentum $v_t$ is approximated by
> $$v_t\approx(const)\sum_{i \in [N]} \nabla L_i(w_t)^2=(const)\sum_{i \in [N]}g_i^2,$$
>
> and does not change under the same epoch. (Note that these simplifications are rigorously established following the proof of Propositions 2.5 and 4.1.) Under these assumptions, for large $s$, we get
>
> $$\\begin{aligned}
> \frac{m_{3s}}{\sqrt{v_{3s}}}+\frac{m_{3s+1}}{\sqrt{v_{3s+1}}}+\frac{m_{3s+2}}{\sqrt{v_{3s+2}}} &\approx (const)\frac{\sum_{\tau=1}^{3s} \beta_1^\tau (1-\beta_1) g_{3s-\tau} + \sum_{\tau=1}^{3s+1} \beta_1^\tau (1-\beta_1) g_{3s+1-\tau} +\sum_{\tau=1}^{3s+2} \beta_1^\tau (1-\beta_1) g_{3s+2-\tau}}{\sqrt{\sum_{i \in [N]}g_i^2}}  \\\\
> &\approx (const) \frac{\sum_{\tau=1}^{3s} \beta_1^\tau (1-\beta_1) (g\_1 + g\_2 +g\_3)}{\sqrt{\sum_{i \in [N]}g_i^2}} \\\\
> &=(const)\frac{g_1+g_2+g_3}{\sqrt{g_1^2+g_2^2+g_3^2}},
> \\end{aligned}$$
> and this is why the numerator of AdamProxy is full-batch gradient, where the mini-batch gradients $g_1, g_2, g_3$ are **uniformly** accumulated within one epoch.
>
> In contrast, under random-shuffling, we can construct a worst-case where the sequence of orders induces different coefficients in the epoch-wise sum of the first momentum $m_t$. Consider 3 data points with indices 1, 2, and 3, and take the sequence of indices in random shuffling as (123 321 123 321 ...). Then, under the same assumptions, we can easily check that the epoch-wise sum $m_t+m_{t+1}+m_{t+2}$ is a **weighted** sum of 3 mini-batch gradients, which can deviate from AdamProxy update. Due to this complicated nature, we rather focused on analyzing incremental Adam where the coefficients are guaranteed to be identical if we consider the epoch-wise update.
>
> Our experiments, however, show that the sampling scheme (incremental/random-shuffling/with-replacement) does not crucially change the converging direction of Adam in the high-$\beta_2$ regime. We expect that random-shuffling Adam with batch size 1 shows a similar trajectory to AdamProxy _in expectation_ sense; while we can construct a worst case, the order of random shuffling is uniformly chosen across all permutations, making the effect of each data point identical. Developing this intuition of in-expectation update of random shuffling Adam into high-probability sense and connecting this to its limiting behavior would be a promising direction, but we leave this as a future work.
>
> ### Q2: Clarification of GR data
> > ... I suggest that the authors revise their presentation regarding the implicit bias on the GR data, at least in the abstract. ...
>
> We appreciate your thoughtful comments. To emphasize the role of Scaled Rademacher (SR) data (as an extreme case where incremental Adam converges to $\ell_2$-max margin solution), we revised our abstract in the revised manuscript.

---

### Official Review · Reviewer_v5Xn · 2025-10-22

**Soundness:** 3
**Presentation:** 3
**Contribution:** 3
**Rating:** 6
**Confidence:** 4

**Summary:**

This paper studies the implicit bias of Adam in the noisy setting and showed it may not converge to $\ell_\infty$-max-margin as deterministic Adam. The converged direction depends on the dataset structure and batch size.

**Strengths:**

1. The paper explicitly points out that previous literature only considers the implicit bias of Adam under deterministic setting and suggest that noise can have different influence on Adam and signum. It can expand the study of Adam.
2. There is one rigorous theoretical example on the Generalized Rademacher data that explicitly shows that Adam will converge $\ell_2$-max-margin.
3. There are many synthetic experiments that show the solution that stochastic Adam converges to is closer to $\ell_2$-max-margin, supporting the theoretical example and more general conjecture. The experiments also show that the converged solution of signum is never close to $\ell_2$-max-margin.
4. The experiments also show the key role of batch size in deciding the implicit bias.

**Weaknesses:**

1. The theory for stochastic Adam is not complete on general data. After the stage of theorem 4.8, the fixed point analysis can only be done by simulation of algorithm 3, which is less satisfying. Also the proposed AdamProxy almost reduces to AdaGrad. So it is not sure whether the analysis really suits for Adam, which works with a fixed $\beta_2<1$.
2. The assumption 4.4 is too strong, which requires the existence of the converged direction. In comparison, Zhang et al., 2024 doesn’t need to assume the existence but directly shows the convergence.
3. The example of Generalized Rademacher data is not general enough. I feel the result in section 3 can hold for any dataset such that the stochastic gradient $g_t$ always satisfying $|g_t[k]|=|g_t[l]|$ for any $k,l \in [d]$. However, this case is also too special in the sense that $v_t$ in Adam always satisfies $|v_t[k]|=|v_t[l]|$. The authors intentionally built the example to eliminate the coordinate-adaptivity of Adam but it also makes me wonder whether it can really help understand Adam in the general case.
4. It would be better to include batch size or noise scale in the framework. Since the figure 6 already shows that batch size 5 can already make the solution deviate from the incremental Adam, it is better to understand the effect the batch size intuitively through theoretical result.

**Questions:**

1. Do you think it is possible to get similar result without assumption 4.4? What is the necessary role of it if we can't remove it?
2. I only want to point out some possible experiments to explore, which might make the story more complete. But it won't affect my rating and doesn't need to be finished in the rebuttal process. I feel the batch size decides the balance between noise variance and deterministic gradient in $v_t$. When the batch size is small and noise dominates $v_t$, the update rule of Adam will become more similar with SGD in the sense that the effective learning rate is divided by $\sigma$. When the batch size is large and deterministic gradient square dominates $v_t$, the update rule will be more similar to deterministic setting. Therefore, the shape of the noise may be as important as batch size. Then it's worth to try the following experiments. Fix deterministic gradient function $g(x)$ and manually inject noise $\delta(t)$ so that the stochastic gradient is $g(x_t)+\delta(t)$. $\delta(t)$ can a random variable from Gaussian distribution and we can adjust the covariance matrix. We can multiply it by large constant or small constant and see how the converged direction of Adam changes under different scenario. For example, for what range of signal to noise ratio will the stochastic Adam converge closer to $\ell_2$-max-margin or $\ell_\infty$-max-margin? We can also try covariance matrix that is different from identity matrix. When the eigenvalues are heterogeneous, will incremental Adam still converge to $\ell_2$-max-margin?

---

> ### Author Response · Authors · 2025-11-23
> **Response to Reviewer v5Xn (1/3)**
>
> Thank you for your efforts and valuable feedback. We address your concerns and questions below.
>
> ### W1: Incomplete Theory on General Data
> > ... the fixed point analysis can only be done by simulation of algorithm 3, which is less satisfying. ...
>
> We acknowledge that our dual fixed-point formulation may appear _less satisfying_ without an explicit closed-form solution. However, such _implicit_ descriptions are standard in the literature on the implicit bias on separable data. For example, Lyu and Li (2020) prove that GD on a homogenous neural network converges to a _KKT point_ of a max-margin optimization problem; the limiting direction is characterized through optimality conditions, and obtaining it still requires solving a secondary optimization problem. Even the classical $\ell_2$-max-margin characterization is not explicit: identifying the direction requires running an algorithm to solve the SVM problem.
>
> Our dual fixed-point formulation follows this tradition. We characterize the limiting direction of AdamProxy as the solution of a _data-adaptive_ margin-maximization problem, induced by a Mahalanobis norm whose covariance matrix is determined by a _data-dependent_ dual fixed-point equation. Notably, our analysis establishes this covariance, and hence the implicit bias, depends on the dataset itself, rather than being fixed a priori. We further demonstrate two concrete data structures (Examples 4.9 and 4.11) where the implicit bias reduces to the classical $\ell_2$- and $\ell_\infty$-max margin solutions.
>
> To emphasize, our work provides a new implicit bias characterization in which the limiting direction is _data-dependent_ in this manner, which we view as a significant and conceptually novel contribution to understanding the implicit bias of adaptive methods with batch size 1.
>
> Lyu and Li, "Gradient descent maximizes the margin of homogeneous neural networks", ICLR 2020.
>
> >Also the proposed AdamProxy almost reduces to AdaGrad. So it is not sure whether the analysis really suits for Adam, which works with a fixed $\beta_2<1$.
>
> First, we clarify why AdamProxy should __not__ be interpreted as simply reducing to AdaGrad. Although both accumulate squared gradients, they are fundamentally different:
> -  AdaGrad uniformly accumulates all past gradients **starting from initialization (time step 1)**. Therefore, its preconditioner depends heavily on the entire history of (mini-batch) gradients, making the update direction at $w_t$ effected by early iterates $w_0,\ldots,w_{t-1}$.
> -  Incremental Adam (Inc-Adam) maintains an EMA of squared gradients. Proposition 2.5 shows that the influence of early gradient _vanishes_ (under Assumption 2.3), and the preconditioner becomes determined solely by the current iterate $w_t$. This key property allows AdamProxy to serve as an effective surrogate of Inc-Adam: AdamProxy accumulates squared gradients **only within the current epoch**, computed at the current iterate, not across the entire history.
>
>
> Also, we acknowledge that our analysis via AdamProxy is limited to the setting of $\beta_2\rightarrow 1$, and it only approximates Inc-Adam when $\beta_2$ is close enough to 1. Our experiments show that AdamProxy and Inc-Adam show a similar limiting behavior on a common choice of $\beta_2=0.95$, enhancing the predictive power of our result. How the suboptimal choice of $\beta_2$ changes the limit direction is a promising future direction (as discussed in Appendix A.2. in detail), while it is beyond the scope of our paper.

---

> ### Author Response · Authors · 2025-11-23
> **Response to Reviewer v5Xn (2/3)**
>
> ### W2 & Q1: More on Assumption 4.4
> > The assumption 4.4 is too strong, which requires the existence of the converged direction. In comparison, Zhang et al., 2024 doesn’t need to assume the existence but directly shows the convergence.
>
> We highlight that the implicit bias of AdamProxy shows a fundamentally **data-dependent** nature, in sharp contrast to the implicit bias of full-batch Adam. In particular, the analysis of Zhang et al. (2024) relies on a _margin based argument_, by showing the margin gap between the $\ell_\infty$-max-margin and the $\ell_\infty$-normalized margin of current iterate, which allows them to conclude convergence without assuming directional convergence. This is possible because full-batch Adam always converges in direction to a solution that maximizes the $\ell_\infty$-max-margin, regardless of the dataset.
>
> For AdamProxy, however, the limiting direction itself depends on the dataset through a data-dependent preconditioner. This significantly complicates the analysis: one cannot apply a margin-gap argument without first knowing _which_ margin (i.e., which geometry) the algorithm converges to. Thus, in contrast to Zhang et al., we require a directional convergence assumption to characterize the specific limit direction. This assumption allows us to isolate and analyze the geometry of the limit, rather than worrying about the question of whether convergence occurs.
>
> We view our results as establishing a foundational starting point for the implicit bias of adaptive optimizers in the mini-batch setup. Assumptions analogous to ours were also common in early implicit bias work. For example, Gunasekar et al. (2018) assumed loss convergence and directional convergence, which were later improved by follow-up works. Ji and Telgarsky (2020) additionally showed that gradient flow converges in direction for homogeneous networks under logistic loss, while extending such convergence guarantees to discrete-time gradient descent remains open. We expect that future work, building upon our framework, will weaken or even remove Assumption 4.4.
>
>
> Zhang et al., "The implicit bias of adam on separable data", NeurIPS 2024
>
> Ji and Telgarsky, "Directional convergence and alignment in deep learning", NeurIPS 2020
>
> Gunasekar et al., "Implicit bias of gradient descent on linear convolutional networks", NeurIPS 2018.
>
> > Do you think it is possible to get similar result without assumption 4.4? What is the necessary role of it if we can't remove it?
>
> As discussed above, the dynamics of AdamProxy is highly nontrivial due to their data-dependent geometry, and some form of regularity on the iterates is needed to make the analysis tractable. Assumption 4.4 (directional convergence) allows us to focus entirely on characterizing the limit direction, without concerning _when_ the directional convergence happens.
>
> We also note that one might consider an alternative approach: viewing AdamProxy as a normalized steepest descent under a Mahalanobis-norm induced by the matrix $P(w_t)$ which depends on the current iterate (see Lemma F.1. in Appendix F.2.). If we assume the inducing matrix converges to a fixed matrix $P$, then margin-based analysis similar to Zhang et al. can be made. However, it is more common to assume the directional convergence in the implicit bias literature, and we decide to take this route.
>
> ### W3: Limited Scope of Generalized Rademacher Data
> > ... I feel the result in section 3 can hold for any dataset such that the stochastic gradient $g_t$ always satisfying $|g_t[k]|=|g_t[l]|$ for any $k,l\in[d]$. However, this case is also too special ... it also makes me wonder whether it can really help understand Adam in the general case.
>
> We acknowledge that the setting of Generalized Rademacher data is not _general_ enough. Rather, we study GR data as an extreme case which describes how incremental sampling can make the limiting behavior of Adam deviate from its full-batch counterpart.
>
> To prevent possible misconceptions, we have changed the term _Generalized Rademacher_ to _Scaled Rademacher_ to give more emphasis on the structural aspect of the dataset in the revised manuscript. Also we revised the abstract to further emphasize the role of our analysis on SR data.

---

> ### Author Response · Authors · 2025-11-23
> **Response to Reviewer v5Xn (3/3)**
>
> ### W4: More on Batch Size and Noise Scale
> > It would be better to include batch size or noise scale in the framework. ... it is better to understand the effect the batch size intuitively through theoretical result.
>
> We agree that incorporating the batch size (or noise scale) in our theory would strengthen our contribution. In particular, we can extend Proposition 2.5. to incorporate the choice of batch size $b$ as
>
> $$
> w_{r+1}^0-w_r^0=-\eta_{rN}\left(C_b(\beta_1, \beta_2) \sum_{i\in[N/b]}\frac{\sum_{j \in [N/b]}\beta_1^{(i,j)}\sum_{k \in [b]} \nabla L_{jb+k}(w_r^0)}{\sum_{j \in [N/b]}\beta_2^{(i,j)}(\sum_{k \in [b]} \nabla L_{jb+k}(w_r^0))^2} + \boldsymbol{\epsilon}_r\right),
> $$
> where $\beta_1^{(i,j)} = \beta_1^{(i-j) \bmod (N/b)}$ and $\beta_2^{(i,j)} = \beta_2^{(i-j) \bmod (N/b)}$.
>
> Yet, our core results including the limit direction on SR data and fixed-point formulation do not naturally extend to this setup. Even on the simplest SR data, choosing batch size larger than 1 does not eliminate coordinate-adaptivity and induces different dynamics. Therefore, investigating the effect of the batch size would be a promising direction, while we leave it as a future work.
>
> ### Q2: More Experiments on Noise Structure
> > For example, for what range of signal to noise ratio will the stochastic Adam converge closer to $\ell_2$-max-margin or $\ell_\infty$-max-margin?
>
> > When the eigenvalues are heterogeneous, will incremental Adam still converge to $\ell_2$-max-margin?
>
> We appreciate your insightful suggestions on additional experiments. Before we present some preliminary results, we want to point out the difference between batch sampling and injecting noise to the true gradient, and highlight why manually injecting Gaussian noises may not be necessarily helpful for understanding the implicit bias.
>
> In logistic regression task, the gradient $\nabla L(w)$ vanishes to 0 as the loss $L(w)$ converges to 0. This implies that when the loss converges, noise dominates signal at the late phase of training _regardless of the magnitude of the noise_. This causes an intractable convergence direction, even in SGD case.
>
> We sample the dataset from Gaussian distribution with $N=10, d=50$. Then we run 5 optimizers: full-batch GD, full-batch Adam, Incremental Adam, noisy GD and noisy Adam (for the last two, full-batch gradient + gaussian noise is used). We use constant learning rates for GD family and use decaying learning rates for Adam family. The result is presented in https://anonymous.4open.science/r/Figures-F58E. In Figure 1, we observe that noisy GD shows a chaotic behavior, failing to converge to the theoretically expected $\ell_2$-max-margin solution. Since mini-batch GD provably converges to the same direction as full-batch GD (Nacson et al.), this result highlights that investigating the implicit bias with batch sampling is more natural than investigating with manually injected stochastic noise. For the case of noisy Adam, its behavior is seemingly more stable, but Figure 2 reveals that its loss convergence is much slower than other methods. This suggests that noisy Adam may have not yet entered the regime where the injected noise dominates, and it could show a similar chaotic behavior in the later stage of training.
>
> A promising future direction would be to _scale the noise_ with respect to current loss and to investigate the limit direction. Under this setting, the structure of noise (e.g. covariance matrix of the noise) might determine the implicit bias, while we leave this as a future work.
>
> Nacson et al., "Stochastic gradient descent on separable
> data: Exact convergence with a fixed learning rate", AISTATS 2019.

---

> > ### Comment · Reviewer_v5Xn · 2025-11-27
> >
> > Thanks for answering my questions and conducting the experiments. The paper's statement and contribution become more clear with the explanation and updates in the paper. I'd like to raise my score.

---

> > > ### Author Response · Authors · 2025-11-28
> > >
> > > Thank you for your thoughtful reconsideration and positive evaluation. We truly appreciate your efforts and constructive feedback throughout.

---

### Official Review · Reviewer_f1uy · 2025-10-29

**Soundness:** 3
**Presentation:** 3
**Contribution:** 3
**Rating:** 8
**Confidence:** 4

**Summary:**

This paper studies the implicit bias of Adam for binary classification on linearly separable data, particularly the effect of batch size on the learned solution. Building on prior work that shows that iterates of full-batch Adam directionally converge to the $\ell_\infty$-max-margin solution, this work shows empirically, and theoretically, under some conditions, that for batch size 1, this is not the case: incremental Adam (using one sample every step) converges to the $\ell_2$-max-margin solution for some dataset class, while for general datasets, it characeterizes the convergence of proxy-Adam (limit $\beta_2\to 1$) using a fixed-point framework. Additionally, the paper shows that other variants of signGD may not share this behaviour: Signum, in the limit of momentum parameters tending to 1, converges to the $\ell_\infty$-max-margin solution for any batch size.

**Strengths:**

1. The paper is well-written and easy to follow, overall.

2. It studies an important problem (what is the effect of batch size on the implicit bias of Adam?), characterizes the implicit bias of per-sample Adam under some conditions and provides interesting insights about how can be different compared to standard max-margin solutions .

**Weaknesses:**

There are a few weaknesses that the paper should address:

1. The paper is missing discussion on some related works that study the implicit bias of Adam in neural networks [1-2], and effect of momentum parameter values [3] and rotations [4]. There is also another paper [5] on effect of batch sizes on SGD and Adam that should be cited in Section 7.

2. In Fig. 1 (left), the cosine similarity of Adam iterates with the $\ell_2$-max-margin solution does not converge to 1. This is later clarified in Fig. 3 (and Section 4). However, it could be useful to either i) add another panel in Fig. 1 (similar to Fig. 3 (right)), or ii) add a pointer to Fig. 3 or Section 4 in Fig. 1 caption. Relatedly, the Introduction mainly discusses how per-sample Adam converges “closer to $\ell_2$-max-margin solution”, and it could be helpful to further clarify this early on, e.g., adding something similar to line 361 in point 3 of the Contributions.

**References:**

[1] Tsilivis et al., “Flavors of margin: Implicit bias of steepest descent in homogeneous neural networks”, ICLR 2025.

[2] Vasudeva et al., “The Rich and the Simple: On the Implicit Bias of Adam and SGD”, NeurIPS 2025.

[3] Orvieto and Gower, “In Search of Adam’s Secret Sauce”, NeurIPS 2025.

[4] Zhang et al., “Understanding Adam Requires Better Rotation Dependent Assumptions”, NeurIPS 2025.

[5] Marek et al., “Small Batch Size Training for Language Models: When Vanilla SGD Works, and Why Gradient Accumulation Is Wasteful”, NeurIPS 2025.

**Questions:**

Some suggestions to further improve clarity are as follows:

1. It should be clarified that for vectors, the division is element-wise (e.g., Eq. 2). Relatedly, it can be mentioned in the proof of Cor. 3.2, that shared elements in the denominator allow moving to common division rather than element-wise.

2. Omit ‘the’ in line 211.

3. More pointers can be added to the Appendix in the main body. For instance, references to proofs, and the additional experiments would be helpful.

---

> ### Author Response · Authors · 2025-11-23
> **Response to Reviewer f1uy**
>
> Thank you for your thoughtful comments. We appreciate that **you found our work provides interesting insights characterizing the implicit bias of per-sample Adam**. We have addressed your feedbacks in the revised manuscript as below.
>
> ### Reponse to Weaknesses
> > The paper is missing discussion on some related works that study the implicit bias of Adam ...
>
> We added more discussions in Related Works to include [1-5] of your comment.
>
> > ... it could be useful to either i) add another panel in Fig. 1 (similar to Fig. 3 (right)), or ii) add a pointer to Fig. 3 or Section 4 in Fig. 1 caption. ... it could be helpful to further clarify this early on, e.g., adding something similar to line 361 in point 3 of the Contributions.
>
> Thank you for the detailed feedback. We added a pointer to Section 4 in Fig.1 caption. We also revised some sentences in the Contributions paragraph.
>
> ### Response to Questions
> > It should be clarified that for vectors, the division is element-wise (e.g., Eq. 2). Relatedly, it can be mentioned in the proof of Cor. 3.2, that shared elements in the denominator allow moving to common division rather than element-wise.
>
> > Omit ‘the’ in line 211.
>
> > More pointers can be added to the Appendix in the main body. For instance, references to proofs, and the additional experiments would be helpful.
>
> Thank you for the detailed suggestions. We revised our manuscripts to incorporate your comments.

---

> > ### Comment · Reviewer_f1uy · 2025-11-27
> >
> > Thanks for the responses and updates, I will maintain my score.

---

> > > ### Author Response · Authors · 2025-11-28
> > >
> > > Thank you for your careful feedback and positive evaluation. We sincerely appreciate your time and effort in reviewing our paper.

---

### Author Response · Authors · 2025-11-23
**Revision of Paper**

Dear reviewers and AC(s),

We thank the reviewers for their helpful feedback and appreciation of our work! In the individual responses, we address any remaining concerns.

The main concerns were raised from the technical assumptions used in our analysis, particularly the $\beta_2  \rightarrow 1$ approximation and the directional convergence of AdamProxy. While these assumptions do impose limitations, we emphasize that **our work provides a novel implicit bias characterization in which the limiting direction is data-dependent**. We believe our result marks an important first step toward understanding the substantially richer implicit bias of adaptive methods in the mini-batch (with batch size 1) regime—an area that has remained underexplored.

In the individual responses, we clarify why these assumptions facilitate tractable analysis and are also reasonable in light of prior literature on implicit bias.

We have also incorporated your constructive suggestions into our revision. In updated manuscript, all newly added/modified sentences and paragraphs are marked in red.


### Some key changes
- We rename Generalized Rademacher (GR) to Scaled Rademacher (SR) to emphase the structural aspect of the class of datasets.
- We revise the abstract to emphasize the role of SR data and our main contribution: characterizing a _data-dependent_ convergence direction.
- We add more discussions on related works.
- We add more explanations and reconstruct some proofs in the appendix for better readability.

---

### Meta-Review · Area_Chair_vVCU · 2025-12-31

**Summary:**

The paper investigates the effect of batch size on the implicit bias of Adam. The problem is important and relevant. Whereas Adam on linearly separable data has an implicit bias towards an L_infinity max-margin solution, the paper shows that for certain data full-batch Adam has a bias towards an L_2 max-margin solution in the extreme case of single sample batch size.

The paper had some variance in the reviewer scores, with two quite positive reviews (after discussion) and one negative review. On the positive side, all the reviewers generally felt that the problem is important and the investigation in the paper is quite thorough. The major negative comments were about some of the technical assumptions (Reviewer v5Xn and yBsj), and a concern about the proof (Reviewer yBsj).

As discussed below, I believe that the concerns are sufficiently addressed. The authors have also undertaken several changes in the manuscript to account for the reviewer's feedback and concern. Therefore, I recommend that the paper should be accepted since it is likely to be of significant interest to the community.

**Reviewer Concerns:**

I believe that the issue in the proof of Proposition 4.3 raised by Reviewer yBsj has been addressed by the authors by adding intermediate steps which preserve the original claim.

Regarding the strength of the assumptions, I am satisfied with the authors explanation regarding Assumption 4.4 on the existence of the converged direction. Reviewer v5Xn also mentioned in the discussion that they were convinced. I think some of the other assumptions are reasonable as a first step to understanding the implicit bias in the small batch size regime.

**Reviewer Scores:**

Reviewer v5Xn would likely have increased their score, in fact they said in their comment that they will. Reviewer v5Xn could also have increased their score.

---

### Decision · Program_Chairs · 2026-01-26

Accept (Poster)